# Data- and Variance-dependent Regret Bounds for Online Tabular MDPs

**Mingyi Li** [1]  **Taira Tsuchiya** [1,2]  **Kenji Yamanishi** [1]

## Abstract

This work studies online episodic tabular Markov decision processes (MDPs) with known transitions and develops best-of-both-worlds algorithms that achieve refined data-dependent regret bounds in the adversarial regime and variance-dependent regret bounds in the stochastic regime. We quantify MDP complexity using a first-order quantity and several new data-dependent measures for the adversarial regime, including a second-order quantity and a path-length measure, as well as variance-based measures for the stochastic regime. To adapt to these measures, we develop algorithms based on global optimization and policy optimization, both built on optimistic follow-the-regularized-leader with log-barrier regularization. For global optimization, our algorithms achieve first-order, second-order, and path-length regret bounds in the adversarial regime, and in the stochastic regime, they achieve a variance-aware gap-independent bound and a variance-aware gap-dependent bound that is polylogarithmic in the number of episodes. For policy optimization, our algorithms achieve the same data- and variance-dependent adaptivity, up to a factor of the episode horizon, by exploiting a new optimistic $Q$-function estimator. Finally, we establish regret lower bounds in terms of data-dependent complexity measures for the adversarial regime and a variance measure for the stochastic regime, implying that the regret upper bounds achieved by the global-optimization approach are nearly optimal.

## 1. Introduction

We study online learning in finite-horizon episodic tabular Markov decision processes (MDPs), a standard model

in reinforcement learning with broad applications, such as robotics (Schulman et al., 2017), games (Mnih et al., 2015), and healthcare decision-making (Komorowski et al., 2018). In this setting, a learner interacts with an environment over $T$ episodes. In each episode, the learner selects a distribution over actions at each state, follows the trajectory induced by the algorithm, and observes the losses incurred along that trajectory. The goal is to minimize regret, defined as the difference between the learner's cumulative expected loss and that of the best fixed policy in hindsight.

Tabular MDP algorithms are typically built on either *global optimization* or *policy optimization*. Global optimization solves an optimization problem over the set of all occupancy measures and can achieve minimax-optimal regret guarantees (Zimin & Neu, 2013; Jin et al., 2020), but it can be computationally demanding for large MDPs. Policy optimization updates an action distribution at each state, which is often practical and computationally efficient, and the per-state updates can be viewed as instances of multi-armed bandits (Shani et al., 2020; Luo et al., 2021).

The difficulty of tabular MDPs depends on how the underlying loss sequence is generated. In the adversarial regime, where losses may be chosen arbitrarily, the minimax-optimal regret typically scales as $\widetilde{O}(\sqrt{T})$ (Jin et al., 2020; Luo et al., 2021), where $T$ is the number of episodes. By contrast, in the stochastic regime with i.i.d. losses, one can achieve much faster gap-dependent regret, typically $O(\log T)$ (Simchowitz & Jamieson, 2019).

Recent work has shown that these regret upper bounds can be improved in various ways to better adapt to the structure of MDPs. One line of work develops *best-of-both-worlds* algorithms, which aim to achieve near-optimal regret in both the adversarial and stochastic regimes with a single algorithm (Jin & Luo, 2020; Jin et al., 2021; Dann et al., 2023a), thereby bridging the gap between the two regimes. As another example, in the adversarial regime, one can derive regret bounds that depend on *first-order complexity measures*: when the optimal policy has a small value function, this benign property yields improved guarantees (Lee et al., 2020; Dann et al., 2023a). Furthermore, in the stochastic regime, variance-aware algorithms have been actively studied, including those with *gap-independent* regret bounds (Zanette & Brunskill, 2019; Zhang et al., 2021) and

---

[1]Department of Mathematical Informatics, The University of Tokyo, Tokyo, Japan [2]RIKEN AIP, Tokyo, Japan. Correspondence to: Mingyi Li <mingyi-mike@g.ecc.u-tokyo.ac.jp>.

*Proceedings of the 43rd International Conference on Machine Learning*, Seoul, South Korea. PMLR 306, 2026. Copyright 2026 by the author(s).

*Table 1.* Comparison of regret upper bounds based on global optimization. Here, $U = \sum_s \sum_{a \neq \pi^\star(s)} \frac{H^2 \log(T)}{\Delta(s,a)}$ and $U_{\mathsf{Var}} = \sum_s \sum_{a \neq \pi^\star(s)} \frac{H \mathbb{V}^c(s) \log(T)}{\Delta(s,a)}$. We only display leading terms and omit logarithmic and lower-order factors.

| Reference | Adversarial regime | Stochastic regime with adversarial corruption |
|---|---|---|
| Zimin & Neu (2013) | $\sqrt{HSAT}$ | $\sqrt{HSAT}$ |
| Lee et al. (2020) | $\sqrt{SAL^\star}$ | $\sqrt{SAL^\star}$ |
| Jin et al. (2021) | $\sqrt{HSAT}$ | $U_{\mathsf{Jin}} + \sqrt{U_{\mathsf{Jin}}\mathcal{C}}$ $\ (U_{\mathsf{Jin}} = U + \frac{H^3 S \log(T)}{\min_{s, a \neq \pi^\star(s)} \Delta(s,a)})$ |
| **This work** (Theorem 4.1) | $\sqrt{SA \min\{L^\star, HT - L^\star, Q_\infty, V_1\}}$ | $\min\{\sqrt{SA(\mathbb{V}T + \mathcal{C})},\ U + \sqrt{U\mathcal{C}}\}$ |
| **This work** (Theorem 4.2) | $\sqrt{SA \min\{L^\star, HT - L^\star, Q_\infty\}}$ | $\min\{\sqrt{SA(\mathbb{V}T + \mathcal{C})},\ U_{\mathsf{Var}} + \sqrt{U_{\mathsf{Var}}\mathcal{C}}\}$ |

*Table 2.* Comparison of regret upper bounds based on policy optimization. Here, $U = \sum_s \sum_{a \neq \pi^\star(s)} \frac{H^2 \log^2(T)}{\Delta(s,a)}$ and $U_{\mathsf{Var}} = \sum_s \sum_{a \neq \pi^\star(s)} \frac{H \mathbb{V}^c(s) \log^2(T)}{\Delta(s,a)}$. We only display leading terms and omit logarithmic and lower-order factors.

| Reference | Adversarial regime | Stochastic regime with adversarial corruption |
|---|---|---|
| Luo et al. (2021) | $\sqrt{H^3 SAT}$ | $\sqrt{H^3 SAT}$ |
| Dann et al. (2023a) | $\sqrt{H^2 SAL^\star}$ | $U + \sqrt{U\mathcal{C}}$ |
| **This work** (Theorem 5.2) | $\sqrt{H^2 SA \min\{L^\star, HT - L^\star, Q_\infty, V_1\}}$ | $\min\{\sqrt{H^2 SA(\mathbb{V}T + \mathcal{C})},\ U + \sqrt{U\mathcal{C}}\}$ |
| **This work** (Theorem 5.3) | $\sqrt{H^2 SA \min\{L^\star, HT - L^\star, Q_\infty\}}$ | $\min\{\sqrt{H^2 SA(\mathbb{V}T + \mathcal{C})},\ U_{\mathsf{Var}} + \sqrt{U_{\mathsf{Var}}\mathcal{C}}\}$ |

those with *gap-dependent* regret bounds (Simchowitz & Jamieson, 2019; Chen et al., 2025) with polylogarithmic dependence on $T$.

Despite these developments, existing algorithms remain unsatisfactory. First, the adaptive guarantees above are typically achieved by different algorithms; in practice, the environment is unknown a priori, making it difficult to choose the most suitable algorithm in advance. Moreover, in adversarial tabular MDPs, the only known data-dependent guarantees are first-order bounds. This contrasts with the broader online learning literature, which studies many other data-dependent guarantees, including *second-order bounds* that adapt to the magnitude of loss fluctuations and *path-length bounds* that adapt to how much losses change over time (see, *e.g.,* Cesa-Bianchi et al. 1996; Allenberg et al. 2006; Neu 2015; Wei & Luo 2018; Bubeck et al. 2019). This naturally raises the following question:

*Can we design a single best-of-both-worlds algorithm for tabular MDPs that achieves first-order, second-order, and path-length bounds in the adversarial regime and achieves variance-dependent bounds that are gap-independent or gap-dependent in the stochastic regime?*

To address this question, we focus on the known-transition setting. This lets us separate the difficulty of loss estimation, which is already nontrivial under bandit feedback. Indeed, even when the transition kernel is known, losses are observed only along realized trajectories, and estimation errors at a state-action pair can affect downstream value estimates through the MDP dynamics. Therefore, unlike in multi-armed bandits, these errors cannot be controlled independently across state-action pairs; we also need to account for how they affect later estimates. This makes it necessary to design loss and $Q$-function estimators whose biases are compatible with refined data-dependent complexity measures.

The unknown-transition setting is also important, but it is beyond the scope of this paper. Handling it would require data-dependent control of transition-estimation errors in addition to the loss-estimation errors studied here. For global optimization, Lee et al. (2020) derive first-order bounds under unknown transitions, but extending these techniques to second-order, path-length, variance-dependent, or best-of-both-worlds guarantees remains open. For policy optimization, even first-order data-dependent guarantees under unknown transitions remain open (Dann et al., 2023a).

**Contributions of This Paper**

In Section 3, we begin by introducing new data-dependent complexity measures. Specifically, we introduce a second-order quantity $Q_\infty$, which captures how large the losses can fluctuate, as well as a path-length measure $V_1$, which quantifies how much the losses change over time. In addition, to quantify variance of MDPs in the stochastic regime, we introduce the occupancy-weighted variance $\mathbb{V}$ and the conditional occupancy-weighted variance $\mathbb{V}^c$ (see (4)–(8) for detailed definitions).

In Section 4, we first develop global optimization algo-

*Table 3.* Regret lower bounds for online episodic tabular MDPs. Note that the lower bounds in terms of $L^\star$, $Q_\infty$, and $V_1$ are constructed for adversarial instances.

| Reference | Lower bound |
|---|---|
| Zimin & Neu (2013) | $\Omega(\sqrt{HSAT})$ |
| **This work** (Section 6) | $\Omega(\sqrt{SAL^\star})$, $\Omega(\sqrt{SAQ_\infty})$, $\Omega(\sqrt{HV_1})$, $\Omega(\sqrt{SA\mathbb{V}T})$ |

rithms whose regret adapts to the data-dependent complexity measures introduced above. They achieve a regret upper bound of $\widetilde{O}(\sqrt{SA\min\{L^\star, HT - L^\star, Q_\infty, V_1\}})$ in the adversarial regime, as well as a variance-aware gap-independent regret bound of $\widetilde{O}(\sqrt{SA\mathbb{V}T})$ and a variance-aware gap-dependent regret bound of $\mathrm{polylog}(T)$ in the stochastic regime[1] (see Theorems 4.1 and 4.2). To our knowledge, these are the first second-order and path-length bounds for online episodic tabular MDPs. Moreover, our gap-dependent bound in the stochastic regime improves over Jin et al. (2021) by adapting the variance and avoiding their additional dependence on $1/\min_{s,a} \Delta(s,a)$. The algorithms are based on optimistic follow-the-regularized-leader (OFTRL) over the set of all occupancy measures with a log-barrier regularizer and an adaptive learning rate. See Table 1 for a detailed comparison.

In Section 5, we develop policy optimization-based algorithms, which achieve a regret upper bound of $\widetilde{O}(\sqrt{H^2SA\min\{L^\star, HT - L^\star, Q_\infty, V_1\}})$ in the adversarial regime, as well as a gap-independent variance-dependent regret bound of $\widetilde{O}(\sqrt{H^2SA\mathbb{V}T})$ and a gap-dependent variance-dependent regret bound of $\mathrm{polylog}(T)$ in the stochastic regime (see Theorems 5.2 and 5.3). See Table 2 for a detailed comparison. The algorithms are also based on OFTRL with a log-barrier regularizer. A particularly notable ingredient is that, to correct a bias induced by the loss predictions in OFTRL, we introduce an even more optimistic $Q$-function estimation scheme than the one used in the existing best-of-both-worlds policy optimization by Dann et al. (2023a) (see Section 5.1 for details).

Finally, in Section 6, we derive data-dependent regret lower bounds of $\Omega(\sqrt{SAL^\star})$, $\Omega(\sqrt{SAQ_\infty})$, and $\Omega(\sqrt{HV_1})$, as well as a variance-dependent lower bound of $\Omega(\sqrt{SA\mathbb{V}T})$. This implies that our regret upper bound for global optimization is nearly optimal in terms of $L^\star$, $Q_\infty$, and $V_1$. See Table 3 for a summary. Due to space limitations, we defer additional related work on MDPs, best-of-both-worlds algorithms, and data-dependent analyses in the adversarial

and stochastic regimes to Appendix B.1.

## 2. Preliminaries

**Notation.** For $N \in \mathbb{N}$, let $[N] \coloneqq \{1, 2, \ldots, N\}$. Given a vector $x$, we write $\|x\|_p$ to denote the $\ell_p$-norm for $p \in [1, \infty]$. The set $\Delta(\mathcal{K})$ denotes the set of all probability distributions over the set $\mathcal{K}$, and the indicator function $\mathbb{1}[\cdot]$ returns 1 if the specified condition holds and 0 otherwise. For sets $\mathcal{A}$ and $\mathcal{B}$, we use $\mathcal{A}^\mathcal{B}$ to denote the set of all functions from $\mathcal{B}$ to $\mathcal{A}$. Given functions $f$ and $g$ with $g(x) > 0$, we write $f \lesssim g$ or $f = O(g)$ if there exists a constant $c > 0$ such that $f(x) \leq cg(x)$ for all $x$ in the relevant domain and $\widetilde{O}(\cdot)$ hides logarithmic factors.

**Episodic tabular MDPs.** We consider a finite-horizon episodic tabular Markov Decision Process (MDP) $\mathcal{M} = (\mathcal{S}, \mathcal{A}, P, H, s_0)$, where $\mathcal{S}$ is a finite state space with $S = |\mathcal{S}|$, $\mathcal{A}$ is a finite action space with $A = |\mathcal{A}|$, and $P : \mathcal{S} \times \mathcal{A} \to \Delta(\mathcal{S})$ is a known transition function. Here, $P(s' \mid s, a)$ specifies the probability of transitioning to state $s'$ after taking action $a$ in state $s$. We adopt the standard layered MDP assumption (Neu et al., 2010; Jin et al., 2020; Luo et al., 2021) that the state space is layered into $H + 1$ disjoint sets $\mathcal{S}_0, \mathcal{S}_1, \ldots, \mathcal{S}_H$, where $\mathcal{S}_0 = \{s_0\}$ is the initial layer and $\mathcal{S}_H = \{s_H\}$ is a terminal absorbing layer. For simplicity, we exclude $s_H$ from $\mathcal{S}$ and note that $H \leq S$. Transitions are restricted to proceed from one layer to the next: for any $(s, a) \in \mathcal{S}_h \times \mathcal{A}$ with $h \in \{0, \ldots, H-1\}$, the distribution $P(\cdot \mid s, a)$ is supported only on $\mathcal{S}_{h+1}$. We write $h(s)$ for the layer index of state $s$. The learning proceeds for $T$ episodes indexed by $t = 1, \ldots, T$. At the beginning of episode $t$, the environment chooses a loss function $\ell_t : \mathcal{S} \times \mathcal{A} \to [0, 1]$. A policy $\pi : \mathcal{S} \to \Delta(\mathcal{A})$ assigns a distribution over actions to each state $s$, with $\pi(a \mid s)$ denoting the probability of action $a$ at state $s$. The set of all stochastic policies is denoted by $\Pi = \Delta(\mathcal{A})^\mathcal{S}$, and the set of deterministic policies by $\Pi_\mathsf{det} = \mathcal{A}^\mathcal{S}$. When $\pi$ is deterministic, we write $\pi(s) \in \mathcal{A}$ for the unique action chosen in $s$. We assume $T \geq \max\{2, S, A\}$ for convenience.[2]

For a policy $\pi$ and a loss function $\ell$, we define the value functions recursively with the terminal condition $V^\pi(s_H; \ell) = 0$. The state value function $V^\pi(s; \ell)$ and the state-action value function $Q^\pi(s, a; \ell)$ (a.k.a. $Q$-function) are defined as $V^\pi(s; \ell) = \mathbb{E}_{a \sim \pi(\cdot|s)}[Q^\pi(s, a; \ell)]$ and $Q^\pi(s, a; \ell) = \ell(s, a) + \mathbb{E}_{s' \sim P(\cdot|s,a)}[V^\pi(s'; \ell)]$. Here we may overload the notation by allowing a general function $m : \mathcal{S} \times \mathcal{A} \to \mathbb{R}$ to replace the loss function $\ell$, and write $V^\pi(s; m)$ and $Q^\pi(s, a; m)$ accordingly.

---

[1]Precisely speaking, for both global optimization and policy optimization, whether we can attain a path-length bound or a variance-aware gap-dependent bound depends on how the loss prediction in OFTRL is chosen (see Tables 1 and 2).

[2]The assumptions $T \geq S$ and $T \geq A$ are not essential. If they do not hold, the analysis remains valid with $\log(T)$ replaced by $\log(SAT)$ or $\log(AT)$.

In each episode $t \in [T]$, the learner chooses a policy $\pi_t$ based on past observations, executes it from the initial state $s_0$, and observes the losses along the realized trajectory $\{(s_{t,h}, a_{t,h}, \ell_t(s_{t,h}, a_{t,h}))\}_{h=0}^{H-1}$. The goal of the learner is to minimize the regret given by

$$\mathrm{Reg}_T = \max_{\pi \in \Pi} \mathbb{E}\left[\sum_{t=1}^T V^{\pi_t}(s_0; \ell_t) - \sum_{t=1}^T V^\pi(s_0; \ell_t)\right],$$

and denote by $\mathring{\pi}$ one of the optimal fixed comparators for the regret definition.

For a policy $\pi$ and a state-action pair $(s, a)$, the occupancy measure $q^\pi(s, a)$ is the probability of visiting $(s, a)$ within an episode under $\pi$. We also use $q^\pi(s' \mid s, a)$ and $q^\pi(s', a' \mid s, a)$ for the corresponding conditional occupancy measures given that $(s, a)$ has already been visited (note that these quantities are zero whenever $h(s') < h(s)$). For each state $s$, we set $q^\pi(s) := \sum_a q^\pi(s, a)$, so that $q^\pi(s, a) = q^\pi(s)\pi(a \mid s)$.

**Additional notation.** We denote $\mathbb{E}_t[\cdot] = \mathbb{E}[\cdot \mid \mathcal{F}_{t-1}]$, where $\{\mathcal{F}_t\}_{t \geq 0}$ is the natural filtration generated by all observations up to the end of episode $t$. Let $\mathbb{I}_t(s, a) = \mathbb{1}[(s_{t,h}, a_{t,h}) = (s, a), \exists h \in \{0, \ldots, H-1\}]$ be the indicator function representing whether the state-action pair $(s, a)$ is visited under the policy $\pi_t$ used in episode $t$ and transition kernel $P$, and let $\mathbb{I}_t(s) = \sum_a \mathbb{I}_t(s, a)$. We also define the visitation counts $N_t(s, a) := \sum_{\tau=1}^t \mathbb{I}_\tau(s, a)$. We write $\ell_t(h) \in [0, 1]^{S_h \times A}$ for the restriction of $\ell_t$ to layer $h$ in episode $t$, and use the same notation for other functions defined on $\mathcal{S} \times \mathcal{A}$.

## 2.1. Regimes of Environments

We consider three regimes for how the loss functions $\ell_1, \ldots, \ell_T$ are generated. In the adversarial regime, we make no generative assumption. At the beginning of episode $t$, the environment arbitrarily selects a loss function $\ell_t \in [0, 1]^{S \times A}$. Specifically, $\ell_t$ may adapt to the past history but not to the learner's fresh randomness in the current episode. In the stochastic regime, the loss functions $\ell_1, \ldots, \ell_T$ are sampled i.i.d. from a fixed and unknown distribution $\mathcal{D}$.

The *stochastic regime with adversarial corruption* generalizes both the stochastic and adversarial regimes. Let $\ell'_1, \ldots, \ell'_T$ be sampled i.i.d. from a fixed and unknown distribution $\mathcal{D}$, and let the observed loss functions $\ell_1, \ldots, \ell_T$ be arbitrary corruptions of $\ell'_1, \ldots, \ell'_T$. We quantify the total corruption level $\mathcal{C} := \mathbb{E}\big[\sum_{t=1}^T \sum_{h=0}^{H-1} \|\ell'_t(h) - \ell_t(h)\|_\infty\big] \in [0, HT]$. In particular, when $\mathcal{C} = 0$, the stochastic regime with adversarial corruption reduces to the stochastic regime, whereas when $\mathcal{C} = \Omega(T)$ it coincides with the adversarial regime. For each state-action

pair $(s, a)$, let $\mu(s, a) := \mathbb{E}_{\ell' \sim \mathcal{D}}[\ell'(s, a)]$ and $\sigma^2(s, a) := \mathbb{E}_{\ell' \sim \mathcal{D}}[(\ell'(s, a) - \mu(s, a))^2]$ denote the mean and variance, respectively. Let $\pi^\star$ be an optimal policy for the uncorrupted mean loss function $\mu$, and define the suboptimality gap $\Delta \colon \mathcal{S} \times \mathcal{A} \to [0, H]$ as $\Delta(s, a) := Q^{\pi^\star}(s, a; \mu) - \min_{a' \in \mathcal{A}} Q^{\pi^\star}(s, a'; \mu)$.

## 2.2. Optimistic Follow-the-Regularized-Leader

Our proposed algorithms are based on the *optimistic follow-the-regularized-leader* (OFTRL) framework (Chiang et al., 2012; Rakhlin & Sridharan, 2013; Steinhardt & Liang, 2014), which has also been adopted in several existing studies (Wei & Luo, 2018; Ito et al., 2022a).

Here, we present OFTRL in the standard online linear optimization setting over a convex set[3] $\mathcal{K}$. At each round $t$, the learner outputs $p_t \in \mathcal{K}$ and incurs linear loss $\langle p_t, c_t \rangle$, where $\{c_t\}_{t=1}^T$ are loss vectors.[4] The OFTRL algorithm with differentiable regularizers $\{\psi_t\}_{t=1}^T$ and loss predictions $\{m_t\}_{t=1}^T$ chooses $p_t$ in $\mathcal{K}$ by

$$p_t = \arg\min_{p \in \mathcal{K}}\left\{\left\langle p, \sum_{\tau=1}^{t-1} c_\tau + m_t \right\rangle + \psi_t(p)\right\}. \quad (1)$$

The FTRL algorithm is recovered as the special case when $m_t = 0$ for all $t$ in (1). The sequence $\{m_t\}_{t=1}^T$ serves as an *optimistic prediction* of the upcoming loss vector. When the prediction is accurate, the algorithm improves regret guarantees, while in the worst case, the regret bound remains of the same order as FTRL.

We consider two schemes to obtain $\{m_t\}_{t=1}^T$ in (1), used for both global optimization and policy optimization. The first scheme is based on a gradient descent approach inspired by Ito (2021); Tsuchiya et al. (2023): we initialize $m_1(s, a) = 1/2$ for all $(s, a)$ and update

$$m_{t+1}(s, a) = $$
$$\begin{cases} (1 - \xi)\, m_t(s, a) + \xi\, \ell_t(s, a) & \text{if } \mathbb{I}_t(s, a) = 1, \\ m_t(s, a) & \text{if } \mathbb{I}_t(s, a) = 0, \end{cases} \quad (2)$$

where $\xi \in (0, 1/2)$ is a step size. We set $\xi = 1/4$ throughout this paper. This approach is useful for obtaining pathlength regret bounds depending on (6). The second scheme is based on the empirical mean predictor:

$$m_t(s, a) = \frac{\sum_{\tau=1}^{t-1} \mathbb{I}_\tau(s, a)\ell_\tau(s, a)}{\max\{1, N_{t-1}(s, a)\}}. \quad (3)$$

We will show that this is useful to obtain variance-aware gap-dependent regret bounds depending on (8).

---

[3]In our applications, $\mathcal{K} = \Omega(P)$ for global optimization and $\mathcal{K} = \Delta(\mathcal{A})$ for policy optimization.

[4]In our applications, $c_t$ serves as a loss estimator, namely $c_t = \widehat{\ell}_t$ for global optimization and $c_t = \widehat{Q}_t - B_t$ for policy optimization.

# 3. Complexity Measures in Online MDPs

This section introduces several complexity measures for online tabular MDPs. In our analysis, we derive guarantees that scale with these data-dependent complexity measures, and our algorithms do not need to know these quantities in advance.

## 3.1. Complexity in the Adversarial Regime

The first-order complexity $L^\star \in [0, HT]$ is defined as

$$L^\star := \min_{\pi \in \Pi} \mathbb{E}\left[\sum_{t=1}^{T} V^\pi(s_0; \ell_t)\right], \qquad (4)$$

which is the cumulative loss of the best fixed policy in hindsight, sometimes referred to as the small-loss quantity, and investigated in Lee et al. (2020); Dann et al. (2023a).

We further introduce new complexity measures for online MDPs. The following two can be seen as extensions of those used in multi-armed bandits. The second-order complexity $Q_\infty \in [0, HT/4]$ is defined as

$$Q_\infty := \min_{\ell^\star \in [0,1]^{S \times A}} \mathbb{E}\left[\sum_{t=1}^{T} \sum_{h=0}^{H-1} \|\ell_t(h) - \ell^\star(h)\|_\infty^2\right], \quad (5)$$

which becomes small when the losses stay close to a single baseline $\ell^\star$ over time. The path-length (or total variation) complexity $V_1 \in [0, SA(T-1)]$ is defined as

$$V_1 := \mathbb{E}\left[\sum_{t=1}^{T-1} \|\ell_{t+1} - \ell_t\|_1\right], \qquad (6)$$

which becomes small when the loss sequence changes slowly over episodes.

## 3.2. Complexity in the Stochastic Regime

We also introduce variance-based complexity measures for the stochastic regime. The *occupancy-weighted variance* $\mathbb{V} \in [0, H/4]$ is defined as

$$\mathbb{V} := \max_{\pi \in \Pi} \sum_{s,a} q^\pi(s,a)\sigma^2(s,a), \qquad (7)$$

which is the stochastic noise weighted by the occupancy measure. The *conditional occupancy-weighted variance* $\mathbb{V}^c(s) \in [0, H/4]$ at state $s$ is defined as

$$\mathbb{V}^c(s) := \max_{\pi \in \Pi, a \in \mathcal{A}} \sum_{s',a'} q^\pi(s', a' \mid s, a)\sigma^2(s', a'), \quad (8)$$

which is the remaining noise after reaching $s$ maximized over the first action $a$ at $s$.

There are known variance-dependent complexity measures in the literature. The maximum (unconditional) total variance (Zhou et al., 2023; Zhang et al., 2024) and maximum conditional total variance (Chen et al., 2025) are defined as

$$\mathsf{Var}_{\max} := \max_{\pi \in \Pi} \sum_{s,a} q^\pi(s,a)\mathsf{Var}^\star(s,a),$$

$$\mathsf{Var}_{\max}^c := \max_{\pi \in \Pi, s \in \mathcal{S}} \sum_{s',a'} \bar{q}^\pi(s', a' \mid s)\mathsf{Var}^\star(s', a'),$$

where $\mathsf{Var}^\star(s,a) := \sigma^2(s,a) + \mathsf{Var}_{s' \sim P(\cdot|s,a)}[V^{\pi^\star}(s')]$ (Zanette & Brunskill, 2019; Simchowitz & Jamieson, 2019) and $\bar{q}^\pi(s', a' \mid s)$ denotes the occupancy of $(s', a')$ over the entire trajectory conditioned on visiting state $s$. These complexity measures were introduced in the context of a value-based approach for the stochastic regime with unknown transitions. In our setting, $\mathbb{V}$ and $\mathbb{V}^c$ are analogous to $\mathsf{Var}_{\max}$ and $\mathsf{Var}_{\max}^c$, respectively. Since we consider known transitions, the second term in $\mathsf{Var}^\star(s,a)$ is unnecessary and can be omitted. Moreover, $\mathbb{V}^c$ is defined using conditional occupancy measures $q^\pi(s', a' \mid s, a)$ and captures variance only after visiting $(s,a)$, whereas $\mathsf{Var}_{\max}^c$ aggregates variance over the entire trajectory by conditioning on visiting state $s$. As a consequence, our variance measures are $H^2$-sharper than those based on $\mathsf{Var}^\star(s,a)$. Further discussion is deferred to Appendix B.2.

# 4. Global Optimization

This section presents an occupancy-measure-based algorithm designed to achieve the data-dependent and variance-adaptive regret guarantees stated in Theorems 4.1 and 4.2.

## 4.1. Algorithm

In global optimization, we optimize directly over occupancy measures. Let $\Omega(P)$ denote the convex set of valid occupancy measures induced by the transition kernel $P$. In each episode $t$, we run OFTRL over $\Omega(P)$ with log-barrier regularizers and loss predictions. Our design is inspired by Jin et al. (2021) but adapted to the OFTRL framework, and thus the loss estimator and the corresponding loss-shifting function differ from their FTRL-based construction. The complete algorithm is described in Algorithm 1.

In particular, we run OFTRL over $\Omega(P)$ with $c_t = \widehat{\ell}_t$ in (1), leading to the occupancy-measure update in (9) (Line 3) with the log-barrier regularizer. Here, we use the optimistic importance-weighted estimator in (10) (Line 5), where $\{m_t\}_{t=1}^{T}$ is chosen as in (2) or (3). This estimator is unbiased in the sense that $\mathbb{E}_t[\widehat{\ell}_t(s,a)] = \ell_t(s,a)$.

To obtain a polylogarithmic regret in the stochastic regime, we use the loss-shifting technique of Jin et al. (2021). Since the stability of OFTRL is controlled by the shifted loss

**Algorithm 1** Global Optimization with Data- and Variance-dependent Bounds

1 **Input:** MDP $\mathcal{M} = (\mathcal{S}, \mathcal{A}, P, H, s_0)$, initial learning rate $\frac{1}{\eta_1(s,a)} = \frac{1}{\eta_1} = 2H$, loss prediction $m_t$ chosen as in (2) or (3).

2 **for** $t = 1, 2, \dots$ **do**

3     Compute the occupancy measure $q^{\pi_t} \in \Omega(P)$ by

$$q^{\pi_t} = \underset{q \in \Omega(P)}{\arg\min}\left\{\left\langle q, \sum_{\tau=1}^{t-1} \widehat{\ell}_\tau + m_t \right\rangle + \psi_t(q)\right\}, \quad \psi_t(q) = \sum_{s,a} \frac{1}{\eta_t(s,a)} \log\left(\frac{1}{q(s,a)}\right). \tag{9}$$

4     Compute policy $\pi_t$ from $q^{\pi_t}$ by $\pi_t(a \mid s) \propto q^{\pi_t}(s,a)$, and obtain a trajectory $\{(s_{t,h}, a_{t,h}, \ell_t(s_{t,h}, a_{t,h}))\}_{h=0}^{H-1}$.

5     Compute the loss estimator $\widehat{\ell}_t(s,a)$ by

$$\widehat{\ell}_t(s,a) = m_t(s,a) + \frac{\mathbb{I}_t(s,a)(\ell_t(s,a) - m_t(s,a))}{q^{\pi_t}(s,a)}. \tag{10}$$

6     Update the learning rate $\eta_{t+1}(s,a)$ by

$$\frac{1}{\eta_{t+1}(s,a)} = \frac{1}{\eta_t(s,a)} + \frac{\eta_t(s,a)\zeta_t(s,a)}{\log(T)}, \tag{11}$$

7     where $\zeta_t(s,a) = q^{\pi_t}(s,a)^2 \min\{(\widehat{\ell}_t(s,a) - m_t(s,a))^2, (\widehat{\ell}_t(s,a) + g_t(s,a) - m_t(s,a))^2\}$ with $g_t$ defined in (12).

8     Update the loss prediction $m_{t+1}(s,a)$ by (2) or (3).

---

$\widetilde{\ell}_t := \widehat{\ell}_t - m_t$, we construct the following loss-shifting function

$$g_t(s,a) = Q^{\pi_t}(s,a;\widetilde{\ell}_t) - V^{\pi_t}(s;\widetilde{\ell}_t) - \widetilde{\ell}_t(s,a). \tag{12}$$

With this shifting function, the learner equivalently runs OFTRL with the advantage function $Q^{\pi_t}(s,a;\widetilde{\ell}_t) - V^{\pi_t}(s;\widetilde{\ell}_t)$, which enables a self-bounding regret analysis in the stochastic regime. Moreover, when $m_t$ is the empirical-mean predictor in (3), the same shifting construction allows us to control the resulting variance term by $\mathbb{V}^c$.

### 4.2. Regret Upper Bounds

With the optimistic estimator, shifted losses, and adaptive log-barrier learning rates, we state the following theorem. We defer all technical lemmas and proofs to Appendix D.

**Theorem 4.1.** *Algorithm 1 with $m_t$ in (2) guarantees*

$$\mathrm{Reg}_T \lesssim \sqrt{SA\log(T)\min\{L^\star, HT - L^\star, Q_\infty, V_1\}} + HSA\log T.$$

*Under the stochastic regime with adversarial corruption, it simultaneously ensures*

$$\mathrm{Reg}_T \lesssim \sqrt{SA\log(T)(\mathbb{V}T + \mathcal{C})} + HSA\log T,$$
$$\mathrm{Reg}_T \lesssim U + \sqrt{U\mathcal{C}} + HSA\log T,$$

*where $U = \sum_s \sum_{a \neq \pi^\star(s)} \frac{H^2\log(T)}{\Delta(s,a)}$.*

Our first-order, second-order, and variance-aware gap-independent bounds are minimax optimal up to logarithmic factors (see lower bounds in Theorems 6.1 and 6.2) and also

recover the worst-case dependence $\widetilde{O}(\sqrt{HSAT})$ in the adversarial regime (Zimin & Neu, 2013). Furthermore, our gap-dependent guarantee improves over Jin et al. (2021) by avoiding their additional dependence on $1/\min_{s,a} \Delta(s,a)$.

**Theorem 4.2.** *Algorithm 1 with $m_t$ in (3) guarantees*

$$\mathrm{Reg}_T \lesssim \sqrt{SA\log(T)\min\{L^\star, HT - L^\star, Q_\infty\}} + HSA\log(T).$$

*Under the stochastic regime with adversarial corruption, it simultaneously ensures*

$$\mathrm{Reg}_T \lesssim \sqrt{SA\log(T)(\mathbb{V}T + \mathcal{C})} + HSA\log(T),$$
$$\mathrm{Reg}_T \lesssim U_{\mathsf{Var}} + \sqrt{U_{\mathsf{Var}}\mathcal{C}} + \sqrt{HS^2A^2\mathcal{C}}\log(T)$$
$$+ H^{\frac{1}{2}}S^{\frac{3}{2}}A^{\frac{3}{2}}\log^{\frac{3}{2}}(T),$$

*where $U_{\mathsf{Var}} = \sum_s \sum_{a \neq \pi^\star(s)} \frac{H\mathbb{V}^c(s)\log(T)}{\Delta(s,a)}$.*

*Remark* 4.3. If the uncorrupted losses are generated independently and are uncorrelated across layers, the variance-aware gap-dependent bound in Theorem 4.2 improves by a factor of $H$ to $U_{\mathsf{Var}} = \sum_s \sum_{a \neq \pi^\star(s)} \frac{\mathbb{V}^c(s)\log(T)}{\Delta(s,a)}$.

## 5. Policy Optimization

This section presents a policy-optimization algorithm with log-barrier regularization that attains data- and variance-dependent regret bounds. Policy optimization can be viewed as solving a multi-armed bandit problem at each state, with $\pi_t(\cdot \mid s)$ as the action distribution. This is formalized by the performance-difference lemma (Kakade & Langford, 2002), which implies $\mathrm{Reg}_T = \mathbb{E}\left[\sum_s \sum_t q^{\mathring{\pi}}(s)\langle \pi_t(\cdot \mid s) - \mathring{\pi}(\cdot \mid s), Q^{\pi_t}(s, \cdot; \ell_t)\rangle\right]$ and motivates using the $Q$-function as the loss in OFTRL.

**Algorithm 2** Policy Optimization with Data- and Variance-dependent Bounds

1 **Input:** MDP $\mathcal{M} = (\mathcal{S}, \mathcal{A}, P, H, s_0)$, regularizer $\psi_t(\pi(\cdot \mid s)) = \sum_a \frac{1}{\eta_t(s,a)} \log(1/\pi(a \mid s))$, exploration rate $\gamma_t = \frac{\sqrt{HS}}{t}$, initial learning rate $1/\eta_1(s,a) = 1/\eta_1 = 180H^3$, loss prediction $m_t$ chosen as in (2) or (3).

2 **for** $t = 1, 2, \ldots$ **do**

3 $\quad$ Compute the policy $\pi_t(\cdot \mid s)$ at each state $s \in \mathcal{S}$ by

$$\pi_t(\cdot \mid s) = \underset{\pi(\cdot \mid s) \in \Delta(\mathcal{A})}{\arg\min} \left\{ \left\langle \pi(\cdot \mid s), \sum_{\tau=1}^{t-1} \Big( \widehat{Q}_\tau(s, \cdot) - B_\tau(s, \cdot) \Big) + Q^{\pi_t}(s, \cdot; m_t) \right\rangle + \psi_t(\pi(\cdot \mid s)) \right\}. \quad (13)$$

4 $\quad$ Compute $Y_t \leftarrow \mathbb{1}\left[ \max_{s,a} \frac{\eta_t(s,a)}{q_t(s)} \leq \frac{1}{18\sqrt{H^3 S}} \right]$, where $q_t(s) = q^{\pi_t}(s) + \gamma_t$. If $Y_t = 0$, we insert a virtual episode and shift the indices of subsequent real episodes.

5 $\quad$ If $Y_t = 1$ (real episode), obtain a trajectory $\{(s_{t,h}, a_{t,h}, \ell_t(s_{t,h}, a_{t,h}))\}_{h=0}^{H-1}$.

6 $\quad$ Let $q_t(s) = q^{\pi_t}(s) + \gamma_t$, $L_{t,h} = \sum_{h'=h}^{H-1} \ell_t(s_{t,h'}, a_{t,h'})$, $M_{t,h} = \sum_{h'=h}^{H-1} m_t(s_{t,h'}, a_{t,h'})$, and

$$\widehat{Q}_t(s,a) = Q^{\pi_t}(s,a; m_t) + \frac{\mathbb{I}_t(s,a)(L_{t,h(s)} - M_{t,h(s)})}{q_t(s)\pi_t(a \mid s)} Y_t - \frac{\gamma_t H}{q_t(s)}. \quad (14)$$

7 $\quad$ Let $(s_t^\dagger, a_t^\dagger) \in \arg\max_{s,a} \frac{\eta_t(s,a)}{q_t(s)}$ (break ties arbitrarily), and update the learning rates $\eta_{t+1}(s,a)$,

$$\frac{1}{\eta_{t+1}(s,a)} = \begin{cases} \frac{1}{\eta_t(s,a)} + \frac{\eta_t(s,a)\zeta_t(s,a)}{q_t(s)^2 \log(T)} & \text{if } t \text{ is a real episode,} \\ \frac{1}{\eta_t(s,a)} \left( 1 + \frac{\mathbb{1}\{(s_t^\dagger, a_t^\dagger) = (s,a)\}}{324 H \log(T)} \right) & \text{if } t \text{ is a virtual episode,} \end{cases} \quad (15)$$

$$\text{where} \quad \zeta_t(s,a) = (\mathbb{I}_t(s,a) - \pi_t(a \mid s)\mathbb{I}_t(s))^2 (L_{t,h(s)} - M_{t,h(s)})^2. \quad (16)$$

8 $\quad$ Compute $b_t(s)$ by (20), and then compute $B_t(s,a)$ by (18).

9 $\quad$ Compute loss prediction $m_{t+1}(s,a)$ by (2) or (3).

## 5.1. Algorithm

Here, we present the policy-optimization procedure in Algorithm 2. For each state $s$, we run OFTRL as in (13) with the log-barrier regularizer

$$\psi_t(\pi(\cdot \mid s)) = \sum_a \frac{1}{\eta_t(s,a)} \log \left( \frac{1}{\pi(a \mid s)} \right), \quad (17)$$

where $\eta_t(s,a) > 0$ are time-varying, data-dependent learning rates updated via (15).

In Line 3, given the policy $\pi_t$ and the loss prediction $m_t$, we can compute $Q^{\pi_t}(s,a; m_t)$ by backward dynamic programming and then determine $\pi_t(\cdot \mid s)$ for each state, starting from the last layer and proceeding backward over $h = H-1, \ldots, 0$. Following Dann et al. (2023a), we choose the exploration rate $\gamma_t = \sqrt{HS}/t$, in order to achieve a polylogarithmic regret in the stochastic regime. The loss prediction $m_t$ is updated by the gradient descent in (2) or empirical mean in (3), which allows us to obtain data-dependent regret bounds in the adversarial regime and variance-dependent regret bounds in the stochastic regime. In what follows, we describe three key technical components of the algorithm: the dilated bonus, virtual episodes, and a novel optimistic $Q$-function estimator.

**Dilated bonus.** In policy optimization, updates are performed locally at each state, which can lead to insufficient exploration. To enforce global exploration, Luo et al. (2021) introduced a dilated exploration bonus $B_t(s,a)$ that is constructed in the same form as a $Q$-function,

$$B_t(s,a) = b_t(s)$$
$$+ \left( 1 + \frac{1}{H} \right) \mathbb{E}_{s' \sim P(\cdot \mid s,a), a' \sim \pi_t(\cdot \mid s')}[B_t(s',a')]. \quad (18)$$

Intuitively, $b_t(s)$ is chosen to scale inversely with the visitation probability $q^{\pi_t}(s)$, so that rarely visited states receive larger exploration incentives (see, e.g., (20) or Luo et al. 2021, Eq. (8)). The resulting bonus $B_t(s,a)$ has the same recursive structure as a $Q$-function and is subtracted from the $Q$-estimate in the policy update as in (13). This construction of bonus $B_t(s,a)$ yields the following lemma, which plays a key role in achieving the best-of-both-worlds guarantees in Dann et al. (2023a) and this work.

**Lemma 5.1** (Luo et al. 2021, Lemma B.2). *Suppose that $b_t(s)$ is a nonnegative loss function, $B_t$ satisfies (18) for all $(s,a)$, and that, for each $s \in \mathcal{S}$ and for some $J(s) \geq 0$,*

$$\mathbb{E}\left[ \sum_{t,a} (\pi_t(a \mid s) - \mathring{\pi}(a \mid s))(Q^{\pi_t}(s,a; \ell_t) - B_t(s,a)) \right.$$

$$\leq J(s) + \mathbb{E}\left[\sum_t b_t(s) + \frac{1}{H}\sum_{t,a} \pi_t(a \mid s)B_t(s,a)\right]. \quad (19)$$

*Then,* $\text{Reg}_T \leq \sum_s q^{\mathring{\pi}}(s)J(s) + 3\mathbb{E}\left[\sum_{t=1}^T V^{\pi_t}(s_0; b_t)\right].$

The factor $(1 + 1/H)$ in (18) slightly inflates the propagated bonus, so that the error due to the bonus term can be absorbed into $\frac{1}{H}\sum_{t,a} \pi_t(a \mid s)B_t(s,a)$ in (19). Consequently, the overall exploration overhead is bounded by a constant factor of the learner's own occupancy term $\sum_{t=1}^T V^{\pi_t}(s_0; b_t)$, as formalized in Lemma 5.1.

To make (19) hold, we use the local bonus $b_t(s)$ given by

$$b_t(s) = 6\sum_a \left(\frac{1}{\eta_{t+1}(s,a)} - \frac{1}{\eta_t(s,a)}\right)\log(T) + \frac{5\gamma_t H}{q_t(s)}. \quad (20)$$

The first term is the OFTRL-regret overhead induced by the use of an adaptive learning rate, and the second term arises from the optimism in the $Q$-estimation (explained later). For further details and intuition behind the bonus term, we refer the reader to Luo et al. (2021); Dann et al. (2023a).

**Virtual episodes.** To motivate the introduction of virtual episodes (Line 4), we first discuss the learning rate design in the OFTRL algorithm with a log-barrier regularizer in (17). For a fixed state $s$, the regret of this algorithm can be roughly bounded by

$$\underbrace{\sum_{t,a}\left(\frac{1}{\eta_{t+1}(s,a)} - \frac{1}{\eta_t(s,a)}\right)\log(T)}_{\text{penalty-term}} + \underbrace{\sum_{t,a}\frac{\eta_t(s,a)\zeta_t(s,a)}{q_t(s)^2}}_{\text{stability-term}},$$

where $\zeta_t(s,a)$ is the data-dependent term defined in (16). Hence, it is natural to choose a data-dependent learning rate like Dann et al. (2023a) that directly balances these two terms, namely $\frac{1}{\eta_{t+1}(s,a)} = \frac{1}{\eta_t(s,a)} + \frac{\eta_t(s,a)\zeta_t(s,a)}{q_t(s)^2 \log(T)}$. With this update, the penalty and stability terms evolve on the same scale. However, to upper bound the error term induced by the bonus by $\frac{1}{H}\sum_{t,a} \pi_t(a \mid s)B_t(s,a)$, the analysis additionally requires $\eta_t(s,a)\pi_t(a \mid s)B_t(s,a) \lesssim \frac{1}{H}$. Since $B_t(s,a)$ is of order $1/q_t(s)^2$ from (20), it becomes large when $q_t(s)$ is small, and the above inequality is not guaranteed by the learning rate schedule alone.

Therefore, following Dann et al. (2023a), we enforce the above condition by inserting virtual episodes (Line 4). At the start of episode $t$, if $\max_{s,a} \eta_t(s,a)/q_t(s)$ is larger than $1/(18\sqrt{H^3 S})$, we set $Y_t = 0$ and declare the episode virtual. In a virtual episode, we set $\ell_t(s,a) = 0$ for all $(s,a)$. We then shrink the learning rate at the state-action pair $(s_t^\dagger, a_t^\dagger) \in \arg\max_{s,a} \frac{\eta_t(s,a)}{q_t(s)}$ by a constant factor $1 + \frac{1}{324H\log(T)}$, and shift the indices of real episodes. The total number of virtual episodes is at most

$O(HSA\log^2(T))$, so we still use $T$ for the total number of episodes and absorb their effect into lower-order terms, while the data-dependent complexity measures are defined over real episodes only.

**New $Q$-function estimator.** A key technical ingredient in our analysis is the construction of our $Q$-function estimator $\widehat{Q}_t$ defined in (14), which is used for OFTRL in (13) (Line 6). Since OFTRL updates the policy using the $Q$-function as a loss, we propagate the loss prediction $m_t$ in (2) or (3) through the $Q$-recursion and obtain the predicted $Q$-function $Q^{\pi_t}(s,a; m_t)$.

The main difficulty is that simply incorporating this predicted term does not provide enough control over the bias. Indeed, if we were to use $Q^{\pi_t}(s,a; m_t)$ alone (*i.e.*, the first two terms in (14)) as the estimator, then the expected deviation $\mathbb{E}_t[\widehat{Q}_t(s,a)] - Q^{\pi_t}(s,a; \ell_t)$ could be positive or negative, making the bias difficult to control directly. To resolve this issue, we subtract a margin of the form $\gamma_t H/q_t(s)$ to ensure that $\widehat{Q}_t(s,a)$ is an optimistic estimator of $Q^{\pi_t}(s,a; \ell_t)$. Indeed, a direct calculation shows that (see Lemma E.2 for details)

$$\mathbb{E}_t\left[\widehat{Q}_t(s,a)\right] = Q^{\pi_t}(s,a; m_t)$$
$$+ \frac{q^{\pi_t}(s)}{q_t(s)}Q^{\pi_t}(s,a; \ell_t - m_t)Y_t - \frac{\gamma_t H}{q_t(s)}.$$

In particular, in a real episode ($Y_t = 1$), we have $0 \leq \mathbb{E}_t\left[Q^{\pi_t}(s,a; \ell_t) - \widehat{Q}_t(s,a)\right] \leq 2\gamma_t H/q_t(s)$. Hence, when $\gamma_t = 0$, $\widehat{Q}_t(s,a)$ is an unbiased estimator of $Q^{\pi_t}(s,a; \ell_t)$. For $\gamma_t > 0$, the estimator is optimistic in expectation, and this controlled optimism is useful in the regret analysis, as it makes the bias term easy to handle while still benefiting from the variance reduction due to the predictor $Q^{\pi_t}(s,a; m_t)$.

By contrast, in virtual episodes ($Y_t = 0$), the loss is zero but the prediction term $Q^{\pi_t}(s,a; m_t)$ still remains in the estimator. Thus, the estimator is not optimistic in the same sense as in real episodes and may introduce additional bias. The effect of this bias is limited, however, because the total number of virtual episodes is small, and it contributes only a lower-order term to the regret bound.

## 5.2. Regret Upper Bounds

We now state the resulting regret guarantee, with all proofs deferred to Appendix E.

**Theorem 5.2.** *Algorithm 2 with $m_t$ in (2) guarantees*

$$\text{Reg}_T \lesssim \sqrt{H^2 SA\log^2(T)\min\{L^\star, HT - L^\star, Q_\infty, V_1\}}$$
$$+ H^{\frac{5}{2}}S^{\frac{3}{2}}A\log^2(T).$$

*Under the stochastic regime with adversarial corruption, it simultaneously ensures*

$$\text{Reg}_T \lesssim \sqrt{H^2 S A \log^2(T)(\mathbb{V} T + \mathcal{C})} + H^{\frac{5}{2}} S^{\frac{3}{2}} A \log^2(T),$$

$$\text{Reg}_T \lesssim U + \sqrt{U \mathcal{C}} + H^{\frac{5}{2}} S^{\frac{3}{2}} A \log^2(T),$$

*where $U = \sum_s \sum_{a \neq \pi^\star(s)} \frac{H^2 \log^2(T)}{\Delta(s,a)}$.*

In the worst case, our bound becomes the known regret bounds based on policy optimization in Luo et al. (2021); Dann et al. (2023a), and the lower-order term $H^3 S^2 A^2 \log^2(T)$ in Dann et al. (2023a, Theorem 4.3) is improved to $H^{\frac{5}{2}} S^{\frac{3}{2}} A \log^2(T)$.

**Theorem 5.3.** *Algorithm 2 with $m_t$ in (3) guarantees*

$$\text{Reg}_T \lesssim \sqrt{H^2 S A \log^2(T) \min\{L^\star, HT - L^\star, Q_\infty\}} + H^{\frac{5}{2}} S^{\frac{3}{2}} A \log^2(T).$$

*Under the stochastic regime with adversarial corruption, it simultaneously ensures*

$$\text{Reg}_T \lesssim \sqrt{H^2 S A \log^2(T)(\mathbb{V} T + \mathcal{C})} + H^{\frac{5}{2}} S^{\frac{3}{2}} A \log^2(T),$$

$$\text{Reg}_T \lesssim U_{\text{Var}} + \sqrt{U_{\text{Var}} \mathcal{C}} + \sqrt{H S^2 A^2 \mathcal{C}} \log^{\frac{3}{2}}(T)$$
$$+ H^{\frac{1}{2}} S^{\frac{3}{2}} A (H^2 + \sqrt{A}) \log^2(T),$$

*where $U_{\text{Var}} = \sum_s \sum_{a \neq \pi^\star(s)} \frac{H \mathbb{V}^c(s) \log^2(T)}{\Delta(s,a)}$.*

*Remark* 5.4. If the uncorrupted losses are generated independently and are uncorrelated across layers, the variance-aware gap-dependent bound in Theorem 5.3 improves by a factor of $H$ to $U_{\text{Var}} = \sum_s \sum_{a \neq \pi^\star(s)} \frac{\mathbb{V}^c(s) \log^2(T)}{\Delta(s,a)}$.

## 6. Regret Lower Bounds

We complement our regret upper bounds with information-theoretic regret lower bounds for MDPs with bandit feedback. In multi-armed bandits, refined lower bounds such as first-order, second-order, and path-length bounds were developed by Gerchinovitz & Lattimore (2016) and Bubeck et al. (2019). For MDPs, the data-independent minimax lower bound $\Omega(\sqrt{HSAT})$ is already known (Zimin & Neu, 2013; Tsuchiya et al., 2025b). Accordingly, our focus is on data-dependent lower bounds in MDPs, identifying the optimal dependence on measures such as $L^\star$, $Q_\infty$, $V_1$, and $\mathbb{V}$. All proofs are deferred to Appendix G.

The refined adversarial lower bounds below are obtained via a simple truncation reduction: we run an instance that induces $\Omega(\sqrt{HSAT})$ regret for only a prefix of episodes and set all losses to zero thereafter. This ensures that the corresponding complexity measure is small, while preserving a nontrivial regret contribution from the active phase.

**Theorem 6.1.** *Suppose that $H \geq 3$, $A \geq 3$, $T \geq \frac{SA}{8H}$ and $\alpha \in \left[ \frac{\lceil SA/8H \rceil}{T}, 1 \right]$. Then, for any policy $\{\pi_t\}_{t=1}^T$, there exists an episodic MDP with adversarial losses satisfying $L^\star \leq \alpha H T$, $Q_\infty \leq \alpha H T$, $V_1 \leq \alpha S A T$ such that $\text{Reg}_T = \Omega(\sqrt{\alpha H S A T})$.*

Consequently, choosing $\alpha$ appropriately for each complexity measure in Theorem 6.1 yields $\text{Reg}_T = \Omega(\sqrt{SAL^\star})$, $\Omega(\sqrt{SAQ_\infty})$, and $\Omega(\sqrt{HV_1})$.

These lower bounds imply that the regret bounds in Section 4 are optimal up to logarithmic factors, except for the path-length bound. For the path-length bounds, our upper bound leaves an $\sqrt{SA/H}$-dependent gap. This gap is consistent with that in multi-armed bandits: the best-known upper bounds come with an explicit $\sqrt{A}$ dependence on the number of actions $A$, whereas the lower bounds scale as $\Omega(\sqrt{V_1})$ and do not require any dependence on $A$ (Bubeck et al., 2019). By contrast, policy optimization often introduces an additional dependence on $H$. In particular, as in Luo et al. (2021); Dann et al. (2023a), the resulting data-independent guarantees can be worse by a factor of $H$ compared to the best-known bounds. Closing this $H$-gap in minimax regret remains an important open problem.

Finally, we turn to the stochastic regime and consider the occupancy-weighted variance $\mathbb{V}$.

**Theorem 6.2.** *Suppose that $H \geq 3$, $A \geq 3$, $T \geq \frac{SA}{8H}$, and $\alpha \in (0, 1/4]$. Then, for any policy $\{\pi_t\}_{t=1}^T$, there exists an episodic MDP with stochastic losses satisfying $\mathbb{V} \leq \alpha H$ such that $\text{Reg}_T = \Omega(\sqrt{\alpha H S A T}) = \Omega(\sqrt{SA\mathbb{V}T})$.*

The above lower bound implies that the regret bound of $\widetilde{O}(\sqrt{SA\mathbb{V}T})$ in Section 4 is optimal up to logarithmic factors. In contrast, policy optimization typically incurs a multiplicative factor of $H$ here as well.

## 7. Conclusion

In this work, we introduced refined complexity measures for tabular MDPs, including the second-order measure $Q_\infty$, the path-length measure $V_1$, and the variance measures $\mathbb{V}$ and $\mathbb{V}^c$. For the known-transition setting, we developed both global optimization and policy optimization algorithms that achieve best-of-both-worlds guarantees with these refined data-dependent bounds. Our lower bounds further show that the guarantees obtained by the global optimization approach are nearly optimal. An important direction for future work is to remove the known-transition assumption and extend our results to unknown transitions, where transition-estimation errors would also need to be controlled in a data-dependent way.

## Acknowledgements

TT was supported by JSPS KAKENHI Grant Number JP24K23852 and partially supported by JSPS KAKENHI Grant Number JP26K21297. KY was partially supported by JSPS KAKENHI Grant Number JP24H00703.

## Impact Statement

This paper presents work whose goal is to advance the field of machine learning. There are many potential societal consequences of our work, none of which we feel must be specifically highlighted here.

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

# Contents

# A. Summary of Notation

*Table 4.* Summary of notation.

| Symbol | Meaning |
|---|---|
| **Online tabular MDPs** | |
| $\mathcal{M} = (\mathcal{S}, \mathcal{A}, P, H, s_0)$ | Episodic finite-horizon MDP with known transition |
| $\mathcal{S}, S$ | State space and its size $S = |\mathcal{S}|$ |
| $\mathcal{A}, A$ | Action space and its size $A = |\mathcal{A}|$ |
| $P$ | Transition kernel |
| $H$ | Horizon length |
| $T$ | Number of episodes |
| $h(s)$ | Layer index of state $s$ |
| $s_{t,h}, a_{t,h}$ | State / action at step $h$ in episode $t$ |
| $\ell_t(s, a)$ | Loss assigned to $(s, a)$ in episode $t$ |
| $\pi_t$ | Policy in episode $t$ |
| $\text{Reg}_T$ | Regret over $T$ episodes |
| $\mathbb{I}_t(s, a)$ | Visitation indicator of $(s, a)$ in episode $t$ |
| $N_t(s, a)$ | Number of visits to $(s, a)$ up to episode $t$ |
| $V^\pi(s; \ell)$ | Value function under policy $\pi$ from state $s$ with loss $\ell$ |
| $Q^\pi(s, a; \ell)$ | $Q$-function under policy $\pi$ from $(s, a)$ with loss $\ell$ |
| $\mathring{\pi}, q^{\mathring{\pi}}$ | Optimal policy and its occupancy measure |
| $q^\pi(s), q^\pi(s, a)$ | Occupancy measure under policy $\pi$ |
| $q^\pi(s', a' \mid s, a)$ | Conditional occupancy from $(s, a)$ to $(s', a')$ under $\pi$ |
| $\ell'_t(s, a)$ | Uncorrupted i.i.d. loss |
| $\mathcal{C} = \mathbb{E}[\sum_{t=1}^T \sum_{h=0}^{H-1} \|\ell'_t(h) - \ell_t(h)\|_\infty]$ | Corruption budget |
| $\mu(s, a) = \mathbb{E}_{\ell' \sim \mathcal{D}}[\ell'(s, a)]$ | Mean of $\ell'_t(s, a)$ |
| $\sigma^2(s, a) = \mathbb{E}_{\ell' \sim \mathcal{D}}[(\ell'(s, a) - \mu(s, a))^2]$ | Variance of $\ell'_t(s, a)$ |
| $\pi^\star$ | Optimal deterministic policy under $\mu$ |
| $\Delta(s, a) = Q^{\pi^\star}(s, a; \mu) - \min_{a' \in \mathcal{A}} Q^{\pi^\star}(s, a'; \mu)$ | Suboptimality gap at $(s, a)$ |
| **Data-dependent complexity measures** | |
| $L^\star = \mathbb{E}[\sum_{t=1}^T V^{\mathring{\pi}}(s_0; \ell_t)]$ | First-order complexity in (4) |
| $Q_\infty = \min_{\ell^\star \in [0,1]^{S \times A}} \mathbb{E}[\sum_{t=1}^T \sum_{h=0}^{H-1} \|\ell_t(h) - \ell^\star(h)\|_\infty^2]$ | Second-order complexity in (5) |
| $V_1 = \mathbb{E}[\sum_{t=1}^{T-1} \|\ell_{t+1} - \ell_t\|_1]$ | Path-length complexity (6) |
| $\mathbb{V} = \max_\pi \mathbb{E}[\sum_{s,a} q^\pi(s, a)\sigma^2(s, a)]$ | Occupancy-weighted variance in (7) |
| $\mathbb{V}^c(s) = \max_{a,\pi} \mathbb{E}[\sum_{s',a'} q^\pi(s', a' \mid s, a)\sigma^2(s', a')]$ | Conditional occupancy-weighted variance at state $s$ in (8) |
| **Common notation for Algorithms 1 and 2** | |
| $\eta_t(s, a)$ | Learning rate for $(s, a)$ in episode $t$ |
| $m_t(s, a)$ | Loss prediction for $(s, a)$ |
| $\zeta_t(s, a)$ | Data-dependent term for updating $\eta_t(s, a)$ |
| **Notation only for Algorithm 1** (global optimization) | |
| $\widehat{\ell}_t(s, a)$ | Loss estimator |
| **Notation only for Algorithm 2** (policy optimization) | |
| $b_t(s)$ | Bonus term at state $s$ in episode $t$ |
| $B_t(s, a)$ | Dilated bonus-to-go at $(s, a)$ in episode $t$ |
| $Y_t \in \{0, 1\}$ | Episode indicator ($Y_t = 1$ real, $Y_t = 0$ virtual) |
| $\mathcal{T}_r, \mathcal{T}_v$ | Sets of real and virtual episodes |
| $\widehat{Q}_t(s, a)$ | $Q$-function estimator |
| $\gamma_t, q_t(s) = q^{\pi_t}(s) + \gamma_t$ | Exploration rate and smoothed state occupancy |
| $L_{t,h(s)}, M_{t,h(s)}$ | Realized / predicted suffix loss from layer $h(s)$ |

For the reader's convenience, Table 4 collects the main notation used throughout the paper.

We formalize the conditional occupancy measure $q_{\pi_t}(s', a' \mid s, a)$ as follows:

$$q^{\pi_t}(s', a' \mid s, a) = \begin{cases} 0 & \text{if } h(s') < h(s), \\ 0 & \text{if } h(s') = h(s) \text{ and } (s', a') \neq (s, a), \\ 1 & \text{if } (s', a') = (s, a), \\ \Pr\big[(s_{t,h(s')}, a_{t,h(s')}) = (s', a') \mid (s_{t,h(s)}, a_{t,h(s)}) = (s, a)\big] & \text{if } h(s') > h(s). \end{cases}$$

# B. Further Discussion of Related Work

## B.1. Additional Related Work

**Online MDPs.** Adversarial MDPs were first studied by Even-Dar et al. (2009); Yu et al. (2008) and later extended to the episodic setting by Zimin & Neu (2013). Episodic MDPs with *bandit feedback*, where the learner observes losses only for the visited state-action pairs, have been extensively studied. In this setting, a line of work has established minimax-optimal regret via *global optimization*, which solves an optimization problem over the set of all occupancy measures. In particular, when the transition dynamics are known, global optimization achieves the minimax regret $\widetilde{\mathcal{O}}(\sqrt{HSAT})$ (Zimin & Neu, 2013), while under unknown transitions, global optimization achieves the regret $\widetilde{\mathcal{O}}(\sqrt{H^2S^2AT})$ (Rosenberg & Mansour, 2019; Jin et al., 2020). While global optimization enjoys optimal regret guarantees, it requires solving a large-scale convex optimization problem over the feasible occupancy-measure polytope at each episode, which can be computationally demanding and limits scalability in practice.

This has motivated a complementary line of *policy optimization*, which typically reduces the problem to separate instances of the multi-armed bandit problem for each state. Policy optimization was first shown to achieve a $\widetilde{O}(T^{2/3})$ regret upper bound under bandit feedback by Shani et al. (2020). Later Luo et al. (2021) attained the optimal $\widetilde{O}(\sqrt{T})$ dependence on the number of episodes by combining a dilated exploration bonus with a refined $Q$-function estimator, achieving $\widetilde{\mathcal{O}}(\sqrt{H^3SAT})$ regret under known transitions and $\widetilde{\mathcal{O}}(\sqrt{H^4S^2AT})$ under unknown transitions. Compared with global optimization, these guarantees incur an additional factor of $H$ in the leading term, and closing this dependence gap remains open. Subsequent work has extended policy optimization to more challenging feedback models, including delayed and aggregate feedback (Lancewicki et al., 2022; Jin et al., 2022; Lancewicki et al., 2023; Lancewicki & Mansour, 2025).

In parallel, in the stochastic setting, both *model-based* algorithms, which learn the transition dynamics and construct confidence sets over the transition and loss functions (Jaksch et al., 2010; Azar et al., 2017), and *value-based* methods, which add exploration bonuses directly to the $Q$-function (Jin et al., 2018; Zanette & Brunskill, 2019), have been developed and achieve near-optimal regret guarantees.

**Best-of-both-worlds algorithms.** The *best-of-both-worlds* guarantee, which aims to achieve near-optimal regret in both adversarial and stochastic regimes with a single algorithm, was first investigated in the multi-armed bandit setting by Bubeck & Slivkins (2012). Subsequent research has refined the analysis through a variety of techniques (Seldin & Slivkins, 2014; Auer & Chiang, 2016; Seldin & Lugosi, 2017; Wei & Luo, 2018; Zimmert & Seldin, 2021; Masoudian & Seldin, 2021; Ito, 2021). A prominent line of work builds on follow-the-regularized-leader (FTRL), showing that suitable regularization yields algorithms that are automatically adaptive between adversarial and stochastic regimes (Wei & Luo, 2018; Zimmert & Seldin, 2021; Ito, 2021). In these algorithms, stochastic-regime bounds are obtained via the self-bounding technique (Zimmert & Seldin, 2021; Masoudian & Seldin, 2021), which also extends naturally to adversarially corrupted stochastic settings. FTRL-based approaches have also been developed in other settings, including linear bandits (Lee et al., 2021; Dann et al., 2023b; Ito & Takemura, 2023), contextual bandits (Dann et al., 2023b), combinatorial semi-bandits (Tsuchiya et al., 2023), and online learning with feedback graphs (Erez & Koren, 2021; Ito et al., 2022b).

Beyond multi-armed bandits, best-of-both-worlds algorithms have been extended to MDPs. For global optimization, Jin & Luo (2020); Jin et al. (2021) developed best-of-both-worlds algorithms. In particular, Jin et al. (2021) introduced the loss-shifting technique, which served as a key component in obtaining stochastic regret bounds. This idea has been further applied to more challenging settings, including adversarial transitions and aggregate feedback (Jin et al., 2023; Ito et al., 2025). For policy optimization, Dann et al. (2023a) established best-of-both-worlds guarantees under bandit feedback, covering Tsallis entropy, Shannon entropy, and log-barrier regularizers.

**Data-dependent bounds in the adversarial regime.** The worst-case analysis, which is driven by the worst-case instance within a problem class, can be overly pessimistic for practical environments. Accordingly, in the broader online learning literature–including learning with expert advice (Littlestone & Warmuth, 1994), multi-armed bandits (Auer et al., 2002), and online convex optimization (Zinkevich, 2003)–it has been shown that regret can often be upper bounded by refined data-dependent complexity measures (Cesa-Bianchi et al., 1996; Allenberg et al., 2006; Neu, 2015). There are several data-dependent complexity measures for adversarial regimes. The *first-order* complexity $L^\star$ scales with the cumulative loss of the best action, yielding $\widetilde{O}(\sqrt{L^\star})$-type regret (Allenberg et al., 2006; Neu, 2015; Zimmert & Seldin, 2021; Ito, 2021). The *second-order* complexity $Q_\infty$ quantifies the magnitude of loss fluctuations, leading to $\widetilde{O}(\sqrt{Q_\infty})$-type regret (Hazan & Kale, 2011; Wei & Luo, 2018; Ito et al., 2022a), path-length complexity $V_1 = \sum_{t=1}^{T-1} \|\ell_{t+1} - \ell_t\|_1$ depends on the cumulative variation of the loss sequence, giving $\widetilde{O}(\sqrt{V_1})$-type regret (Wei & Luo, 2018; Ito et al., 2022a).

Extending data-dependent complexity measures from bandits to Markov decision processes remains an active and challenging direction. For MDPs, first-order regret complexity was introduced by Lee et al. (2020), who showed that under unknown transitions one can achieve a first-order bound $\widetilde{\mathcal{O}}(\sqrt{HS^2AL^\star})$ in global optimization. For policy optimization, Dann et al. (2023a) established best-of-both-worlds guarantees with a first-order bound under known transitions.

In contrast, second-order and path-length measures have been much less studied in MDPs. In this work, we introduce these notions (see (4)–(6)) and prove regret bounds for them when the transition is known. Our results cover both global optimization and policy optimization in the best-of-both-worlds setting. The case with unknown transitions is still largely open. The main obstacle is the error caused by estimating the transition kernel. It is not clear how to control this error with data-dependent quantities. As a result, even first-order guarantees in the best-of-both-worlds regime are not known when transitions are unknown. For policy optimization with unknown transitions, it is not even known whether any data-dependent guarantee is possible without best-of-both-worlds adaptivity (Dann et al., 2023a).

**Variance-dependent bounds in the stochastic regime.** In stochastic regimes, variance-aware and gap-dependent bounds originate from UCB-V (Audibert et al., 2007), which augments UCB with empirical variance-based bonuses and yields tighter regret when variances are small. In the context of best-of-both-worlds algorithms, FTRL-based methods can incorporate variance information into the optimization, achieving variance-dependent regret guarantees while retaining robustness in adversarial settings (Ito et al., 2022a; Tsuchiya et al., 2023; Ito & Takemura, 2023).

For episodic stochastic MDPs, Zanette & Brunskill (2019) were among the first to obtain variance-dependent guarantees for model-based optimistic methods, introducing a maximum per-step conditional variance. Using this type of variance, Simchowitz & Jamieson (2019) derive variance-aware gap-dependent guarantees. From a value-based perspective, the Monotonic Value Propagation (MVP) algorithm provides a baseline via optimistic value iteration with Bernstein-type bonuses (Zhang et al., 2021), and subsequent works establish variance-dependent bounds in terms of the maximum total variance (Zhou et al., 2023; Zhang et al., 2024; Zheng et al., 2025). Continuing this line, Chen et al. (2025) derive gap-dependent guarantees by the maximum conditional total variance, explicitly conditioning variance on the state. Related variance-based guarantees have also been explored in linear contextual bandits and linear MDPs (Wagenmaker et al., 2022; Zhao et al., 2023). To our knowledge, variance-dependent guarantees have not been established within FTRL-based analyses for MDPs.

## B.2. Comparison with Existing Variance Measures

We restate our variance measures in (7) and (8).

$$\mathbb{V} = \max_{\pi \in \Pi} \sum_{s,a} q^\pi(s,a)\sigma^2(s,a) \in \left[0, \frac{H}{4}\right],$$

$$\mathbb{V}^c(s) = \max_{\pi \in \Pi, a \in \mathcal{A}} \sum_{s',a'} q^\pi(s',a' \mid s,a)\sigma^2(s',a') \in \left[0, \frac{H}{4}\right].$$

Most model-based and value-based algorithms focus on the stochastic setting with unknown transitions and without corruption. In particular, Zanette & Brunskill (2019); Simchowitz & Jamieson (2019) introduce

$$\mathsf{Var}^\star(s,a) = \sigma^2(s,a) + \mathsf{Var}_{s' \sim P(\cdot|s,a)}[V^{\pi^*}(s')] \in \left[0, \frac{H^2}{4}\right],$$

and $\mathbb{Q}^\star = \max_{s,a} \mathsf{Var}^\star(s,a)$, where $\pi^*$ differs slightly from our definition of $\pi^\star$, in that it is defined as a policy that simultaneously achieves the minimum of $Q(s,a)$ and $V(s)$ for all state-action pairs $(s,a)$ with uncorrupted loss.

Subsequent works further aggregate $\mathsf{Var}^\star$ along trajectories and consider total-variance measures. Zhou et al. (2023); Zhang et al. (2024) define an unconditional total variance, while Chen et al. (2025) introduce a conditional total variance,

$$\mathsf{Var}_{\max} = \max_{\pi \in \Pi} \sum_{s,a} q^\pi(s,a)\mathsf{Var}^\star(s,a) \in \left[0, \frac{H^3}{4}\right],$$

$$\mathsf{Var}_{\max}^c = \max_{\pi \in \Pi, s \in \mathcal{S}} \sum_{s',a'} \bar{q}^\pi(s',a' \mid s)\mathsf{Var}^\star(s',a') \in \left[0, \frac{H^3}{4}\right],$$

where $\bar{q}^\pi(s',a' \mid s)$ denotes the occupancy of $(s',a')$ over the entire trajectory conditioned on visiting state $s$. In our setting, $\mathbb{V}$ and $\mathbb{V}^c$ are analogous to $\mathsf{Var}_{\max}$ and $\mathsf{Var}_{\max}^c$, respectively, and can be interpreted as natural variance measures for MDPs with known transitions. Since we consider known transitions, the second term of $\mathsf{Var}^\star(s,a)$ is unnecessary and can be omitted. Moreover, $\mathbb{V}^c$ is defined using conditional occupancy measures $q^\pi(s',a' \mid s,a)$ and captures variance only after visiting $(s,a)$, whereas $\mathsf{Var}_{\max}^c$ aggregates variance over the entire trajectory by conditioning on the event of visiting state $s$. As a consequence, our variance measures are $H^2$-sharper than those based on $\mathsf{Var}^\star(s,a)$. Moreover, unlike $\mathsf{Var}_{\max}^c$, our $\mathbb{V}^c(s)$ is state-dependent, resulting in a more refined and potentially smaller regret bound.

The correspondence between $\mathbb{V}$ and $\mathsf{Var}_{\max}$, $\mathbb{V}^c$ and $\mathsf{Var}_{\max}^c$ can also be seen from regret bounds. Our variance-dependent leading term scales as $O(\sqrt{\mathbb{V}T})$, mirroring the $O(\sqrt{\mathsf{Var}_{\max}T})$ dependence in Zhou et al. (2023). In gap-dependent regret bounds, variance typically appears as a multiplicative coefficient of the suboptimality gap: in our bound this coefficient is $H\mathbb{V}^c(s)$ (whose worst-case value matches that of Dann et al. 2023a), while Simchowitz & Jamieson (2019) use $H\mathbb{Q}^\star$ and Chen et al. (2025) use $\min\{H^2, \mathsf{Var}_{\max}^c\}$.

## C. Regret Analysis of Optimistic Follow-the-Regularized-Leader

In this section, we provide a regret analysis of optimistic follow-the-regularized-leader (OFTRL) for the MDP setting. General OFTRL bounds of this type appear in Ito et al. (2022a, Lemma 1) and Tsuchiya et al. (2025a, Lemma 16). For completeness, we restate the argument here. We then specialize the bound to our two instances, Lemma C.3 for global optimization and Lemma C.4 for policy optimization. These lemmas will be used in Appendices D and E, respectively. Given a strictly convex function $\psi$, we use $D_\psi(y,x) := \psi(y) - \psi(x) - \langle \nabla\psi(x), y - x \rangle$ to denote the Bregman divergence induced by $\psi$.

**Lemma C.1.** *Let $\{\ell_t\}_{t=1}^T$ be a sequence of loss vectors. Suppose that $p_t$ is defined by the OFTRL algorithm over a convex set $\mathcal{K}$ and differentiable regularizers $\{\psi_t\}_{t=1}^{T+1}$ and loss predictions $\{m_t\}_{t=1}^{T+1}$:*

$$p_t = \arg\min_{p \in \mathcal{K}} \left\{ \left\langle p, \sum_{\tau=1}^{t-1} \ell_\tau + m_t \right\rangle + \psi_t(p) \right\}. \tag{21}$$

*Then, for any $u \in \mathcal{K}$ it holds that*

$$\sum_{t=1}^T \langle p_t - u, \ell_t \rangle \leq \psi_1(u) - \psi_1(p_1) + \sum_{t=1}^T (\psi_{t+1}(u) - \psi_t(u) + \psi_t(p_{t+1}) - \psi_{t+1}(p_{t+1}))$$

$$+ \sum_{t=1}^T (\langle p_t - p_{t+1}, \ell_t - m_t \rangle - D_{\psi_t}(p_{t+1}, p_t)) + \langle u - p_{T+1}, m_{T+1} \rangle.$$

*Proof.* Define $\widetilde{\psi}_t(p) = \psi_t(p) + \langle p, m_t \rangle$, and let

$$F_t(p) = \left\langle p, \sum_{\tau=1}^{t-1} \ell_\tau + m_t \right\rangle + \psi_t(p) = \left\langle p, \sum_{\tau=1}^{t-1} \ell_\tau \right\rangle + \widetilde{\psi}_t(p).$$

Since $-\sum_{t=1}^{T}\langle u, \ell_t\rangle = \widetilde{\psi}_{T+1}(u) - F_{T+1}(u)$, we have

$$
\begin{aligned}
\sum_{t=1}^{T}\langle p_t - u, \ell_t\rangle &= \sum_{t=1}^{T}\langle p_t, \ell_t\rangle + \widetilde{\psi}_{T+1}(u) - F_{T+1}(u) \\
&= \sum_{t=1}^{T}\langle p_t, \ell_t\rangle + \widetilde{\psi}_{T+1}(u) - F_{T+1}(u) - F_1(p_1) + F_1(p_1) - F_{T+1}(p_{T+1}) + F_{T+1}(p_{T+1}) \\
&= \sum_{t=1}^{T}\langle p_t, \ell_t\rangle + \widetilde{\psi}_{T+1}(u) - F_{T+1}(u) - F_1(p_1) + \sum_{t=1}^{T}(F_t(p_t) - F_{t+1}(p_{t+1})) + F_{T+1}(p_{T+1}) \\
&\le \widetilde{\psi}_{T+1}(u) - \widetilde{\psi}_1(p_1) + \sum_{t=1}^{T}(F_t(p_t) - F_{t+1}(p_{t+1}) + \langle p_t, \ell_t\rangle), \qquad (22)
\end{aligned}
$$

where the last inequality follows from $F_{T+1}(p_{T+1}) - F_{T+1}(u) \le 0$ because $p_{T+1}$ is the minimizer of $F_{T+1}$ and $F_1(p_1) = \widetilde{\psi}_1(p_1)$ by definition.

Hence,

$$
\begin{aligned}
&F_t(p_t) - F_{t+1}(p_{t+1}) + \langle p_t, \ell_t\rangle \\
&= F_t(p_t) - F_t(p_{t+1}) + F_t(p_{t+1}) - F_{t+1}(p_{t+1}) + \langle p_t, \ell_t\rangle \\
&= F_t(p_t) - F_t(p_{t+1}) - \langle p_{t+1}, \ell_t\rangle + \widetilde{\psi}_t(p_{t+1}) - \widetilde{\psi}_{t+1}(p_{t+1}) + \langle p_t, \ell_t\rangle \\
&= F_t(p_t) - F_t(p_{t+1}) + \langle p_t - p_{t+1}, \ell_t\rangle + \psi_t(p_{t+1}) - \psi_{t+1}(p_{t+1}) + \langle p_{t+1}, m_t - m_{t+1}\rangle \\
&\le -D_{F_t}(p_{t+1}, p_t) + \langle p_t - p_{t+1}, \ell_t\rangle + \psi_t(p_{t+1}) - \psi_{t+1}(p_{t+1}) + \langle p_{t+1}, m_t - m_{t+1}\rangle, \qquad (23)
\end{aligned}
$$

where the last inequality follows from first-order optimality. Since $F_t$ is convex and differentiable and $p_t \in \arg\min_{p\in\mathcal{K}} F_t(p)$, we have $\langle \nabla F_t(p_t), y - p_t\rangle \ge 0$ for all $y \in \mathcal{K}$. Using the definition of the Bregman divergence $D_{F_t}(p_{t+1}, p_t) = F_t(p_{t+1}) - F_t(p_t) - \langle p_{t+1} - p_t, \nabla F_t(p_t)\rangle$, we obtain

$$
F_t(p_t) - F_t(p_{t+1}) = -D_{F_t}(p_{t+1}, p_t) - \langle \nabla F_t(p_t), p_{t+1} - p_t\rangle \le -D_{F_t}(p_{t+1}, p_t).
$$

Then, from (22) and (23) and using that adding a linear term does not change the Bregman divergence ($D_{F_t} = D_{\psi_t}$), we obtain

$$
\begin{aligned}
\sum_{t=1}^{T}\langle p_t - u, \ell_t\rangle &\le \widetilde{\psi}_{T+1}(u) - \widetilde{\psi}_1(p_1) + \sum_{t=1}^{T}(\psi_t(p_{t+1}) - \psi_{t+1}(p_{t+1}) - D_{\psi_t}(p_{t+1}, p_t)) \\
&\quad + \sum_{t=1}^{T}\langle p_t - p_{t+1}, \ell_t\rangle + \sum_{t=1}^{T}\langle p_{t+1}, m_t - m_{t+1}\rangle \\
&= \psi_{T+1}(u) - \psi_1(p_1) + \sum_{t=1}^{T}(\psi_t(p_{t+1}) - \psi_{t+1}(p_{t+1}) - D_{\psi_t}(p_{t+1}, p_t)) \\
&\quad + \sum_{t=1}^{T}\langle p_t - p_{t+1}, \ell_t\rangle + \sum_{t=1}^{T}\langle p_{t+1} - p_t, m_t\rangle + \langle u - p_{T+1}, m_{T+1}\rangle \\
&= \psi_1(u) - \psi_1(p_1) + \sum_{t=1}^{T}(\psi_{t+1}(u) - \psi_t(u) + \psi_t(p_{t+1}) - \psi_{t+1}(p_{t+1}) - D_{\psi_t}(p_{t+1}, p_t)) \\
&\quad + \sum_{t=1}^{T}\langle p_t - p_{t+1}, \ell_t - m_t\rangle + \langle u - p_{T+1}, m_{T+1}\rangle,
\end{aligned}
$$

which completes the proof. $\qquad\square$

**Lemma C.2.** *Let $D_\phi$ denote the Bregman divergence associated with $\phi(x) = -\log x$ and define $g(x) = x - \log(1+x)$. Then, for any $x > 0$ and $a \geq -1/x$, it holds that*

$$\max_{y \in \mathbb{R}_{>0}} \{a(x-y) - D_\phi(y,x)\} = g(ax).$$

*Proof.* This can be proven by simply considering the worst-case w.r.t. $y$ and the proof can be found *e.g.,* in Ito et al. (2022a, Lemma 5). $\qquad\square$

The following lemma will be used in the regret analysis for global optimization. There are a few prior works that analyze global optimization with time-varying log-barrier learning rates (see Jin et al. 2023, Appendix C.7.2 for a related approach).

**Lemma C.3** (OFTRL with global optimization). *Suppose that a sequence of occupancy measures $p_1, \ldots, p_T \in \Omega(P)$ is given by OFTRL in (21) with regularizer $\psi_t$ given by $\psi_t(p) = \sum_{s,a} \frac{1}{\eta_t(s,a)} \log\left(\frac{1}{p(s,a)}\right)$ as in (9), for some nonincreasing learning rate $\eta_t(s,a)$ with $\eta_1(s,a) = \eta_1$ for all $s, a$, and let losses $\{\ell_t\}_{t=1}^T$ and loss predictions $\{m_t\}_{t=1}^{T+1}$ satisfy*

$$\eta_t(s,a) p_t(s,a)(\ell_t(s,a) - m_t(s,a)) \geq -\frac{1}{2} \tag{24}$$

*for all $t, s, a$. Then, for any $u \in \Omega(P)$, it holds that*

$$\sum_{t=1}^T \langle p_t - u, \ell_t \rangle \leq \frac{SA \log(SAT)}{\eta_1} + \sum_{t=1}^T \sum_{s,a} \left( \frac{1}{\eta_{t+1}(s,a)} - \frac{1}{\eta_t(s,a)} \right) \log(SAT)$$

$$+ \sum_{t=1}^T \sum_{s,a} \eta_t(s,a) p_t(s,a)^2 (\ell_t(s,a) - m_t(s,a))^2$$

$$+ \frac{1}{T} \sum_{t=1}^T \left\langle -u + \frac{1}{SA} \sum_{s,a} p_{s,a}^{\max}, \ell_t \right\rangle + 2H \|m_{T+1}\|_\infty,$$

*where $p_{s,a}^{\max}$ denotes the occupancy measure induced by a policy that maximizes the probability of visiting state-action pair $(s,a)$ in transition $P$.*

*Proof.* Let

$$u' = \left(1 - \frac{1}{T}\right) u + \frac{1}{TSA} \sum_{s,a} p_{s,a}^{\max}.$$

Then we have $u' \in \Omega(P)$, since $u \in \Omega(P)$, each $p_{s,a}^{\max} \in \Omega(P)$, and $\Omega(P)$ is convex. We also define $\phi(x) = \log\left(\frac{1}{x}\right)$. Thus, by the definition of the regularizer, we have

$$\psi_t(u') = \sum_{s,a} \frac{1}{\eta_t(s,a)} \log\left(\frac{1}{u'(s,a)}\right) \leq \sum_{s,a} \frac{1}{\eta_t(s,a)} \log\left(\frac{SAT}{p_{s,a}^{\max}(s,a)}\right).$$

First, for any $u \in \Omega(P)$, we decompose the regret as

$$\sum_{t=1}^T \langle p_t - u, \ell_t \rangle = \sum_{t=1}^T \langle p_t - u', \ell_t \rangle + \sum_{t=1}^T \langle u' - u, \ell_t \rangle$$

$$= \sum_{t=1}^T \langle p_t - u', \ell_t \rangle + \frac{1}{T} \sum_{t=1}^T \left\langle -u + \frac{1}{SA} \sum_{s,a} p_{s,a}^{\max}, \ell_t \right\rangle. \tag{25}$$

Using Lemma C.1, the first term in the last equality is upper-bounded by

$$\sum_{t=1}^{T} \langle p_t - u', \ell_t \rangle \leq \underbrace{\psi_1(u') - \psi_1(p_1) + \sum_{t=1}^{T} (\psi_{t+1}(u') - \psi_t(u') + \psi_t(p_{t+1}) - \psi_{t+1}(p_{t+1}))}_{\text{penalty-term}}$$

$$+ \underbrace{\sum_{t=1}^{T} (\langle p_t - p_{t+1}, \ell_t - m_t \rangle - D_{\psi_t}(p_{t+1}, p_t)) + \langle u' - p_{T+1}, m_{T+1} \rangle}_{\text{stability-term}}.$$

The penalty-term can be upper bounded by

$$\text{penalty-term} \leq \sum_{s,a} \frac{1}{\eta_1} \log\left(\frac{p_1(s,a)}{u'(s,a)}\right) + \sum_{t=1}^{T} \sum_{s,a} \left(\frac{1}{\eta_{t+1}(s,a)} - \frac{1}{\eta_t(s,a)}\right)(\phi(u'(s,a)) - \phi(p_{t+1}(s,a)))$$

$$\leq \sum_{s,a} \frac{1}{\eta_1} \log\left(\frac{p_1(s,a)}{u'(s,a)}\right) + \sum_{t=1}^{T} \sum_{s,a} \left(\frac{1}{\eta_{t+1}(s,a)} - \frac{1}{\eta_t(s,a)}\right) \log\left(\frac{p_{t+1}(s,a)}{u'(s,a)}\right)$$

$$\leq \sum_{s,a} \frac{1}{\eta_1} \log\left(\frac{SATp_1(s,a)}{p_{s,a}^{\max}(s,a)}\right) + \sum_{t=1}^{T} \sum_{s,a} \left(\frac{1}{\eta_{t+1}(s,a)} - \frac{1}{\eta_t(s,a)}\right) \log\left(\frac{SATp_{t+1}(s,a)}{p_{s,a}^{\max}(s,a)}\right)$$

$$\leq \frac{SA\log(SAT)}{\eta_1} + \sum_{t=1}^{T} \sum_{s,a} \left(\frac{1}{\eta_{t+1}(s,a)} - \frac{1}{\eta_t(s,a)}\right) \log(SAT), \tag{26}$$

where the last inequality follows from $p_\tau(s,a) \leq p_{s,a}^{\max}(s,a)$ for all $\tau \leq T+1$.

Next, we bound the stability term. The Bregman divergence $D_{\psi_t}(p_{t+1}, p_t)$ can be written as

$$D_{\psi_t}(p_{t+1}(s,a), p_t(s,a)) = \sum_{s,a} \frac{1}{\eta_t(s,a)} D_\phi(p_{t+1}(s,a), p_t(s,a)),$$

where we recall that $\phi(x) = -\log x$ and $g(x) = x - \log(1+x)$. By using Lemma C.2, we have

$$\langle p_t - p_{t+1}, \ell_t - m_t \rangle - D_{\psi_t}(p_{t+1}, p_t)$$

$$\leq \sum_{s,a} \left((\ell_t(s,a) - m_t(s,a))(p_t(s,a) - p_{t+1}(s,a)) - \frac{1}{\eta_t(s,a)} D_\phi(p_{t+1}(s,a), p_t(s,a))\right)$$

$$\leq \sum_{s,a} \frac{1}{\eta_t(s,a)} g(\eta_t(s,a)(\ell_t(s,a) - m_t(s,a))p_t(s,a)) \qquad \text{(by Lemma C.2 and (24))}$$

$$\leq \sum_{s,a} \eta_t(s,a)p_t(s,a)^2(\ell_t(s,a) - m_t(s,a))^2,$$

where the last inequality follows from $g(x) = x - \log(1+x) \leq x^2$ for $x \geq -\frac{1}{2}$ and (24).

Therefore,

$$\text{stability-term} = \sum_{t=1}^{T} (\langle p_t - p_{t+1}, \ell_t - m_t \rangle - D_{\psi_t}(p_{t+1}, p_t)) + \langle u - p_{T+1}, m_{T+1} \rangle$$

$$\leq \sum_{t=1}^{T} \sum_{s,a} \eta_t(s,a)p_t(s,a)^2(\ell_t(s,a) - m_t(s,a))^2 + 2H\|m_{T+1}\|_\infty, \tag{27}$$

where the last bound follows from Hölder's inequality $\langle u - p_{T+1}, m_{T+1} \rangle \leq \|u - p_{T+1}\|_1 \|m_{T+1}\|_\infty \leq 2H\|m_{T+1}\|_\infty$, since $u, p_{T+1} \in \Omega(P)$ imply $\|u - p_{T+1}\|_1 \leq 2H$. Combining (25)–(27) completes the proof. □

The following lemma will be used in the regret analysis for policy optimization.

**Lemma C.4** (OFTRL with policy optimization). *Suppose that a sequence of probability vectors $p_1, \ldots, p_T \in \triangle(\mathcal{A})$ is given by OFTRL in (21) with regularizer $\psi_t(p)$ in (17) for some nonincreasing learning rate $\eta_t(a)$ with $\eta_1(a) = \eta_1$ for all $a$, and let losses $\{\ell_t\}_{t=1}^T$, loss predictions $\{m_t\}_{t=1}^{T+1}$ and $\{x_t\}_{t=1}^T$ be such that*

$$\eta_t(a)p_t(a)(\ell_t(a) - m_t(a) + x_t(a)) \geq -\frac{1}{2} \tag{28}$$

*for all $t, a$. Then for any $u \in \triangle(\mathcal{A})$, the OFTRL algorithm achieves*

$$\sum_{t=1}^T \langle p_t - u, \ell_t \rangle \leq \frac{A \log(AT^2)}{\eta_1} + \sum_{t=1}^T \sum_a \left( \frac{1}{\eta_{t+1}(a)} - \frac{1}{\eta_t(a)} \right) \log(AT^2)$$

$$+ \sum_{t=1}^T \sum_a \eta_t(a)p_t(a)^2(\ell_t(a) - m_t(a) + x_t(a))^2$$

$$+ \frac{1}{T^2} \sum_{t=1}^T \left\langle -u + \frac{1}{A}\mathbf{1}, \ell_t \right\rangle + 2\|m_{T+1}\|_\infty.$$

*Proof.* Let $u' = \left(1 - \frac{1}{T^2}\right)u + \frac{1}{AT^2}\mathbf{1} \in \triangle(\mathcal{A})$ and $\phi(x) = \log\left(\frac{1}{x}\right)$. Then, we have

$$\psi_t(u') = \sum_a \frac{1}{\eta_t(a)} \log\left(\frac{1}{u'(a)}\right) \leq \sum_a \frac{1}{\eta_t(a)} \log(AT^2) \leq \sum_a \frac{1}{\eta_t(a)} \log(AT^2).$$

First, for any $u \in \triangle(\mathcal{A})$, we decompose the regret as

$$\sum_{t=1}^T \langle p_t - u, \ell_t \rangle = \sum_{t=1}^T \langle p_t - u', \ell_t \rangle + \sum_{t=1}^T \langle u' - u, \ell_t \rangle$$

$$= \sum_{t=1}^T \langle p_t - u', \ell_t \rangle + \frac{1}{T^2} \sum_{t=1}^T \left\langle -u + \frac{1}{A}\mathbf{1}, \ell_t \right\rangle. \tag{29}$$

For the first term, using Lemma C.1, we obtain

$$\sum_{t=1}^T \langle p_t - u', \ell_t \rangle \leq \underbrace{\psi_1(u') - \psi_1(p_1) + \sum_{t=1}^T (\psi_{t+1}(u') - \psi_t(u') + \psi_t(p_{t+1}) - \psi_{t+1}(p_{t+1}))}_{\text{penalty-term}}$$

$$+ \underbrace{\sum_{t=1}^T (\langle p_t - p_{t+1}, \ell_t - m_t \rangle - D_{\psi_t}(p_{t+1}, p_t)) + \langle u' - p_{T+1}, m_{T+1} \rangle}_{\text{stability-term}}.$$

Then,

$$\text{penalty-term} \leq \frac{A \log(AT^2)}{\eta_1} + \sum_{t=1}^T \sum_a \left( \frac{1}{\eta_{t+1}(a)} - \frac{1}{\eta_t(a)} \right) (\phi(u'(a)) - \phi(p_{t+1}(a)))$$

$$\leq \frac{A \log(AT^2)}{\eta_1} + \sum_{t=1}^T \sum_a \left( \frac{1}{\eta_{t+1}(a)} - \frac{1}{\eta_t(a)} \right) \log(AT^2), \tag{30}$$

where the last inequality follows from $\phi(p_{t+1}(a)) \geq 0$.

Next, we bound the stability term. The Bregman divergence $D_{\psi_t}(p_{t+1}, p_t)$ can be written as

$$D_{\psi_t}(p_{t+1}(a), p_t(a)) = \sum_a \frac{1}{\eta_t(a)} D_\phi(p_{t+1}(a), p_t(a)),$$

where $D_\phi$ denotes the Bregman divergence associated with $\phi(x) = -\log(x)$. Recall that $g(x) = x - \log(1+x)$ by Lemma C.2, we have

$$
\begin{aligned}
&\langle p_t - p_{t+1}, \ell_t - m_t \rangle - D_{\psi_t}(p_{t+1}, p_t) \\
&= \langle p_t - p_{t+1}, \ell_t - m_t + x_t \mathbf{1} \rangle - D_{\psi_t}(p_{t+1}, p_t) \\
&\leq \sum_a \left( (\ell_t(a) - m_t(a) + x_t)(p_t(a) - p_{t+1}(a)) - \frac{1}{\eta_t(a)} D_\phi(p_{t+1}(a), p_t(a)) \right) \\
&\leq \sum_a \frac{1}{\eta_t(a)} g(\eta_t(a)(\ell_t(a) - m_t(a) + x_t) p_t(a)) \qquad\qquad \text{(by Lemma C.2 and (28))} \\
&\leq \sum_a \eta_t(a) p_t(a)^2 (\ell_t(a) - m_t(a) + x_t)^2,
\end{aligned}
$$

where the last inequality follows from $g(x) = x - \log(1+x) \leq x^2$ for $x \geq -\frac{1}{2}$ and (28).

Therefore,

$$
\begin{aligned}
\text{stability-term} &= \sum_{t=1}^{T} (\langle p_t - p_{t+1}, \ell_t - m_t \rangle - D_{\psi_t}(p_{t+1}, p_t)) + \langle u - p_{T+1}, m_{T+1} \rangle \\
&\leq \sum_{t=1}^{T} \sum_a \eta_t(a) p_t(a)^2 (\ell_t(a) - m_t(a) + x_t)^2 + 2\|m_{T+1}\|_\infty, \qquad (31)
\end{aligned}
$$

where the last bound follows from Hölder's inequality $\langle u - p_{T+1}, m_{T+1} \rangle \leq \|u - p_{T+1}\|_1 \|m_{T+1}\|_\infty \leq 2\|m_{T+1}\|_\infty$, since $u, p_{T+1} \in \triangle(\mathcal{A})$ implies $\|u - p_{T+1}\|_1 \leq 2$.

Combining (29)–(31) completes the proof. $\qquad\square$

## D. Regret Analysis of Global Optimization (deferred from Section 4)

In this section, we provide the details omitted from Section 4 and complete the regret analysis for Theorems 4.1 and 4.2.

### D.1. Auxiliary Lemmas

We first note the unbiasedness of the estimator in (10). For any state-action pair $(s, a)$,

$$
\mathbb{E}_t \left[ \widehat{\ell}_t(s, a) \right] = m_t(s, a) + \frac{\ell_t(s, a) - m_t(s, a)}{q^{\pi_t}(s, a)} \mathbb{E}_t[\mathbb{I}_t(s, a)] = \ell_t(s, a).
$$

Hence, using this and $V^\pi(s_0; \ell_t) = \sum_{s,a} q^\pi(s, a)\ell_t(s, a) = \langle q^\pi, \ell_t \rangle$, we can rewrite the regret as follows:

$$
\begin{aligned}
\text{Reg}_T &= \mathbb{E} \left[ \sum_{t=1}^{T} V^{\pi_t}(s_0; \ell_t) - \sum_{t=1}^{T} V^{\mathring{\pi}}(s_0; \ell_t) \right] \\
&= \mathbb{E} \left[ \sum_{t=1}^{T} \langle q^{\pi_t} - q^{\mathring{\pi}}, \ell_t \rangle \right] \\
&= \mathbb{E} \left[ \sum_{t=1}^{T} \langle q^{\pi_t} - q^{\mathring{\pi}}, \widehat{\ell}_t \rangle \right]. \qquad (32)
\end{aligned}
$$

We also recall the loss-shifting technique introduced by Jin et al. (2021), which is useful to prove logarithmic regret bounds in the stochastic regime.

**Lemma D.1** (special case of Jin et al. 2021, Lemma A.1.1)**.** *Fix the transition function $P$. For any policy $\pi$ and loss function $\mathring{\ell}: \mathcal{S} \times \mathcal{A} \to \mathbb{R}$, define the invariant function $g: \mathcal{S} \times \mathcal{A} \to \mathbb{R}$ as*

$$
g^\pi(s, a; \mathring{\ell}) := Q^\pi(s, a; \mathring{\ell}) - V^\pi(s; \mathring{\ell}) - \mathring{\ell}(s, a).
$$

*Then, it holds for any policy $\pi'$ that*

$$\left\langle q^{P,\pi'}(\cdot,\cdot), g^{\pi}(\cdot,\cdot;\mathring{\ell}) \right\rangle := \sum_{s,a} q^{\pi'}(s,a)\, g^{\pi}(s,a;\mathring{\ell}) = -V^{\pi}(s_0;\mathring{\ell}),$$

*where $V^{\pi}(s_0;\mathring{\ell})$ only depends on $\pi$ and $\mathring{\ell}$ (but not $\pi'$).*

The following lemma extends Jin et al. (2021, Lemma A.1.2) from standard FTRL to OFTRL. It immediately follows from Lemma D.1.

**Lemma D.2.** *Consider the occupancy measure $q^{\pi_t}$ selected by OFTRL with regularizer $\psi_t$, loss sequence $\{\widehat{\ell}_\tau\}_{\tau<t}$, and predictor $m_t$ over the decision set $\Omega(P)$. Then,*

$$q^{\pi_t} = \underset{q \in \Omega(P)}{\arg\min}\left\{ \left\langle q, \sum_{\tau=1}^{t-1} \widehat{\ell}_\tau + m_t \right\rangle + \psi_t(q) \right\} = \underset{q \in \Omega(P)}{\arg\min}\left\{ \left\langle q, \sum_{\tau=1}^{t-1} (\widehat{\ell}_\tau + g_\tau) + m_t \right\rangle + \psi_t(q) \right\}.$$

*for any sequence of invariant functions $\{g_\tau\}_{\tau<t}$ which are constructed with hypothesized losses $\{\mathring{\ell}_\tau\}_{\tau<t}$ and policies $\{\pi'_\tau\}_{\tau<t}$.*

Since we use the OFTRL framework, we slightly modify the loss-shifting construction of Jin et al. (2021). In their analysis, the invariant function $g^{\pi_t}(s,a;\widehat{\ell})$ is defined using the estimated loss $\widehat{\ell}_t$. By contrast, as can be seen from Lemmas C.3 and C.4, the stability of OFTRL is controlled by the shifted loss $\widetilde{\ell}_t := \widehat{\ell}_t - m_t$. Accordingly, we construct the invariant function from $\widehat{\ell}_t - m_t$ so that the invariance property in Lemma D.2 holds under OFTRL.

To this end, we define $\widetilde{\ell}_t$ by

$$\widetilde{\ell}_t(s,a) := \frac{\mathbb{I}_t(s,a)(\ell_t(s,a) - m_t(s,a))}{q^{\pi_t}(s,a)} = \widehat{\ell}_t(s,a) - m_t(s,a).$$

We then define the corresponding loss-shifting (invariant) function induced by $\widetilde{\ell}_t$ as

$$g_t(s,a) := Q^{\pi_t}(s,a;\widetilde{\ell}_t) - V^{\pi_t}(s;\widetilde{\ell}_t) - \widetilde{\ell}_t(s,a).$$

In what follows, we collect several basic properties of $g_t$ for bounding the regret. All of them hold for an arbitrary loss prediction $m_t \in [0,1]^{S \times A}$.

**Lemma D.3.** *For any loss prediction $m_t \in [0,1]^{S \times A}$, it holds that*

$$\widehat{\ell}_t(s,a) + g_t(s,a) - m_t(s,a) \geq \frac{-H}{q^{\pi_t}(s,a)}$$

*for all state-action pairs $(s,a)$.*

*Proof.* Fix any state-action pair $(s,a)$. By the definitions of $g_t$ and $\widetilde{\ell}_t$, we have

$$\begin{aligned}
\widehat{\ell}_t(s,a) + g_t(s,a) - m_t(s,a) &= \widehat{\ell}_t(s,a) + Q^{\pi_t}(s,a;\widetilde{\ell}_t) - V^{\pi_t}(s;\widetilde{\ell}_t) - \widetilde{\ell}_t(s,a) - m_t(s,a) \\
&= Q^{\pi_t}(s,a;\widetilde{\ell}_t) - V^{\pi_t}(s;\widetilde{\ell}_t) \\
&= (1 - \pi_t(a \mid s))Q^{\pi_t}(s,a;\widetilde{\ell}_t) - \sum_{b \neq a} \pi_t(b \mid s)Q^{\pi_t}(s,b;\widetilde{\ell}_t),
\end{aligned} \tag{33}$$

where the last line uses $V^{\pi_t}(s;\widetilde{\ell}_t) = \sum_b \pi_t(b \mid s)Q^{\pi_t}(s,b;\widetilde{\ell}_t)$.

We first lower bound $Q^{\pi_t}(s, a; \widetilde{\ell}_t)$. By the definition of the $Q$-function,

$$
\begin{aligned}
Q^{\pi_t}(s, a; \widetilde{\ell}_t) &= \sum_{h=h(s)}^{H-1} \sum_{(s',a') \in \mathcal{S}_h \times \mathcal{A}} q^{\pi_t}(s', a' \mid s, a) \widetilde{\ell}_t(s', a') \\
&= \sum_{h=h(s)}^{H-1} \sum_{(s',a') \in \mathcal{S}_h \times \mathcal{A}} q^{\pi_t}(s', a' \mid s, a) \frac{\mathbb{I}_t(s', a')(\ell_t(s', a') - m_t(s', a'))}{q^{\pi_t}(s', a')} \\
&\geq \frac{-1}{q^{\pi_t}(s, a)} \sum_{h=h(s)}^{H-1} \sum_{(s',a') \in \mathcal{S}_h \times \mathcal{A}} \frac{q^{\pi_t}(s, a) q^{\pi_t}(s', a' \mid s, a)}{q^{\pi_t}(s', a')} \mathbb{I}_t(s', a') \\
&\geq \frac{-1}{q^{\pi_t}(s, a)} \sum_{h=h(s)}^{H-1} \sum_{(s',a') \in \mathcal{S}_h \times \mathcal{A}} \mathbb{I}_t(s', a') \geq \frac{-H}{q^{\pi_t}(s, a)},
\end{aligned}
$$

where the third line uses $\ell_t(s', a') - m_t(s', a') \geq -1$, the fourth line uses that $\frac{q^{\pi_t}(s,a) q^{\pi_t}(s',a'|s,a)}{q^{\pi_t}(s',a')} \leq 1$, and the last inequality follows from $\sum_{(s',a') \in \mathcal{S}_h \times \mathcal{A}} \mathbb{I}_t(s', a') \leq 1$ for each $h$.

Next, we evaluate the second term of (33). By the similar argument as above, we have

$$
\begin{aligned}
\sum_{b \neq a} \pi_t(b \mid s) Q^{\pi_t}(s, b; \widetilde{\ell}_t) &= \sum_{b \neq a} \pi_t(b \mid s) \sum_{h=h(s)}^{H-1} \sum_{(s',a') \in \mathcal{S}_h \times \mathcal{A}} q^{\pi_t}(s', a' \mid s, b) \widetilde{\ell}_t(s', a') \\
&= \sum_{h=h(s)}^{H-1} \sum_{(s',a') \in \mathcal{S}_h \times \mathcal{A}} \sum_{b \neq a} \pi_t(b \mid s) q^{\pi_t}(s', a' \mid s, b) \frac{\mathbb{I}_t(s', a')(\ell_t(s', a') - m_t(s', a'))}{q^{\pi_t}(s', a')} \\
&\leq \frac{1}{q^{\pi_t}(s)} \sum_{h=h(s)}^{H-1} \sum_{(s',a') \in \mathcal{S}_h \times \mathcal{A}} \sum_{b \neq a} \frac{q^{\pi_t}(s, b) q^{\pi_t}(s', a' \mid s, b)}{q^{\pi_t}(s', a')} \mathbb{I}_t(s', a') \\
&\leq \frac{1}{q^{\pi_t}(s)} \sum_{h=h(s)}^{H-1} \sum_{(s',a') \in \mathcal{S}_h \times \mathcal{A}} \mathbb{I}_t(s', a') \leq \frac{H}{q^{\pi_t}(s)}.
\end{aligned}
$$

Combining the two bounds yields

$$
\begin{aligned}
\widehat{\ell}_t(s, a) + g_t(s, a) - m_t(s, a) &\geq (1 - \pi_t(a \mid s)) \frac{-H}{q^{\pi_t}(s, a)} - \frac{H}{q^{\pi_t}(s)} \\
&= \frac{-H}{q^{\pi_t}(s, a)} ((1 - \pi_t(a \mid s)) + \pi_t(a \mid s)) = \frac{-H}{q^{\pi_t}(s, a)}.
\end{aligned}
$$

$\square$

**Lemma D.4.** *For an arbitrary loss prediction $m_t \in [0, 1]^{S \times A}$, it holds that*

$$
\mathbb{E}_t \left[ \left( \widehat{\ell}_t(s, a) + g_t(s, a) - m_t(s, a) \right)^2 \right] \leq \frac{2H^2}{q^{\pi_t}(s, a)} (1 - \pi_t(a \mid s))
$$

*for all state-action pairs $(s, a)$.*

*Proof.* Fix any $(s, a)$. By the definitions of $g_t$ and $\widetilde{\ell}_t$, we have

$$
\mathbb{E}_t\left[\left(\widehat{\ell}_t(s, a) + g_t(s, a) - m_t(s, a)\right)^2\right] = \mathbb{E}_t\left[\left(\widehat{\ell}_t(s, a) + Q^{\pi_t}(s, a; \widetilde{\ell}_t) - V^{\pi_t}(s; \widetilde{\ell}_t) - \widetilde{\ell}_t(s, a) - m_t(s, a)\right)^2\right]
$$

$$
= \mathbb{E}_t\left[\left(Q^{\pi_t}(s, a; \widetilde{\ell}_t) - V^{\pi_t}(s; \widetilde{\ell}_t)\right)^2\right]
$$

$$
= \mathbb{E}_t\left[\left((1 - \pi_t(a \mid s))Q^{\pi_t}(s, a; \widetilde{\ell}_t) - \sum_{b \neq a} \pi_t(b \mid s)Q^{\pi_t}(s, b; \widetilde{\ell}_t)\right)^2\right]
$$

$$
\leq 2\mathbb{E}_t\left[(1 - \pi_t(a \mid s))^2 Q^{\pi_t}(s, a; \widetilde{\ell}_t)^2 + \left(\sum_{b \neq a} \pi_t(b \mid s)Q^{\pi_t}(s, b; \widetilde{\ell}_t)\right)^2\right], \quad (34)
$$

where the last inequality follows from $(x - y)^2 \leq 2(x^2 + y^2)$.

By the definition of the $Q$-function, the first term in (34) is evaluated as

$$
\mathbb{E}_t\left[Q^{\pi_t}(s, a; \widetilde{\ell}_t)^2\right] = \mathbb{E}_t\left[\left(\sum_{h=h(s)}^{H-1} \sum_{(s', a') \in \mathcal{S}_h \times \mathcal{A}} q^{\pi_t}(s', a' \mid s, a)\widetilde{\ell}_t(s', a')\right)^2\right]
$$

$$
= \mathbb{E}_t\left[\left(\sum_{h=h(s)}^{H-1} \sum_{(s', a') \in \mathcal{S}_h \times \mathcal{A}} q^{\pi_t}(s', a' \mid s, a)\frac{\mathbb{I}_t(s', a')(\ell_t(s', a') - m_t(s', a'))}{q^{\pi_t}(s', a')}\right)^2\right]
$$

$$
\leq H\mathbb{E}_t\left[\sum_{h=h(s)}^{H-1} \sum_{(s', a') \in \mathcal{S}_h \times \mathcal{A}} q^{\pi_t}(s', a' \mid s, a)^2 \frac{\mathbb{I}_t(s', a')(\ell_t(s', a') - m_t(s', a'))^2}{q^{\pi_t}(s', a')^2}\right],
$$

where the last inequality applies the Cauchy–Schwarz inequality across at most $H$ stages combined with the fact that $\sum_{(s,a) \in \mathcal{S}_h \times \mathcal{A}} \mathbb{I}_t(s, a) \leq 1$. Then, this can be further bounded as

$$
H\mathbb{E}_t\left[\sum_{h=h(s)}^{H-1} \sum_{(s', a') \in \mathcal{S}_h \times \mathcal{A}} q^{\pi_t}(s', a' \mid s, a)^2 \frac{\mathbb{I}_t(s', a')(\ell_t(s', a') - m_t(s', a'))^2}{q^{\pi_t}(s', a')^2}\right]
$$

$$
\leq H \sum_{h=h(s)}^{H-1} \sum_{(s', a') \in \mathcal{S}_h \times \mathcal{A}} \frac{q^{\pi_t}(s', a' \mid s, a)^2}{q^{\pi_t}(s', a')}
$$

$$
\leq \frac{H}{q^{\pi_t}(s, a)} \sum_{h=h(s)}^{H-1} \sum_{(s', a') \in \mathcal{S}_h \times \mathcal{A}} \frac{q^{\pi_t}(s, a)q^{\pi_t}(s', a' \mid s, a)}{q^{\pi_t}(s', a')} \cdot q^{\pi_t}(s', a' \mid s, a)
$$

$$
\leq \frac{H}{q^{\pi_t}(s, a)} \sum_{h=h(s)}^{H-1} \sum_{(s', a') \in \mathcal{S}_h \times \mathcal{A}} q^{\pi_t}(s', a' \mid s, a) \leq \frac{H^2}{q^{\pi_t}(s, a)},
$$

where the last inequality follows from $\sum_{h=h(s)}^{H-1} \sum_{(s', a') \in \mathcal{S}_h \times \mathcal{A}} q^{\pi_t}(s', a' \mid s, a) \leq H$. Consequently, we obtain

$$
\mathbb{E}_t\left[Q^{\pi_t}(s, a; \widetilde{\ell}_t)^2\right] \leq \frac{H^2}{q^{\pi_t}(s, a)}. \quad (35)
$$

For the second term in (34), by repeating the similar arguments, we have

$$
\mathbb{E}_t\left[\left(\sum_{b\neq a}\pi_t(b\mid s)Q^{\pi_t}(s,b;\widetilde{\ell}_t)\right)^2\right]
$$

$$
=\mathbb{E}_t\left[\left(\sum_{h=h(s)}^{H-1}\sum_{(s',a')\in\mathcal{S}_h\times\mathcal{A}}\left(\sum_{b\neq a}\pi_t(b\mid s)q^{\pi_t}(s',a'\mid s,b)\right)\widetilde{\ell}_t(s',a')\right)^2\right]
$$

$$
=\mathbb{E}_t\left[\left(\sum_{h=h(s)}^{H-1}\sum_{(s',a')\in\mathcal{S}_h\times\mathcal{A}}\left(\sum_{b\neq a}\pi_t(b\mid s)q^{\pi_t}(s',a'\mid s,b)\right)\frac{\mathbb{I}_t(s',a')(\ell_t(s',a')-m_t(s',a'))}{q^{\pi_t}(s',a')}\right)^2\right]
$$

$$
\leq H\mathbb{E}_t\left[\sum_{h=h(s)}^{H-1}\sum_{(s',a')\in\mathcal{S}_h\times\mathcal{A}}\left(\sum_{b\neq a}\pi_t(b\mid s)q^{\pi_t}(s',a'\mid s,b)\right)^2\frac{\mathbb{I}_t(s',a')(\ell_t(s',a')-m_t(s',a'))^2}{q^{\pi_t}(s',a')^2}\right],
$$

where the last inequality applies the Cauchy–Schwarz inequality across at most $H$ stages, combined with the fact that $\sum_{(s,a)\in\mathcal{S}_h\times\mathcal{A}}\mathbb{I}_t(s,a)\leq 1$. This can be further bounded as

$$
H\mathbb{E}_t\left[\sum_{h=h(s)}^{H-1}\sum_{(s',a')\in\mathcal{S}_h\times\mathcal{A}}\left(\sum_{b\neq a}\pi_t(b\mid s)q^{\pi_t}(s',a'\mid s,b)\right)^2\frac{\mathbb{I}_t(s',a')(\ell_t(s',a')-m_t(s',a'))^2}{q^{\pi_t}(s',a')^2}\right]
$$

$$
\leq H\sum_{h=h(s)}^{H-1}\sum_{(s',a')\in\mathcal{S}_h\times\mathcal{A}}\frac{\left(\sum_{b\neq a}\pi_t(b\mid s)q^{\pi_t}(s',a'\mid s,b)\right)^2}{q^{\pi_t}(s',a')}
$$

$$
=\frac{H}{q^{\pi_t}(s)}\sum_{h=h(s)}^{H-1}\sum_{(s',a')\in\mathcal{S}_h\times\mathcal{A}}\frac{\sum_{c\neq a}q^{\pi_t}(s,c)q^{\pi_t}(s',a'\mid s,c)}{q^{\pi_t}(s',a')}\left(\sum_{b\neq a}\pi_t(b\mid s)q^{\pi_t}(s',a'\mid s,b)\right)
$$

$$
\leq\frac{H}{q^{\pi_t}(s)}\sum_{h=h(s)}^{H-1}\sum_{(s',a')\in\mathcal{S}_h\times\mathcal{A}}\sum_{b\neq a}\pi_t(b\mid s)q^{\pi_t}(s',a'\mid s,b)
$$

$$
=\frac{H}{q^{\pi_t}(s)}\sum_{b\neq a}\pi_t(b\mid s)\sum_{h=h(s)}^{H-1}\sum_{(s',a')\in\mathcal{S}_h\times\mathcal{A}}q^{\pi_t}(s',a'\mid s,b)
$$

$$
\leq\frac{H^2}{q^{\pi_t}(s)}\sum_{b\neq a}\pi_t(b\mid s)=\frac{H^2}{q^{\pi_t}(s)}(1-\pi_t(a\mid s)).
$$

Consequently, we obtain

$$
\mathbb{E}_t\left[\left(\sum_{b\neq a}\pi_t(b\mid s)Q^{\pi_t}(s,b;\widetilde{\ell}_t)\right)^2\right]\leq\frac{H^2}{q^{\pi_t}(s)}(1-\pi_t(a\mid s)). \tag{36}
$$

Finally, combining (34)–(36), we have

$$
\mathbb{E}_t\left[\left(\widehat{\ell}_t(s,a)+g_t(s,a)-m_t(s,a)\right)^2\right]\leq 2\left((1-\pi_t(a\mid s))^2\frac{H^2}{q^{\pi_t}(s,a)}+\frac{H^2}{q^{\pi_t}(s)}(1-\pi_t(a\mid s))\right)
$$

$$
\leq\frac{2H^2}{q^{\pi_t}(s,a)}\left((1-\pi_t(a\mid s))^2+\pi_t(a\mid s)(1-\pi_t(a\mid s))\right)
$$

$$
=\frac{2H^2}{q^{\pi_t}(s,a)}(1-\pi_t(a\mid s)),
$$

which is the desired bound. □

We next extend Lemma D.4 to derive a variance-aware upper bound. The additional terms that arise can be controlled, and become $\mathrm{polylog}(T)$ when $m_t$ is chosen as in (3) (see Lemma F.8).

**Lemma D.5.** *Under the stochastic regime with adversarial corruption, for an arbitrary loss prediction $m_t \in [0, 1]^{S \times A}$, it holds that*

$$
\mathbb{E}_t\left[\left(\widehat{\ell}_t(s, a) + g_t(s, a) - m_t(s, a)\right)^2\right]
$$
$$
\leq \frac{4H\mathbb{V}^c(s)}{q^{\pi_t}(s, a)}(1 - \pi_t(a \mid s)) + \frac{4H}{q^{\pi_t}(s, a)^2}\sum_{s', a'} q^{\pi_t}(s', a')\mathbb{E}_t\left[(\ell_t(s', a') - \ell'_t(s', a') + \mu(s', a') - m_t(s', a'))^2\right]
$$

*for all state-action pairs $(s, a)$, where we recall that $\mathbb{V}^c(s)$ is defined in (8).*

*Proof.* Fix any $(s, a)$. Define $\kappa_t(s, a) = \ell'_t(s, a) - \mu(s, a)$ and $\lambda_t(s, a) = \ell_t(s, a) - \ell'_t(s, a) + \mu(s, a) - m_t(s, a)$ so that

$$
\ell_t(s, a) - m_t(s, a) = \kappa_t(s, a) + \lambda_t(s, a).
$$

By the definitions of $g_t$ and $\widetilde{\ell}_t$, we have

$$
\mathbb{E}_t\left[\left(\widehat{\ell}_t(s, a) + g_t(s, a) - m_t(s, a)\right)^2\right] = \mathbb{E}_t\left[\left(\widehat{\ell}_t(s, a) + Q^{\pi_t}(s, a; \widetilde{\ell}_t) - V^{\pi_t}(s; \widetilde{\ell}_t) - \widetilde{\ell}_t(s, a) - m_t(s, a)\right)^2\right]
$$
$$
= \mathbb{E}_t\left[\left(Q^{\pi_t}(s, a; \widetilde{\ell}_t) - V^{\pi_t}(s; \widetilde{\ell}_t)\right)^2\right]
$$
$$
= \mathbb{E}_t\left[\left((1 - \pi_t(a \mid s))Q^{\pi_t}(s, a; \widetilde{\ell}_t) - \sum_{b \neq a} \pi_t(b \mid s)Q^{\pi_t}(s, b; \widetilde{\ell}_t)\right)^2\right]
$$
$$
\leq 2\mathbb{E}_t\left[(1 - \pi_t(a \mid s))^2 Q^{\pi_t}(s, a; \widetilde{\ell}_t)^2 + \left(\sum_{b \neq a} \pi_t(b \mid s)Q^{\pi_t}(s, b; \widetilde{\ell}_t)\right)^2\right], \quad (37)
$$

where the last inequality follows from $(x - y)^2 \leq 2(x^2 + y^2)$.

Then, we bound the two expectations on the right-hand side of (37) in turn.

**Bounding $\mathbb{E}_t\left[Q^{\pi_t}(s, a; \widetilde{\ell}_t)^2\right]$ (the first term in (37)).** Using the definition of the $Q$-function, we obtain

$$
\mathbb{E}_t\left[Q^{\pi_t}(s, a; \widetilde{\ell}_t)^2\right]
$$
$$
= \mathbb{E}_t\left[\left(\sum_{s', a'} q^{\pi_t}(s', a' \mid s, a)\widetilde{\ell}_t(s', a')\right)^2\right]
$$
$$
= \mathbb{E}_t\left[\left(\sum_{s', a'} q^{\pi_t}(s', a' \mid s, a)\frac{\mathbb{I}_t(s', a')(\ell_t(s', a') - m_t(s', a'))}{q^{\pi_t}(s', a')}\right)^2\right]
$$
$$
= \mathbb{E}_t\left[\left(\sum_{s', a'} \frac{q^{\pi_t}(s', a' \mid s, a)}{q^{\pi_t}(s', a')}\mathbb{I}_t(s', a')(\kappa_t(s', a') + \lambda_t(s', a'))\right)^2\right]
$$
$$
\leq 2\mathbb{E}_t\left[\left(\sum_{s', a'} \frac{q^{\pi_t}(s', a' \mid s, a)}{q^{\pi_t}(s', a')}\mathbb{I}_t(s', a')\kappa_t(s', a')\right)^2\right] + 2\mathbb{E}_t\left[\left(\sum_{s', a'} \frac{q^{\pi_t}(s', a' \mid s, a)}{q^{\pi_t}(s', a')}\mathbb{I}_t(s', a')\lambda_t(s', a')\right)^2\right], \quad (38)
$$

where the last inequality follows from $(x + y)^2 \leq 2(x^2 + y^2)$.

For the first term ($\kappa$-term) of (38), by using $\mathbb{E}_t\big[\kappa_t(s, a)^2\big] = \sigma^2(s, a)$, we have

$$
2\mathbb{E}_t\left[\left(\sum_{s',a'} \frac{q^{\pi_t}(s', a' \mid s, a)}{q^{\pi_t}(s', a')}\mathbb{I}_t(s', a')\kappa_t(s', a')\right)^2\right] \leq 2H\mathbb{E}_t\left[\sum_{s',a'} \frac{q^{\pi_t}(s', a' \mid s, a)^2}{q^{\pi_t}(s', a')^2}\mathbb{I}_t(s', a')\kappa_t(s', a')^2\right]
$$

$$
= 2H\sum_{s',a'} \frac{q^{\pi_t}(s', a' \mid s, a)^2}{q^{\pi_t}(s', a')}\sigma^2(s', a')
$$

$$
\leq \frac{2H}{q^{\pi_t}(s, a)}\sum_{s',a'} \frac{q^{\pi_t}(s, a)q^{\pi_t}(s', a' \mid s, a)}{q^{\pi_t}(s', a')}q^{\pi_t}(s', a' \mid s, a)\sigma^2(s', a')
$$

$$
\leq \frac{2H}{q^{\pi_t}(s, a)}\sum_{s',a'} q^{\pi_t}(s', a' \mid s, a)\sigma^2(s', a')
$$

$$
\leq \frac{2H\mathbb{V}^c(s)}{q^{\pi_t}(s, a)}, \tag{39}
$$

where the first inequality applies the Cauchy–Schwarz inequality across at most $H$ stages, combined with the fact that $\sum_{(s,a) \in \mathcal{S}_h \times \mathcal{A}} \mathbb{I}_t(s, a) \leq 1$.

Similarly, for the second term (the $\lambda$-term) of (38), we have

$$
2\mathbb{E}_t\left[\left(\sum_{s',a'} \frac{q^{\pi_t}(s', a' \mid s, a)}{q^{\pi_t}(s', a')}\mathbb{I}_t(s', a')\lambda_t(s', a')\right)^2\right]
$$

$$
\leq 2H\mathbb{E}_t\left[\sum_{s',a'} \frac{q^{\pi_t}(s', a' \mid s, a)^2}{q^{\pi_t}(s', a')^2}\mathbb{I}_t(s', a')\lambda_t(s', a')^2\right]
$$

$$
= 2H\sum_{s',a'} \frac{q^{\pi_t}(s', a' \mid s, a)^2}{q^{\pi_t}(s', a')}\mathbb{E}_t\big[\lambda_t(s', a')^2\big]
$$

$$
= \frac{2H}{q^{\pi_t}(s, a)}\sum_{s',a'} \frac{q^{\pi_t}(s, a)q^{\pi_t}(s', a' \mid s, a)}{q^{\pi_t}(s', a')}q^{\pi_t}(s', a' \mid s, a)\mathbb{E}_t\big[\lambda_t(s', a')^2\big]
$$

$$
\leq \frac{2H}{q^{\pi_t}(s, a)}\sum_{s',a'} q^{\pi_t}(s', a' \mid s, a)\mathbb{E}_t\big[\lambda_t(s', a')^2\big]
$$

$$
\leq \frac{2H}{q^{\pi_t}(s, a)^2}\sum_{s',a'} q^{\pi_t}(s', a')\mathbb{E}_t\big[\lambda_t(s', a')^2\big], \tag{40}
$$

where the first inequality applies the Cauchy–Schwarz inequality, and the last inequality follows from $q^{\pi_t}(s', a' \mid s, a) \leq \frac{q^{\pi_t}(s', a')}{q^{\pi_t}(s, a)}$.

Consequently, combining (38)–(40) yields

$$
\mathbb{E}_t\big[Q^{\pi_t}(s, a; \widetilde{\ell}_t)^2\big] \leq \frac{2H\mathbb{V}^c(s)}{q^{\pi_t}(s, a)} + \frac{2H}{q^{\pi_t}(s, a)^2}\sum_{s',a'} q^{\pi_t}(s', a')\mathbb{E}_t\big[\lambda_t(s', a')^2\big]. \tag{41}
$$

**Bounding $\mathbb{E}_t\left[\left(\sum_{b \neq a} \pi_t(b \mid s)Q^{\pi_t}(s, b; \widetilde{\ell}_t)\right)^2\right]$ (the second term in (37)).** By repeating the similar arguments,

$$
\mathbb{E}_t\left[\left(\sum_{b \neq a} \pi_t(b \mid s)Q^{\pi_t}(s, b; \widetilde{\ell}_t)\right)^2\right] = \mathbb{E}_t\left[\left(\sum_{s',a'}\left(\sum_{b \neq a} \pi_t(b \mid s)q^{\pi_t}(s', a' \mid s, b)\right)\widetilde{\ell}_t(s', a')\right)^2\right]
$$

$$= \mathbb{E}_t \left[ \left( \sum_{s',a'} \left( \sum_{b \neq a} \pi_t(b \mid s) q^{\pi_t}(s',a' \mid s,b) \right) \frac{\mathbb{I}_t(s',a')(\ell_t(s',a') - m_t(s',a'))}{q^{\pi_t}(s',a')} \right)^2 \right]$$

$$= \mathbb{E}_t \left[ \left( \sum_{s',a'} \frac{\sum_{b \neq a} \pi_t(b \mid s) q^{\pi_t}(s',a' \mid s,b)}{q^{\pi_t}(s',a')} \mathbb{I}_t(s',a')(\kappa_t(s',a') + \lambda_t(s',a')) \right)^2 \right]$$

$$\leq 2\mathbb{E}_t \left[ \left( \sum_{s',a'} \frac{\sum_{b \neq a} \pi_t(b \mid s) q^{\pi_t}(s',a' \mid s,b)}{q^{\pi_t}(s',a')} \mathbb{I}_t(s',a') \kappa_t(s',a') \right)^2 \right]$$

$$+ 2\mathbb{E}_t \left[ \left( \sum_{s',a'} \frac{\sum_{b \neq a} \pi_t(b \mid s) q^{\pi_t}(s',a' \mid s,b)}{q^{\pi_t}(s',a')} \mathbb{I}_t(s',a') \lambda_t(s',a') \right)^2 \right], \qquad (42)$$

For the first term (the $\kappa$-term) in (42), repeating the same argument as in (39), we obtain

$$2\mathbb{E}_t \left[ \left( \sum_{s',a'} \frac{\sum_{b \neq a} \pi_t(b \mid s) q^{\pi_t}(s',a' \mid s,b)}{q^{\pi_t}(s',a')} \mathbb{I}_t(s',a') \kappa_t(s',a') \right)^2 \right]$$

$$\leq 2H \mathbb{E}_t \left[ \sum_{s',a'} \frac{\left( \sum_{b \neq a} \pi_t(b \mid s) q^{\pi_t}(s',a' \mid s,b) \right)^2}{q^{\pi_t}(s',a')^2} \mathbb{I}_t(s',a') \kappa_t(s',a')^2 \right]$$

$$= 2H \sum_{s',a'} \frac{\left( \sum_{b \neq a} \pi_t(b \mid s) q^{\pi_t}(s',a' \mid s,b) \right)^2}{q^{\pi_t}(s',a')} \sigma^2(s',a')$$

$$\leq \frac{2H}{q^{\pi_t}(s)} \sum_{s',a'} \frac{\sum_{b \neq a} q^{\pi_t}(s,b) q^{\pi_t}(s',a' \mid s,b)}{q^{\pi_t}(s',a')} \left( \sum_{b \neq a} \pi_t(b \mid s) q^{\pi_t}(s',a' \mid s,b) \right) \sigma^2(s',a')$$

$$\leq \frac{2H}{q^{\pi_t}(s)} \sum_{s',a'} \sum_{b \neq a} \pi_t(b \mid s) q^{\pi_t}(s',a' \mid s,b) \sigma^2(s',a')$$

$$\leq \frac{2H}{q^{\pi_t}(s)} \sum_{b \neq a} \pi_t(b \mid s) \sum_{s',a'} q^{\pi_t}(s',a' \mid s,b) \sigma^2(s',a')$$

$$\leq \frac{2H}{q^{\pi_t}(s)} \sum_{b \neq a} \pi_t(b \mid s) \mathbb{V}^c(s)$$

$$\leq \frac{2H \mathbb{V}^c(s)}{q^{\pi_t}(s)} (1 - \pi_t(a \mid s)). \qquad (43)$$

The second term (the $\lambda$-term) in (42) can be bounded similarly by

$$2\mathbb{E}_t \left[ \left( \sum_{s',a'} \frac{\sum_{b \neq a} \pi_t(b \mid s) q^{\pi_t}(s',a' \mid s,b)}{q^{\pi_t}(s',a')} \mathbb{I}_t(s',a') \lambda_t(s',a') \right)^2 \right]$$

$$\leq 2H \mathbb{E}_t \left[ \sum_{s',a'} \frac{\left( \sum_{b \neq a} \pi_t(b \mid s) q^{\pi_t}(s',a' \mid s,b) \right)^2}{q^{\pi_t}(s',a')^2} \mathbb{I}_t(s',a') \lambda_t(s',a')^2 \right]$$

$$= 2H \sum_{s',a'} \frac{\left( \sum_{b \neq a} \pi_t(b \mid s) q^{\pi_t}(s',a' \mid s,b) \right)^2}{q^{\pi_t}(s',a')} \mathbb{E}_t \left[ \lambda_t(s',a')^2 \right]$$

$$
\begin{aligned}
&= \frac{2H}{q^{\pi_t}(s)} \sum_{s',a'} \frac{\sum_{c \neq a} q^{\pi_t}(s,c) q^{\pi_t}(s',a' \mid s,c)}{q^{\pi_t}(s',a')} \left( \sum_{b \neq a} \pi_t(b \mid s) q^{\pi_t}(s',a' \mid s,b) \right) \mathbb{E}_t\big[\lambda_t(s',a')^2\big] \\
&\leq \frac{2H}{q^{\pi_t}(s)} \sum_{s',a'} \sum_{b \neq a} \pi_t(b \mid s) q^{\pi_t}(s',a' \mid s,b) \mathbb{E}_t\big[\lambda_t(s',a')^2\big] \\
&\leq \frac{2H}{q^{\pi_t}(s)^2} \sum_{s',a'} q^{\pi_t}(s',a') \mathbb{E}_t\big[\lambda_t(s',a')^2\big],
\end{aligned}
\tag{44}
$$

where the last inequality follows from $\sum_{b \neq a} \pi_t(b \mid s) q^{\pi_t}(s',a' \mid s,b) \leq \frac{q^{\pi_t}(s',a')}{q^{\pi_t}(s)}$.

Consequently, combining (42)–(44) yields

$$
\mathbb{E}_t\left[ \left( \sum_{b \neq a} \pi_t(b \mid s) Q^{\pi_t}(s,b;\widetilde{\ell}_t) \right)^2 \right] \leq \frac{2H\mathbb{V}^c(s)}{q^{\pi_t}(s)}(1 - \pi_t(a \mid s)) + \frac{2H}{q^{\pi_t}(s)^2} \sum_{s',a'} q^{\pi_t}(s',a') \mathbb{E}_t\big[\lambda_t(s',a')^2\big].
\tag{45}
$$

Therefore, by (37), (41) and (45),

$$
\begin{aligned}
&\mathbb{E}_t\left[ \left( \widehat{\ell}_t(s,a) + g_t(s,a) - m_t(s,a) \right)^2 \right] \\
&\leq \frac{4H\mathbb{V}^c(s)}{q^{\pi_t}(s,a)}(1 - \pi_t(a \mid s))^2 + \frac{4H(1 - \pi_t(a \mid s))^2}{q^{\pi_t}(s,a)^2} \sum_{s',a'} q^{\pi_t}(s',a') \mathbb{E}_t\big[\lambda_t(s',a')^2\big] \\
&\quad + \frac{4H\mathbb{V}^c(s)}{q^{\pi_t}(s)}(1 - \pi_t(a \mid s)) + \frac{4H}{q^{\pi_t}(s)^2} \sum_{s',a'} q^{\pi_t}(s',a') \mathbb{E}_t\big[\lambda_t(s',a')^2\big] \\
&= \frac{4H\mathbb{V}^c(s)}{q^{\pi_t}(s,a)}(1 - \pi_t(a \mid s))(1 - \pi_t(a \mid s) + \pi_t(a \mid s)) \\
&\quad + \frac{4H((1 - \pi_t(a \mid s))^2 + \pi_t(a \mid s)^2)}{q^{\pi_t}(s,a)^2} \sum_{s',a'} q^{\pi_t}(s',a') \mathbb{E}_t\big[\lambda_t(s',a')^2\big] \\
&\leq \frac{4H\mathbb{V}^c(s)}{q^{\pi_t}(s,a)}(1 - \pi_t(a \mid s)) + \frac{4H}{q^{\pi_t}(s,a)^2} \sum_{s',a'} q^{\pi_t}(s',a') \mathbb{E}_t\big[\lambda_t(s',a')^2\big],
\end{aligned}
$$

and this completes the proof. $\qquad\square$

**Corollary D.6.** *In the stochastic regime with adversarial corruption, suppose that the uncorrupted losses are generated independently and are uncorrelated across layers. Then, it holds that*

$$
\begin{aligned}
&\mathbb{E}_t\left[ \left( \widehat{\ell}_t(s,a) + g_t(s,a) - m_t(s,a) \right)^2 \right] \\
&\leq \frac{4\mathbb{V}^c(s)}{q^{\pi_t}(s,a)}(1 - \pi_t(a \mid s)) + \frac{4H}{q^{\pi_t}(s,a)^2} \sum_{s',a'} q^{\pi_t}(s',a') \mathbb{E}_t\big[ (\ell_t(s',a') - \ell'_t(s',a') + \mu(s',a') - m_t(s',a'))^2 \big]
\end{aligned}
$$

*for all state-action pairs $(s,a)$.*

*Proof.* This corollary can be viewed as a simple variant of Lemma D.5. Let $\kappa_t(s,a) := \ell'_t(s,a) - \mu(s,a)$. Since the uncorrupted losses are generated independently and are uncorrelated across layers, it holds that for any $(s_1,a_1) \neq (s_2,a_2)$,

$$
\mathbb{E}_t[\mathbb{I}_t(s_1,a_1)\mathbb{I}_t(s_2,a_2)\kappa_t(s_1,a_1)\kappa_t(s_2,a_2)] = 0.
$$

Then, for any function $\alpha : \mathcal{S} \times \mathcal{A} \to \mathbb{R}$, we have

$$\left( \sum_{s,a} \alpha(s,a) \mathbb{I}_t(s,a) \kappa_t(s,a) \right)^2 = \sum_{s_1,a_1} \sum_{s_2,a_2} \alpha(s_1,a_1) \alpha(s_2,a_2) \mathbb{I}_t(s_1,a_1) \mathbb{I}_t(s_2,a_2) \kappa_t(s_1,a_1) \kappa_t(s_2,a_2)$$

$$= \sum_{s,a} \alpha(s,a)^2 \mathbb{I}_t(s,a) \kappa_t(s,a)^2. \tag{46}$$

Thus, for the first term ($\kappa$-term) of (38), we have

$$2\mathbb{E}_t \left[ \left( \sum_{s',a'} \frac{q^{\pi_t}(s',a' \mid s,a)}{q^{\pi_t}(s',a')} \mathbb{I}_t(s',a') \kappa_t(s',a') \right)^2 \right] \leq 2\mathbb{E}_t \left[ \sum_{s',a'} \frac{q^{\pi_t}(s',a' \mid s,a)^2}{q^{\pi_t}(s',a')^2} \mathbb{I}_t(s',a') \kappa_t(s',a')^2 \right] \qquad \text{(by (46))}$$

$$= 2 \sum_{s',a'} \frac{q^{\pi_t}(s',a' \mid s,a)^2}{q^{\pi_t}(s',a')} \sigma^2(s',a')$$

$$\leq \frac{2}{q^{\pi_t}(s,a)} \sum_{s',a'} \frac{q^{\pi_t}(s,a) q^{\pi_t}(s',a' \mid s,a)}{q^{\pi_t}(s',a')} q^{\pi_t}(s',a' \mid s,a) \sigma^2(s',a')$$

$$\leq \frac{2}{q^{\pi_t}(s,a)} \sum_{s',a'} q^{\pi_t}(s',a' \mid s,a) \sigma^2(s',a')$$

$$\leq \frac{2\mathbb{V}^c(s)}{q^{\pi_t}(s,a)}. \tag{47}$$

For the first term (the $\kappa$-term) in (42), repeating the same argument as in (47), we also obtain

$$2\mathbb{E}_t \left[ \left( \sum_{s',a'} \frac{\sum_{b \neq a} \pi_t(b \mid s) q^{\pi_t}(s',a' \mid s,b)}{q^{\pi_t}(s',a')} \mathbb{I}_t(s',a') \kappa_t(s',a') \right)^2 \right]$$

$$\leq 2\mathbb{E}_t \left[ \sum_{s',a'} \frac{\left( \sum_{b \neq a} \pi_t(b \mid s) q^{\pi_t}(s',a' \mid s,b) \right)^2}{q^{\pi_t}(s',a')^2} \mathbb{I}_t(s',a') \kappa_t(s',a')^2 \right] \qquad \text{(by (46))}$$

$$\leq \frac{2\mathbb{V}^c(s)}{q^{\pi_t}(s)}(1 - \pi_t(a \mid s)). \tag{48}$$

Therefore, compared with Lemma D.5, we obtain an $H$-times sharper bound in (47) and (48) than (39) and (43). As a consequence, the corresponding $\mathbb{V}^c$ term is also improved by a factor of $H$. $\qquad \square$

**Lemma D.7.** *Suppose that the learning rates are updated according to* (11). *Then, it holds that*

$$\eta_t(s,a) \leq \frac{\sqrt{\log(T)}}{\sqrt{2H^2 \log(T) + \sum_{\tau=1}^t \zeta_\tau(s,a)}}$$

*for any episode $t$ and state-action pair $(s,a)$.*

*Proof.* By the update rule of the learning rate (11),

$$\frac{1}{\eta_{t+1}(s,a)^2} = \left( \frac{1}{\eta_t(s,a)} + \frac{\eta_t(s,a)}{\log(T)} \zeta_t(s,a) \right)^2$$

$$\geq \frac{1}{\eta_t(s,a)^2} + \frac{2}{\log(T)} \zeta_t(s,a).$$

Repeatedly applying the above inequality yields

$$\frac{1}{\eta_t(s,a)^2} \geq \frac{1}{\eta_1^2} + \sum_{\tau=1}^{t-1} \frac{2}{\log(T)} \zeta_\tau(s,a).$$

Taking reciprocals and then taking square roots yields

$$\eta_t(s,a) \leq \frac{1}{\sqrt{\eta_1^{-2} + \sum_{\tau=1}^{t-1} \frac{2}{\log(T)} \zeta_\tau(s,a)}}$$

$$\leq \frac{\sqrt{\log(T)}}{\sqrt{2}\sqrt{\frac{1}{2}\eta_1^{-2}\log(T) + \sum_{\tau=1}^{t-1} \zeta_\tau(s,a)}}$$

$$\leq \frac{\sqrt{\log(T)}}{\sqrt{\frac{1}{2}\eta_1^{-2}\log(T) + \sum_{\tau=1}^{t} \zeta_\tau(s,a)}},$$

where the last inequality follows from $\zeta_t(s,a) \leq 1 \leq \frac{1}{2}\eta_1^{-2}\log(T) = 2H^2\log(T)$ for $T \geq 2$. Finally, using $\frac{1}{2}\eta_1^{-2}\log(T) = 2H^2\log(T)$, we obtain

$$\eta_t(s,a) \leq \frac{\sqrt{\log(T)}}{\sqrt{2H^2\log(T) + \sum_{\tau=1}^{t} \zeta_\tau(s,a)}}.$$

$\square$

## D.2. Common Regret Analysis

**Lemma D.8.** *Algorithm 1 guarantees*

$$\mathrm{Reg}_T \lesssim HSA\log(T) + \sum_{s,a} \sqrt{\log(T)\mathbb{E}\left[\sum_{t=1}^{T} \zeta_t(s,a)\right]},$$

*where* $\zeta_t(s,a) = q^{\pi_t}(s,a)^2 \min\left\{(\widehat{\ell}_t(s,a) - m_t(s,a))^2, (\widehat{\ell}_t(s,a) + g_t(s,a) - m_t(s,a))^2\right\}.$

*Proof.* From (32), we have $\mathrm{Reg}_T = \mathbb{E}\left[\sum_{t=1}^{T}\langle q^{\pi_t} - q^{\mathring{\pi}}, \widehat{\ell}_t\rangle\right]$, and we will apply Lemma C.3 with $p_t = q^{\pi_t}$ and $\ell_t \in \{\widehat{\ell}_t, \widehat{\ell}_t + g_t\}$ combined with Lemma D.2. To do so, we will check the conditions of Lemma C.3. For any $(s,a)$, we have

$$\eta_t(s,a)q^{\pi_t}(s,a)(\widehat{\ell}_t(s,a) - m_t(s,a)) = \eta_t(s,a)q^{\pi_t}(s,a)\frac{\mathbb{I}_t(s,a)(\ell_t(s,a) - m_t(s,a))}{q^{\pi_t}(s,a)}$$

$$\geq -\eta_t(s,a) \qquad\qquad (\text{by } \ell_t(s,a) - m_t(s,a) \geq -1)$$

$$\geq -\eta_1 \geq -\frac{1}{2}, \qquad\qquad (\text{by } \eta_1 = \tfrac{1}{2H})$$

and

$$\eta_t(s,a)q^{\pi_t}(s,a)(\widehat{\ell}_t(s,a) + g_t(s,a) - m_t(s,a)) \geq \eta_t(s,a)q^{\pi_t}(s,a)\frac{-H}{q^{\pi_t}(s,a)} \qquad (\text{by Lemma D.3})$$

$$= -H\eta_t(s,a)$$

$$\geq -H\eta_1 = -\frac{1}{2}. \qquad\qquad (\text{by } \eta_1 = \tfrac{1}{2H})$$

Moreover, define

$$\widetilde{q} = \frac{1}{SA}\sum_{s,a} q_{s,a}^{\max} \in \Omega(P),$$

where $q_{s,a}^{\max}$ denotes the occupancy measure induced by a policy that maximizes the probability of visiting the state-action pair $(s, a)$ under transition kernel $P$.

Therefore, by Lemmas C.3 and D.2, we obtain

$$\mathbb{E}\left[\sum_{t=1}^{T}\left\langle q^{\pi_t} - q^{\mathring{\pi}}, \widehat{\ell}_t \right\rangle\right]$$

$$\leq \frac{SA\log(SAT)}{\eta_1} + \mathbb{E}\left[\sum_{t=1}^{T}\sum_{s,a}\left(\frac{1}{\eta_{t+1}(s,a)} - \frac{1}{\eta_t(s,a)}\right)\log(SAT)\right]$$

$$+ \mathbb{E}\left[\sum_{t=1}^{T}\sum_{s,a}\eta_t(s,a)q^{\pi_t}(s,a)^2 \min\left\{(\widehat{\ell}_t(s,a) - m_t(s,a))^2, (\widehat{\ell}_t(s,a) + g_t(s,a) - m_t(s,a))^2\right\}\right]$$

$$+ \mathbb{E}\left[\frac{1}{T}\sum_{t=1}^{T}\left\langle -q^{\mathring{\pi}} + \widetilde{q}, \widehat{\ell}_t\right\rangle\right] + 2H\mathbb{E}[\|m_{T+1}\|_\infty]$$

$$\leq \frac{3SA\log(T)}{\eta_1} + 2H + 2H + 3\mathbb{E}\left[\sum_{t=1}^{T}\sum_{s,a}\left(\frac{1}{\eta_{t+1}(s,a)} - \frac{1}{\eta_t(s,a)}\right)\log(T)\right]$$

$$\text{(by } T \geq S, T \geq A, \text{ and } \|m_{T+1}\|_\infty \leq 1)$$

$$+ \mathbb{E}\left[\sum_{t=1}^{T}\sum_{s,a}\eta_t(s,a)\zeta_t(s,a)\right]$$

$$\leq \frac{3SA\log(T)}{\eta_1} + 4H + 4\mathbb{E}\left[\sum_{t=1}^{T}\sum_{s,a}\eta_t(s,a)\zeta_t(s,a)\right].$$

Here, the second inequality follows from

$$\mathbb{E}\left[\frac{1}{T}\sum_{t=1}^{T}\left\langle -q^{\mathring{\pi}} + \widetilde{q}, \widehat{\ell}_t\right\rangle\right] = \frac{1}{T}\left\langle -q^{\mathring{\pi}} + \widetilde{q}, \sum_{t=1}^{T}\ell_t\right\rangle \leq \frac{1}{T}\left\|-q^{\mathring{\pi}} + \widetilde{q}\right\|_1 \left\|\sum_{t=1}^{T}\ell_t\right\|_\infty \leq \frac{2HT}{T} = 2H,$$

where we used $\left\|-q^{\mathring{\pi}} + \widetilde{q}\right\|_1 \leq 2H$ and $\left\|\sum_{t=1}^{T}\ell_t\right\|_\infty \leq T$, and the last inequality follows from the update rule of the learning rate (11) and the definition of $\zeta_t(s, a)$.

It remains to bound $\sum_{t=1}^{T}\sum_{s,a}\eta_t(s,a)\zeta_t(s,a)$. From Lemma D.7,

$$\sum_{t=1}^{T}\sum_{s,a}\eta_t(s,a)\zeta_t(s,a)$$

$$\leq \sqrt{\log(T)}\sum_{t=1}^{T}\sum_{s,a}\frac{\zeta_t(s,a)}{\sqrt{2H^2\log(T) + \sum_{\tau=1}^{t}\zeta_\tau(s,a)}}$$

$$\leq 2\sqrt{\log(T)}\sum_{t=1}^{T}\sum_{s,a}\left(\sqrt{2H^2\log(T) + \sum_{\tau=1}^{t}\zeta_\tau(s,a)} - \sqrt{2H^2\log(T) + \sum_{\tau=1}^{t-1}\zeta_\tau(s,a)}\right)$$

$$= 2\sqrt{\log(T)}\sum_{s,a}\left(\sqrt{2H^2\log(T) + \sum_{\tau=1}^{T}\zeta_\tau(s,a)} - \sqrt{2H^2\log(T)}\right)$$

$$\leq 2\sqrt{\log(T)}\sum_{s,a}\sqrt{\sum_{t=1}^{T}\zeta_t(s,a)},$$

where the second inequality follows from

$$2\left(\sqrt{2H^2\log(T) + \sum_{\tau=1}^{t}\zeta_\tau(s,a)} - \sqrt{2H^2\log(T) + \sum_{\tau=1}^{t-1}\zeta_\tau(s,a)}\right)$$

$$= \frac{2\zeta_t(s,a)}{\sqrt{2H^2\log(T) + \sum_{\tau=1}^{t}\zeta_\tau(s,a)} + \sqrt{2H^2\log(T) + \sum_{\tau=1}^{t-1}\zeta_\tau(s,a)}}$$

$$\geq \frac{\zeta_t(s,a)}{\sqrt{2H^2\log(T) + \sum_{\tau=1}^{t}\zeta_\tau(s,a)}}.$$

Therefore, combining the above argument with (32), we obtain

$$\mathrm{Reg}_T = \mathbb{E}\left[\sum_{t=1}^{T}\left\langle q^{\pi_t} - q^{\mathring{\pi}}, \widehat{\ell}_t\right\rangle\right]$$

$$\leq \frac{3SA\log(T)}{\eta_1} + 4H + 8\sum_{s,a}\sqrt{\log(T)\mathbb{E}\left[\sum_{t=1}^{T}\zeta_t(s,a)\right]}$$

$$\lesssim HSA\log(T) + \sum_{s,a}\sqrt{\log(T)\mathbb{E}\left[\sum_{t=1}^{T}\zeta_t(s,a)\right]},$$

which completes the proof. □

### D.3. Proof of Theorem 4.1

Now we are ready to prove Theorem 4.1.

**Theorem D.9** (Restatement of Theorem 4.1). *Algorithm 1 with the loss prediction $m_t$ defined in (2) guarantees*

$$\mathrm{Reg}_T \lesssim \sqrt{SA\log(T)\min\{L^\star, HT - L^\star, Q_\infty, V_1\}} + HSA\log(T).$$

*Under the stochastic regime with adversarial corruption, it simultaneously ensures*

$$\mathrm{Reg}_T \lesssim \sqrt{SA\log(T)(\mathbb{V}T + \mathcal{C})} + HSA\log(T),$$

*and*

$$\mathrm{Reg}_T \lesssim U + \sqrt{U\mathcal{C}} + HSA\log(T),$$

*where $U = \sum_s \sum_{a\neq\pi^\star(s)} \frac{H^2\log(T)}{\Delta(s,a)}$.*

*Proof.* We start from Lemma D.8, which gives

$$\mathrm{Reg}_T \lesssim HSA\log(T) + \sum_{s,a}\sqrt{\log(T)\mathbb{E}\left[\sum_{t=1}^{T}\zeta_t(s,a)\right]}. \tag{49}$$

By the definition of $\zeta_t(s, a)$, we have

$$\sum_{s,a} \sqrt{\log(T)\mathbb{E}\left[\sum_{t=1}^{T} \zeta_t(s, a)\right]}$$

$$= \sum_{s,a} \sqrt{\log(T)\mathbb{E}\left[\sum_{t=1}^{T} q^{\pi_t}(s, a)^2 \min\left\{(\widehat{\ell}_t(s, a) - m_t(s, a))^2, (\widehat{\ell}_t(s, a) + g_t(s, a) - m_t(s, a))^2\right\}\right]}$$

$$\leq \sqrt{SA\log(T)\mathbb{E}\left[\sum_{t=1}^{T}\sum_{s,a} q^{\pi_t}(s, a)^2(\widehat{\ell}_t(s, a) - m_t(s, a))^2\right]} \qquad \text{(by the Cauchy–Schwarz inequality)}$$

$$= \sqrt{SA\log(T)\mathbb{E}\left[\sum_{t=1}^{T}\sum_{s,a} \mathbb{I}_t(s, a)(\ell_t(s, a) - m_t(s, a))^2\right]}, \qquad (50)$$

where the last equality uses $(\widehat{\ell}_t(s, a) - m_t(s, a))^2 = \frac{\mathbb{I}_t(s,a)(\ell_t(s,a)-m_t(s,a))^2}{q^{\pi_t}(s,a)^2}$.

**1. Bounds for the adversarial regime.** By Lemma F.12, we can evaluate (50) as

$$\sqrt{SA\log(T)\mathbb{E}\left[\sum_{t=1}^{T}\sum_{s,a} \mathbb{I}_t(s, a)(\ell_t(s, a) - m_t(s, a))^2\right]}$$

$$\lesssim \sqrt{SA\log(T)(\min\{L^\star + \mathrm{Reg}_T, HT - L^\star - \mathrm{Reg}_T, Q_\infty, V_1\} + SA)}$$

$$\leq \sqrt{SA\log(T)\min\{L^\star + \mathrm{Reg}_T, HT - L^\star - \mathrm{Reg}_T, Q_\infty, V_1\}} + SA\sqrt{\log(T)}.$$

Absorbing the lower-order term into $HSA\log(T)$, we obtain

$$\mathrm{Reg}_T \lesssim \sqrt{SA\log(T)(L^\star + \mathrm{Reg}_T)} + HSA\log(T), \qquad (51)$$

$$\mathrm{Reg}_T \lesssim \sqrt{SA\log(T)(HT - L^\star - \mathrm{Reg}_T)} + HSA\log(T), \qquad (52)$$

$$\mathrm{Reg}_T \lesssim \sqrt{SA\log(T)Q_\infty} + HSA\log(T), \qquad (53)$$

$$\mathrm{Reg}_T \lesssim \sqrt{SA\log(T)V_1} + HSA\log(T). \qquad (54)$$

From (51),

$$\mathrm{Reg}_T \leq c\sqrt{SA\log(T)L^\star} + c\sqrt{SA\log(T)\mathrm{Reg}_T} + cHSA\log(T) \qquad \text{(for some absolute constant } c\text{)}$$

$$\leq c\sqrt{SA\log(T)L^\star} + \frac{c^2}{2}SA\log(T) + \frac{1}{2}\mathrm{Reg}_T + cHSA\log(T)$$

$$\leq \frac{1}{2}\mathrm{Reg}_T + O(\sqrt{SA\log(T)L^\star} + HSA\log(T)),$$

where the second line follows from the AM–GM inequality. Therefore,

$$\mathrm{Reg}_T \lesssim \sqrt{SA\log(T)L^\star} + HSA\log(T). \qquad (55)$$

From (52), we also have

$$\mathrm{Reg}_T \lesssim \sqrt{SA\log(T)(HT - L^\star - \mathrm{Reg}_T)} + HSA\log(T)$$

$$\leq \sqrt{SA\log(T)(HT - L^\star)} + HSA\log(T) \qquad (56)$$

Combining (53)–(56), we obtain

$$\mathrm{Reg}_T \lesssim \sqrt{SA\log(T)\min\{L^\star, HT - L^\star, Q_\infty, V_1\}} + HSA\log(T)$$

**2. Stochastic variance bound.** In the stochastic regime, combining (49) and (50) with Lemma F.12 implies

$$\text{Reg}_T \lesssim \sqrt{SA \log(T)(\mathbb{V}T + \mathcal{C})} + HSA \log(T).$$

**3. Stochastic gap-dependent bound.** We can evaluate (50) as

$$\sum_{s,a} \sqrt{\log(T)\mathbb{E}\left[\sum_{t=1}^{T} \zeta_t(s,a)\right]}$$

$$= \sum_{s,a} \sqrt{\log(T)\mathbb{E}\left[\sum_{t=1}^{T} q^{\pi_t}(s,a)^2 \min\left\{(\widehat{\ell}_t(s,a) - m_t(s,a))^2, (\widehat{\ell}_t(s,a) + g_t(s,a) - m_t(s,a))^2\right\}\right]}$$

$$\leq \sum_{s,a} \sqrt{\log(T)\mathbb{E}\left[\sum_{t=1}^{T} q^{\pi_t}(s,a)^2 (\widehat{\ell}_t(s,a) + g_t(s,a) - m_t(s,a))^2\right]}$$

$$\leq \sum_{s,a} \sqrt{\log(T)\mathbb{E}\left[\sum_{t=1}^{T} q^{\pi_t}(s,a)^2 \frac{2H^2}{q^{\pi_t}(s,a)}(1 - \pi_t(a \mid s))\right]} \qquad \text{(by Lemma D.4)}$$

$$= \sqrt{2}H \sum_{s,a} \sqrt{\log(T)\mathbb{E}\left[\sum_{t=1}^{T} q^{\pi_t}(s,a)(1 - \pi_t(a \mid s))\right]}$$

$$\leq \sqrt{2}H \sum_s \sum_{a \neq \pi^\star(s)} \sqrt{\log(T)\mathbb{E}\left[\sum_{t=1}^{T} q^{\pi_t}(s,a)\right]} + \sqrt{2}H \sum_s \sqrt{\log(T)\mathbb{E}\left[\sum_{t=1}^{T} q^{\pi_t}(s)(1 - \pi_t(\pi^\star(s) \mid s))\right]}$$

$$= \sqrt{2}H \sum_s \sum_{a \neq \pi^\star(s)} \sqrt{\log(T)\mathbb{E}\left[\sum_{t=1}^{T} q^{\pi_t}(s,a)\right]} + \sqrt{2}H \sum_s \sqrt{\log(T)\sum_{t=1}^{T} \mathbb{E}\left[\sum_{a \neq \pi^\star(s)} q^{\pi_t}(s)\pi_t(a \mid s)\right]}$$

$$\leq 2\sqrt{2}H \sum_s \sum_{a \neq \pi^\star(s)} \sqrt{\log(T)\mathbb{E}\left[\sum_{t=1}^{T} q^{\pi_t}(s,a)\right]}.$$

Hence, combining this with (49), we obtain

$$\text{Reg}_T \lesssim H \sum_s \sum_{a \neq \pi^\star(s)} \sqrt{\log(T)\mathbb{E}\left[\sum_{t=1}^{T} q^{\pi_t}(s,a)\right]} + HSA \log(T).$$

Finally, applying Lemma F.15 to the last inequality yields

$$\text{Reg}_T \lesssim U + \sqrt{U\mathcal{C}} + HSA \log(T),$$

where $U = \sum_s \sum_{a \neq \pi^\star(s)} \frac{H^2 \log(T)}{\Delta(s,a)}$. $\qquad \qquad \square$

### D.4. Proof of Theorem 4.2

Here we provide the proof of Theorem 4.2.

**Theorem D.10** (Restatement of Theorem 4.2). *Algorithm 1 with the loss prediction $m_t$ defined in (3) guarantees*

$$\text{Reg}_T \lesssim \sqrt{SA \log(T) \min\{L^\star, HT - L^\star, Q_\infty\}} + HSA \log(T).$$

*Under the stochastic regime with adversarial corruption, it simultaneously ensures*

$$\text{Reg}_T \lesssim \sqrt{SA \log(T)(\mathbb{V}T + \mathcal{C})} + HSA \log(T),$$

*and*

$$\text{Reg}_T \lesssim U_{\text{Var}} + \sqrt{U_{\text{Var}}\mathcal{C}} + \sqrt{HS^2A^2\mathcal{C}}\log(T) + H^{\frac{1}{2}}S^{\frac{3}{2}}A^{\frac{3}{2}}\log^{\frac{3}{2}}(T),$$

*where* $U_{\text{Var}} = \sum_s \sum_{a \neq \pi^\star(s)} \frac{H\mathbb{V}^c(s)\log(T)}{\Delta(s,a)}$.

*Proof.* The proof follows the same argument as Theorem 4.1. The main differences are that the stochastic gap-dependent bound becomes variance-aware, at the cost of not deriving a path-length bound.

We start from Lemma D.8, which gives

$$\text{Reg}_T \lesssim HSA\log(T) + \sum_{s,a}\sqrt{\log(T)\mathbb{E}\left[\sum_{t=1}^T \zeta_t(s,a)\right]}. \tag{57}$$

By the definition of $\zeta_t(s,a)$, the same argument as in (50) yields

$$\sum_{s,a}\sqrt{\log(T)\mathbb{E}\left[\sum_{t=1}^T \zeta_t(s,a)\right]} \leq \sqrt{SA\log(T)\mathbb{E}\left[\sum_{t=1}^T\sum_{s,a}\mathbb{I}_t(s,a)(\ell_t(s,a) - m_t(s,a))^2\right]}. \tag{58}$$

**1. Bounds for the adversarial regime.** Applying Lemma F.13 to (58) gives

$$\sqrt{SA\log(T)\mathbb{E}\left[\sum_{t=1}^T\sum_{s,a}\mathbb{I}_t(s,a)(\ell_t(s,a) - m_t(s,a))^2\right]}$$

$$\lesssim \sqrt{SA\log(T)(\min\{L^\star + \text{Reg}_T, HT - L^\star - \text{Reg}_T, Q_\infty\} + SA\log(T) + SA)}$$

$$\lesssim \sqrt{SA\log(T)\min\{L^\star + \text{Reg}_T, HT - L^\star - \text{Reg}_T, Q_\infty\}} + SA\log(T).$$

Absorbing the lower-order term into $HSA\log(T)$, we obtain

$$\text{Reg}_T \lesssim \sqrt{SA\log(T)(L^\star + \text{Reg}_T)} + HSA\log(T)$$

$$\text{Reg}_T \lesssim \sqrt{SA\log(T)(HT - L^\star - \text{Reg}_T)} + HSA\log(T)$$

$$\text{Reg}_T \lesssim \sqrt{SA\log(T)Q_\infty} + HSA\log(T)$$

Applying the same calculation as in (51)–(53) gives

$$\text{Reg}_T \lesssim \sqrt{SA\log(T)\min\{L^\star, HT - L^\star, Q_\infty\}} + HSA\log(T).$$

**2. Stochastic variance bound.** Under the stochastic regime, Lemma F.13 further implies

$$\text{Reg}_T \lesssim \sqrt{SA\log(T)(\mathbb{V}T + \mathcal{C})} + HSA\log(T).$$

**3. Stochastic gap-dependent bound.** Moreover, by using Lemmas D.5 and F.8, we have

$$\sum_{s,a} \sqrt{\log(T)\mathbb{E}\left[\sum_{t=1}^{T} \zeta_t(s,a)\right]}$$

$$= \sum_{s,a} \sqrt{\log(T)\mathbb{E}\left[\sum_{t=1}^{T} q^{\pi_t}(s,a)^2 \min\left\{(\widehat{\ell}_t(s,a) - m_t(s,a))^2, (\widehat{\ell}_t(s,a) + g_t(s,a) - m_t(s,a))^2\right\}\right]}$$

$$\leq \sum_{s,a} \sqrt{\log(T)\mathbb{E}\left[\sum_{t=1}^{T} q^{\pi_t}(s,a)^2 (\widehat{\ell}_t(s,a) + g_t(s,a) - m_t(s,a))^2\right]} \tag{59}$$

$$\leq \sum_{s,a} \sqrt{\log(T)\mathbb{E}\left[\sum_{t=1}^{T} 4H\mathbb{V}^c(s)q^{\pi_t}(s,a)(1 - \pi_t(a \mid s))\right]} \tag{60}$$

$$+ \sum_{s,a} \sqrt{\log(T)\mathbb{E}\left[4H \sum_{t=1}^{T} \sum_{s',a'} q^{\pi_t}(s',a')(\ell_t(s',a') - \ell'_t(s',a') + \mu(s',a') - m_t(s',a'))^2\right]} \quad \text{(by Lemma D.5)}$$

$$\lesssim \sum_{s,a} \sqrt{\log(T)\mathbb{E}\left[\sum_{t=1}^{T} H\mathbb{V}^c(s)q^{\pi_t}(s,a)(1 - \pi_t(a \mid s))\right]} + \sum_{s,a} \sqrt{H \log(T)\big(SA \log^2(T) + \mathcal{C}\log(T)\big)} \quad \text{(by Lemma F.8)}$$

$$\lesssim \sum_{s,a} \sqrt{\log(T)\mathbb{E}\left[\sum_{t=1}^{T} H\mathbb{V}^c(s)q^{\pi_t}(s,a)(1 - \pi_t(a \mid s))\right]} + \sqrt{HS^2A^2\mathcal{C}}\log(T) + H^{\frac{1}{2}}S^{\frac{3}{2}}A^{\frac{3}{2}}\log^{\frac{3}{2}}(T). \tag{61}$$

Further, the first term in (61) can be rewritten as

$$\sum_{s,a} \sqrt{\log(T)\mathbb{E}\left[\sum_{t=1}^{T} H\mathbb{V}^c(s)q^{\pi_t}(s,a)(1 - \pi_t(a \mid s))\right]}$$

$$\leq \sum_{s} \sqrt{H\mathbb{V}^c(s)} \sum_{a \neq \pi^\star(s)} \sqrt{\log(T)\mathbb{E}\left[\sum_{t=1}^{T} q^{\pi_t}(s,a)\right]}$$

$$+ \sum_{s} \sqrt{H\mathbb{V}^c(s)} \sqrt{\log(T)\mathbb{E}\left[\sum_{t=1}^{T} q^{\pi_t}(s)(1 - \pi_t(\pi^\star(s) \mid s))\right]}$$

$$= \sum_{s} \sqrt{H\mathbb{V}^c(s)} \sum_{a \neq \pi^\star(s)} \sqrt{\log(T)\mathbb{E}\left[\sum_{t=1}^{T} q^{\pi_t}(s,a)\right]}$$

$$+ \sum_{s} \sqrt{H\mathbb{V}^c(s)} \sqrt{\log(T) \sum_{t=1}^{T} \mathbb{E}\left[\sum_{a \neq \pi^\star(s)} q^{\pi_t}(s)\pi_t(a \mid s)\right]}$$

$$\leq \sum_{s} 2\sqrt{H\mathbb{V}^c(s)} \sum_{a \neq \pi^\star(s)} \sqrt{\log(T)\mathbb{E}\left[\sum_{t=1}^{T} q^{\pi_t}(s,a)\right]}.$$

Hence, combining the last inequality with (57), we obtain

$$\mathrm{Reg}_T \lesssim \sum_{s} \sqrt{H\mathbb{V}^c(s)} \sum_{a \neq \pi^\star(s)} \sqrt{\log(T)\mathbb{E}\left[\sum_{t=1}^{T} q^{\pi_t}(s,a)\right]} + \sqrt{HS^2A^2\mathcal{C}}\log(T) + H^{\frac{1}{2}}S^{\frac{3}{2}}A^{\frac{3}{2}}\log^{\frac{3}{2}}(T).$$

Finally, applying Lemma F.15 to the last inequality yields

$$\text{Reg}_T \lesssim U_{\text{Var}} + \sqrt{U_{\text{Var}}C} + \sqrt{HS^2A^2\mathcal{C}}\log(T) + H^{\frac{1}{2}}S^{\frac{3}{2}}A^{\frac{3}{2}}\log^{\frac{3}{2}}(T),$$

where $U_{\text{Var}} = \sum_s \sum_{a \neq \pi^\star(s)} \frac{H\mathbb{V}^c(s)\log(T)}{\Delta(s,a)}$. $\qquad\square$

*Remark* D.11 (Restatement of Remark 4.3). In the stochastic regime with adversarial corruption, suppose that the uncorrupted losses are generated independently and are uncorrelated across layers, Theorem 4.2 improves by a factor of $H$ to $U_{\text{Var}} = \sum_s \sum_{a \neq \pi^\star(s)} \frac{\mathbb{V}^c(s)\log(T)}{\Delta(s,a)}$.

*Proof.* In the proof of Theorem 4.2, applying Corollary D.6 to (59) yields the following inequality in place of (60):

$$\sum_{s,a} \sqrt{\log(T)\mathbb{E}\left[\sum_{t=1}^T q^{\pi_t}(s,a)^2(\widehat{\ell}_t(s,a) + g_t(s,a) - m_t(s,a))^2\right]}$$

$$\leq \sum_{s,a} \sqrt{\log(T)\mathbb{E}\left[\sum_{t=1}^T 4\mathbb{V}^c(s)q^{\pi_t}(s,a)(1 - \pi_t(a \mid s))\right]}$$

$$+ \sum_{s,a} \sqrt{\log(T)\mathbb{E}\left[4H\sum_{t=1}^T \sum_{s',a'} q^{\pi_t}(s',a')(\ell_t(s',a') - \ell'_t(s',a') + \mu(s',a') - m_t(s',a'))^2\right]}.$$

Compared to (60), this bound is improved by a factor of $H$, and can be interpreted as replacing the $H\mathbb{V}^c(s)$ term by $\mathbb{V}^c(s)$. The remainder of the proof follows by the same steps as in Theorem 4.2. $\qquad\square$

# E. Regret Analysis of Policy Optimization (deferred from Section 5)

In this section, we provide the missing details from Section 5 and present the full regret analysis leading to the proof of Theorems 5.2 and 5.3.

Throughout this section, sums over $t = 1, \ldots, T$ are taken over the augmented sequence including both real and virtual episodes, whereas the data-dependent complexity measures in the main statements are defined only over real episodes. We denote by $\mathcal{T}_r = \{t \in [T] : Y_t = 1\}$ and $\mathcal{T}_v = \{t \in [T] : Y_t = 0\}$ the sets of real and virtual episodes, with cardinalities $|\mathcal{T}_r|$ and $|\mathcal{T}_v|$, respectively.

## E.1. Auxiliary Lemmas

Building on the policy optimization framework of Luo et al. (2021) and Dann et al. (2023a), we use the following key lemma to derive our regret bounds.

**Lemma E.1** (Restatement of Lemma 5.1). *Suppose that $b_t(s)$ is a nonnegative loss function and that, for all $s, a$,*

$$B_t(s,a) = b_t(s) + \left(1 + \frac{1}{H}\right)\mathbb{E}_{s' \sim P(\cdot|s,a), a' \sim \pi_t(\cdot|s')}[B_t(s',a')].$$

*Suppose also that for some $J(s) \geq 0$ it holds that*

$$\mathbb{E}\left[\sum_s q^{\mathring{\pi}}(s)\sum_{t,a}(\pi_t(a \mid s) - \mathring{\pi}(a \mid s))(Q^{\pi_t}(s,a;\ell_t) - B_t(s,a))\right]$$

$$\leq \sum_s q^{\mathring{\pi}}(s)J(s) + \mathbb{E}\left[\sum_{t=1}^T \sum_s q^{\mathring{\pi}}(s)b_t(s)\right] + \mathbb{E}\left[\frac{1}{H}\sum_{t=1}^T \sum_s \sum_a q^{\mathring{\pi}}(s)\pi_t(a \mid s)B_t(s,a)\right]. \qquad (62)$$

*Then,*

$$\text{Reg}_T \leq \sum_s q^{\mathring{\pi}}(s)J(s) + 3\,\mathbb{E}\left[\sum_{t=1}^T V^{\pi_t}(s_0;b_t)\right].$$

Lemma E.1 reduces the regret analysis to proving (62) for an appropriate bonus $b_t(s)$ and its dilated version $B_t(s,a)$. Here $B_t(s,a)$ is the exploration bonus in $Q$-space, and $b_t(s)$ is the one-step bonus that generates it. Once (62) is established, the regret is controlled by the cumulative values $\sum_t V^{\pi_t}(s_0; b_t)$.

To show (62), we choose $b_t$ in (20) and decompose the LHS of (62) as

$$
\sum_s q^{\mathring{\pi}}(s) \sum_{t,a} (\pi_t(a \mid s) - \mathring{\pi}(a \mid s))(Q^{\pi_t}(s,a;\ell_t) - B_t(s,a))
$$

$$
= \sum_s q^{\mathring{\pi}}(s) \underbrace{\sum_{t,a}(\pi_t(a \mid s) - \mathring{\pi}(a \mid s))\Big(\widehat{Q}_t(s,a) - B_t(s,a)\Big)}_{\textbf{reg-term}(s)}
$$

$$
+ \sum_s q^{\mathring{\pi}}(s) \underbrace{\sum_{t,a}(\pi_t(a \mid s) - \mathring{\pi}(a \mid s))\Big(Q^{\pi_t}(s,a;\ell_t) - \widehat{Q}_t(s,a)\Big)}_{\textbf{bias-term}(s)}. \tag{63}
$$

**Lemma E.2.** *It holds that*

$$
\mathbb{E}_t\Big[\widehat{Q}_t(s,a)\Big] = Q^{\pi_t}(s,a;m_t) + \frac{q^{\pi_t}(s)}{q_t(s)} Q^{\pi_t}(s,a;\ell_t - m_t)Y_t - \frac{\gamma_t H}{q_t(s)}.
$$

*for all state-action pairs $(s,a)$.*

*Proof.* By the definition of (14), we have

$$
\mathbb{E}_t\Big[\widehat{Q}_t(s,a)\Big] = \mathbb{E}_t\Big[Q^{\pi_t}(s,a;m_t) + \frac{\mathbb{I}_t(s,a)(L_{t,h(s)} - M_{t,h(s)})}{q_t(s)\pi_t(a \mid s)}Y_t - \frac{\gamma_t H}{q_t(s)}\Big]
$$

$$
= Q^{\pi_t}(s,a;m_t) + q_t(s,a)\mathbb{E}_t\Big[\frac{L_{t,h(s)} - M_{t,h(s)}}{q_t(s)\pi_t(a \mid s)} \Big| \mathbb{I}_t(s,a) = 1\Big]Y_t - \frac{\gamma_t H}{q_t(s)}
$$

$$
= Q^{\pi_t}(s,a;m_t) + \frac{q^{\pi_t}(s,a)}{q_t(s)\pi_t(a \mid s)} Q^{\pi_t}(s,a;\ell_t - m_t)Y_t - \frac{\gamma_t H}{q_t(s)}
$$

$$
= Q^{\pi_t}(s,a;m_t) + \frac{q^{\pi_t}(s)}{q_t(s)} Q^{\pi_t}(s,a;\ell_t - m_t)Y_t - \frac{\gamma_t H}{q_t(s)}.
$$

$\square$

**Lemma E.3.** *The variables $b_t(s)$ in (20) and $B_t(s,a)$ in (18) satisfy*

$$
\eta_t(s,a)\pi_t(a \mid s)B_t(s,a) \leq \frac{1}{5H}, \quad B_t(s,a) \leq \frac{2\sqrt{HS}}{\gamma_t} + 15H^2 \tag{64}
$$

*for any episode $t$ and state-action pair $(s,a)$.*

*Proof.* Let $R_t = \max_{s,a} \frac{\eta_t(s,a)}{q_t(s)}$. We first note that the dilated bonus-to-go $B_t(s,a)$ is bounded via the dilated recursion. Unrolling it for at most $H$ steps and using $(1 + 1/H)^H \leq 3$, we obtain

$$
B_t(s,a) \leq 3 \sum_{h=h(s)}^{H-1} \sum_{s' \in \mathcal{S}_h} q^{\pi_t}(s' \mid s,a)\, b_t(s'). \tag{65}
$$

Then, we first consider the case when $t$ is a real episode. In real episodes, by the definition of $b_t$ and the learning-rate

update,

$$
\begin{aligned}
b_t(s) &= 6 \sum_a \left( \frac{1}{\eta_{t+1}(s,a)} - \frac{1}{\eta_t(s,a)} \right) \log(T) + 5 \frac{\gamma_t H}{q_t(s)} \\
&= 6 \sum_a \frac{\eta_t(s,a)\zeta_t(s,a)}{q_t(s)^2} + 5H \\
&= 6 \sum_a \frac{\eta_t(s,a)\big(\mathbb{I}_t(s,a) - \pi_t(a \mid s)\mathbb{I}_t(s)\big)^2 \big(L_{t,h(s)} - M_{t,h(s)}\big)^2}{q_t(s)^2} + 5H \\
&\leq \frac{6H^2}{q_t(s)^2} \max_a \eta_t(s,a) \sum_a \big(\mathbb{I}_t(s,a) - \pi_t(a \mid s)\mathbb{I}_t(s)\big)^2 + 5H \\
&\leq \frac{12H^2}{q_t(s)^2} \max_a \eta_t(s,a) + 5H \\
&\leq \frac{12H^2}{q_t(s)} \max_a \frac{\eta_t(s,a)}{q_t(s)} + 5H.
\end{aligned}
\tag{66}
$$

Using (65) and (66), we have

$$
\begin{aligned}
B_t(s,a) &\leq 3 \sum_{h=h(s)}^{H-1} \sum_{s' \in \mathcal{S}_h} q^{\pi_t}(s' \mid s,a) b_t(s') \\
&\leq 36 H^2 R_t \left( \sum_{h=h(s)}^{H-1} \sum_{s' \in \mathcal{S}_h} q^{\pi_t}(s' \mid s,a) \frac{1}{q_t(s')} \right) + 15H^2 \\
&\leq 36 H^2 R_t \left( \sum_{h=h(s)}^{H-1} \sum_{s' \in \mathcal{S}_h} q^{\pi_t}(s' \mid s,a) \frac{1}{q^{\pi_t}(s,a) q^{\pi_t}(s' \mid s,a) + \gamma_t} \right) + 15H^2 \\
&\leq 36 H^2 R_t \left( \sum_{h=h(s)}^{H-1} \sum_{s' \in \mathcal{S}_h} \frac{1}{q^{\pi_t}(s,a) + \gamma_t} \right) + 15H^2 \\
&\leq 36 H^2 S R_t \cdot \frac{1}{q^{\pi_t}(s,a) + \gamma_t} + 15H^2 \\
&\leq \frac{2\sqrt{HS}}{\gamma_t} + 15H^2,
\end{aligned}
\tag{67}
$$

where in the last inequality we used $R_t \leq \frac{1}{18\sqrt{H^3 S}}$ that holds in real episodes. This is the desired second inequality in (64). By using (67), we also have

$$
\begin{aligned}
\eta_t(s,a)\pi_t(a \mid s) B_t(s,a) &\leq 36 H^2 S R_t \cdot \frac{\eta_t(s,a)\pi_t(a \mid s)}{q^{\pi_t}(s,a) + \gamma_t} + 15\eta_1 H^2 \\
&\leq 36 H^2 S R_t \cdot \frac{\eta_t(s,a)}{q_t(s)} + 15\eta_1 H^2 \\
&\leq 36 H^2 S R_t^2 + 15\eta_1 H^2 \\
&\leq \frac{1}{9H} + \frac{1}{12H} \leq \frac{1}{5H},
\end{aligned}
$$

where we used $\frac{\eta_t(s,a)}{q_t(s)} \leq R_t$ and $\eta_1 \leq \frac{1}{180H^3}$. This is the desired first inequality in (64).

We next consider the case when $t$ is a virtual episode. In a virtual episode, only the single pair $(s_t^\dagger, a_t^\dagger)$ is updated, and thus

$$
b_t(s) = 6 \sum_a \left( \frac{1}{\eta_{t+1}(s,a)} - \frac{1}{\eta_t(s,a)} \right) \log(T) + 5 \frac{\gamma_t H}{q_t(s)}
$$

$$
= \sum_a \frac{\mathbb{1}\{(s_t^\dagger, a_t^\dagger) = (s,a)\}}{54\eta_t(s,a)H} + 5H
$$

$$
= \sum_a \frac{\mathbb{1}\{(s_t^\dagger, a_t^\dagger) = (s,a)\}}{54Hq_t(s)} \cdot \frac{1}{\max_{s',a'} \frac{\eta_t(s',a')}{q_t(s')}} + 5H \qquad \left(\text{since } (s_t^\dagger, a_t^\dagger) \in \arg\max_{s,a} \frac{\eta_t(s,a)}{q_t(s)}\right)
$$

$$
= \frac{\mathbb{1}\{s_t^\dagger = s\}}{q_t(s)} \cdot \frac{1}{54H \max_{s',a'} \frac{\eta_t(s',a')}{q_t(s')}} + 5H. \tag{68}
$$

Using (65) and (68), we have

$$
\begin{aligned}
B_t(s,a) &\leq 3 \sum_{h=h(s)}^{H-1} \sum_{s' \in \mathcal{S}_h} q^{\pi_t}(s' \mid s, a) b_t(s') \\
&\leq \frac{1}{18HR_t} \sum_{h=h(s)}^{H-1} \sum_{s' \in \mathcal{S}_h} q^{\pi_t}(s' \mid s, a) \frac{\mathbb{1}\{s_t^\dagger = s'\}}{q_t(s')} + 15H^2 \\
&\leq \frac{1}{18HR_t} \sum_{h=h(s)}^{H-1} \sum_{s' \in \mathcal{S}_h} q^{\pi_t}(s' \mid s, a) \frac{\mathbb{1}\{s_t^\dagger = s'\}}{q^{\pi_t}(s,a) q^{\pi_t}(s' \mid s, a) + \gamma_t} + 15H^2 \\
&\leq \frac{1}{18HR_t} \sum_{h=h(s)}^{H-1} \sum_{s' \in \mathcal{S}_h} \frac{\mathbb{1}\{s_t^\dagger = s'\}}{q^{\pi_t}(s,a) + \gamma_t} + 15H^2 \\
&\leq \frac{1}{18HR_t} \frac{1}{q^{\pi_t}(s)\pi_t(a \mid s) + \gamma_t} + 15H^2 \\
&\leq \frac{\sqrt{HS}}{\gamma_t} + 15H^2, \tag{69}
\end{aligned}
$$

where in the last inequality we used $R_t > \frac{1}{18\sqrt{H^3 S}}$ in a virtual episode. This is the desired second inequality in (64).

By using (69), we also have

$$
\begin{aligned}
\eta_t(s,a)\pi_t(a \mid s) B_t(s,a) &\leq \frac{1}{18HR_t} \frac{\eta_t(s,a)\pi_t(a \mid s)}{q^{\pi_t}(s)\pi_t(a \mid s) + \gamma_t} + 15\eta_1 H^2 \\
&\leq \frac{1}{18HR_t} \frac{\eta_t(s,a)}{q_t(s)} + 15\eta_1 H^2 \\
&\leq \frac{1}{18H} + \frac{1}{12H} \leq \frac{1}{5H},
\end{aligned}
$$

where we used $\frac{\eta_t(s,a)}{q_t(s)} \leq R_t$ and $\eta_1 = \frac{1}{180H^3}$. This is the desired first inequality in (64). $\qquad \square$

**Lemma E.4** (Dann et al. 2023a, Lemma G.3). *Let $\eta_1 > 0, \eta_2, \eta_3, \ldots$ be updated by*

$$
\frac{1}{\eta_{t+1}} = \frac{1}{\eta_t} + \eta_t \phi_t \qquad \forall t \geq 1
$$

*with $0 \leq \phi_t \leq \eta_t^{-2}$. Then,*

$$
\frac{1}{\eta_{t+1}} \geq \frac{1}{2} \sqrt{\sum_{\tau=1}^{t+1} \phi_\tau}.
$$

**Lemma E.5.** *Suppose that the learning rates are updated according to (15). Then, it holds*

$$
\eta_t(s,a) \leq \frac{2\sqrt{\log(T)}}{\sqrt{\sum_{\tau \leq t : \tau \in \mathcal{T}_r} \frac{\zeta_\tau(s,a)}{q_\tau(s)^2}}}
$$

*for any episode $t$ and state-action pair $(s, a)$.*

*Proof.* Let $\phi_t(s, a) = \frac{\zeta_t(s,a)}{q_t(s)^2 \log(T)}$ in real episodes and $\phi_t(s, a) = \frac{\mathbb{I}\{(s_t^\dagger, a_t^\dagger) = (s,a)\}}{324 \eta_t(s,a)^2 H \log(T)}$ in virtual episodes. Then the update rule of learning rates can be written as

$$\frac{1}{\eta_{t+1}(s, a)} = \frac{1}{\eta_t(s, a)} + \eta_t(s, a) \phi_t(s, a).$$

To apply Lemma E.4, it suffices to show that $\phi_t(s, a) \leq \frac{1}{\eta_t(s,a)^2}$. This is clear for virtual episodes. For real episodes,

$$\phi_t(s, a) \eta_t(s, a)^2 = \frac{\eta_t(s, a)^2 \zeta_t(s, a)}{q_t(s)^2 \log(T)} \leq \frac{H^2}{\log(T)} \left( \frac{\eta_t(s, a)}{q_t(s)} \right)^2 \leq \frac{H^2}{\log(T)} \cdot \frac{1}{18^2 H^3 S} \leq 1,$$

which follows from $\zeta_t(s, a) \leq H^2$ and $\frac{\eta_t(s,a)}{q_t(s)} \leq \frac{1}{18\sqrt{H^3 S}}$ in real episodes.

Then, by Lemma E.4, we have

$$\eta_t(s, a) \leq \frac{2}{\sqrt{\sum_{\tau \leq t} \phi_\tau}} \leq \frac{2\sqrt{\log(T)}}{\sqrt{\sum_{\tau \leq t : \tau \in \mathcal{T}_r} \frac{\zeta_\tau(s,a)}{q_\tau(s)^2}}}.$$

$\square$

**Lemma E.6.** *The number of virtual episodes $|\mathcal{T}_v|$ is upper bounded by*

$$|\mathcal{T}_v| \lesssim HSA \log^2(T).$$

*Proof.* By the definition of virtual episodes, whenever $t \in \mathcal{T}_v$, there exists a pair $(s, a)$ such that $\frac{\eta_t(s,a)}{q_t(s)} > \frac{1}{18\sqrt{H^3 S}}$. Moreover, in virtual episodes, the corresponding learning rate will shrink by a factor of $\left( 1 + \frac{1}{324 H \log(T)} \right)$ for a state-action pair $(s_t^\dagger, a_t^\dagger)$. Hence, for each fixed $(s, a)$, the number of virtual updates on this pair is at most the number of multiplicative shrink steps needed to reduce $\eta_t(s, a)$ from its initial value $\eta_1$ to $\frac{\max_{t \in [T]} q_t(s)}{18\sqrt{H^3 S}} \geq \frac{\gamma_T}{18\sqrt{H^3 S}}$. Hence,

$$|\mathcal{T}_v| \lesssim SA \cdot \frac{\log\left( \frac{\eta_1}{\gamma_T / (18\sqrt{H^3 S})} \right)}{\log\left( 1 + \frac{1}{H \log T} \right)} \lesssim SA \cdot \frac{\log \frac{\sqrt{H^3 S} T \eta_1}{\sqrt{HS}}}{\log\left( 1 + \frac{1}{H \log T} \right)} \lesssim HSA \log^2(T),$$

where we used $\log(1 + x) \geq x/2$ for $x \in (0, 1]$ and $\gamma_T = \frac{\sqrt{HS}}{T}$. $\square$

**Lemma E.7.** *It holds that*

$$\mathbb{E}_t[\zeta_t(s, a)]$$
$$\leq 2H \mathbb{V}^c(s) q^{\pi_t}(s, a)(1 - \pi_t(a \mid s)) + 2H \sum_{s', a'} q^{\pi_t}(s', a') \mathbb{E}_t\left[ (\ell_t(s', a') - \ell_t'(s', a') + \mu(s', a') - m_t(s', a'))^2 \right]$$

*for all state-action pairs $(s, a)$.*

*Proof.* Fix any $(s, a)$. Define $\kappa_t(s, a) = \ell_t'(s, a) - \mu(s, a), \lambda_t(s, a) = \ell_t(s, a) - \ell_t'(s, a) + \mu(s, a) - m_t(s, a)$ so that

$$\ell_t(s, a) - m_t(s, a) = \kappa_t(s, a) + \lambda_t(s, a).$$

Conditioning on which action is taken at state $s$ in episode $t$, we write

$$
\begin{aligned}
\mathbb{E}_t[\zeta_t(s,a)] &= \mathbb{E}_t\big[(\mathbb{I}_t(s,a) - \pi_t(a \mid s)\mathbb{I}_t(s))^2(L_{t,h(s)} - M_{t,h(s)})^2\big] \\
&= q^{\pi_t}(s,a)\mathbb{E}_t\big[(\mathbb{I}_t(s,a) - \pi_t(a \mid s)\mathbb{I}_t(s))^2(L_{t,h(s)} - M_{t,h(s)})^2 \mid \mathbb{I}_t(s,a) = 1\big] \\
&\quad + \sum_{b \neq a} q^{\pi_t}(s,b)\mathbb{E}_t\big[(\mathbb{I}_t(s,a) - \pi_t(a \mid s)\mathbb{I}_t(s))^2(L_{t,h(s)} - M_{t,h(s)})^2 \mid \mathbb{I}_t(s,b) = 1\big] \\
&= q^{\pi_t}(s,a)(1 - \pi_t(a \mid s))^2\mathbb{E}_t\big[(L_{t,h(s)} - M_{t,h(s)})^2 \mid \mathbb{I}_t(s,a) = 1\big] \\
&\quad + \sum_{b \neq a} q^{\pi_t}(s,b)\pi_t(a \mid s)^2\mathbb{E}_t\big[(L_{t,h(s)} - M_{t,h(s)})^2 \mid \mathbb{I}_t(s,b) = 1\big]. \tag{70}
\end{aligned}
$$

By the definitions of $L_{t,h(s)}$ and $M_{t,h(s)}$, we have

$$
\begin{aligned}
L_{t,h(s)} - M_{t,h(s)} &= \sum_{h'=h(s)}^{H-1} (\ell_t(s_{t,h'}, a_{t,h'}) - m_t(s_{t,h'}, a_{t,h'})) \\
&= \sum_{h'=h(s)}^{H-1} (\kappa_t(s_{t,h'}, a_{t,h'}) + \lambda_t(s_{t,h'}, a_{t,h'})).
\end{aligned}
$$

Then, for any $b \in \mathcal{A}$ we obtain

$$
\begin{aligned}
&\mathbb{E}_t\Big[\big(L_{t,h(s)} - M_{t,h(s)}\big)^2 \,\Big|\, \mathbb{I}_t(s,b) = 1\Big] \\
&= \mathbb{E}_t\left[\left(\sum_{h'=h(s)}^{H-1} (\kappa_t(s_{t,h'}, a_{t,h'}) + \lambda_t(s_{t,h'}, a_{t,h'}))\right)^2 \,\Bigg|\, \mathbb{I}_t(s,b) = 1\right] \\
&\leq 2\mathbb{E}_t\left[\left(\sum_{h'=h(s)}^{H-1} \kappa_t(s_{t,h'}, a_{t,h'})\right)^2 \,\Bigg|\, \mathbb{I}_t(s,b) = 1\right] + 2\mathbb{E}_t\left[\left(\sum_{h'=h(s)}^{H-1} \lambda_t(s_{t,h'}, a_{t,h'})\right)^2 \,\Bigg|\, \mathbb{I}_t(s,b) = 1\right], \tag{71}
\end{aligned}
$$

where we used $(x + y)^2 \leq 2(x^2 + y^2)$ for $x, y \in \mathbb{R}$. By the Cauchy–Schwarz inequality, the first term in (71) is evaluated as

$$
\begin{aligned}
2\mathbb{E}_t\left[\left(\sum_{h'=h(s)}^{H-1} \kappa_t(s_{t,h'}, a_{t,h'})\right)^2 \,\Bigg|\, \mathbb{I}_t(s,b) = 1\right] &\leq 2H\mathbb{E}_t\left[\sum_{h'=h(s)}^{H-1} \kappa_t(s_{t,h'}, a_{t,h'})^2 \,\Bigg|\, \mathbb{I}_t(s,b) = 1\right] \\
&= 2H\sum_{s',a'} q^{\pi_t}(s',a' \mid s,b)\sigma^2(s',a') \\
&\leq 2H\mathbb{V}^c(s). \tag{72}
\end{aligned}
$$

For the second term in (71), the same argument yields

$$
\begin{aligned}
2\mathbb{E}_t\left[\left(\sum_{h'=h(s)}^{H-1} \lambda_t(s_{t,h'}, a_{t,h'})\right)^2 \,\Bigg|\, \mathbb{I}_t(s,b) = 1\right] &\leq 2H\mathbb{E}_t\left[\sum_{h'=h(s)}^{H-1} \lambda_t(s_{t,h'}, a_{t,h'})^2 \,\Bigg|\, \mathbb{I}_t(s,b) = 1\right] \\
&\leq 2H\sum_{s',a'} q^{\pi_t}(s',a' \mid s,b)\mathbb{E}_t\big[\lambda_t(s',a')^2\big].
\end{aligned}
$$

Combining the above two bounds, we obtain

$$
\mathbb{E}_t\Big[\big(L_{t,h(s)} - M_{t,h(s)}\big)^2 \,\Big|\, \mathbb{I}_t(s,b) = 1\Big] \leq 2H\mathbb{V}^c(s) + 2H\sum_{s',a'} q^{\pi_t}(s',a' \mid s,b)\mathbb{E}_t\big[\lambda_t(s',a')^2\big]. \tag{73}
$$

Thus, combining (73) with (70) yields

$$
\mathbb{E}_t[\zeta_t(s,a)] \leq q^{\pi_t}(s,a)(1-\pi_t(a\mid s))^2 \left( 2H\mathbb{V}^c(s) + 2H\sum_{s',a'} q^{\pi_t}(s',a'\mid s,a)\mathbb{E}_t\big[\lambda_t(s',a')^2\big] \right)
$$

$$
+ \sum_{b\neq a} q^{\pi_t}(s,b)\pi_t(a\mid s)^2 \left( 2H\mathbb{V}^c(s) + 2H\sum_{s',a'} q^{\pi_t}(s',a'\mid s,b)\mathbb{E}_t\big[\lambda_t(s',a')^2\big] \right)
$$

$$
= 2H\mathbb{V}^c(s)q^{\pi_t}(s,a)(1-\pi_t(a\mid s))^2 + 2H(1-\pi_t(a\mid s))^2\sum_{s',a'} q^{\pi_t}(s,a)q^{\pi_t}(s',a'\mid s,a)\mathbb{E}_t\big[\lambda_t(s',a')^2\big]
$$

$$
+ 2H\mathbb{V}^c(s)\sum_{b\neq a} q^{\pi_t}(s,b)\pi_t(a\mid s)^2 + 2H\pi_t(a\mid s)^2\sum_{s',a'}\sum_{b\neq a} q^{\pi_t}(s,b)q^{\pi_t}(s',a'\mid s,b)\mathbb{E}_t\big[\lambda_t(s',a')^2\big]
$$

$$
\leq 2H\mathbb{V}^c(s)q^{\pi_t}(s)\pi_t(a\mid s)(1-\pi_t(a\mid s)) + 2H\sum_{s',a'} q^{\pi_t}(s',a')\mathbb{E}_t\big[\lambda_t(s',a')^2\big],
$$

which completes the proof. $\qquad\square$

**Corollary E.8.** *Under the stochastic regime with adversarial corruption, suppose that the uncorrupted losses are generated independently and are uncorrelated across layers. Then, it holds that*

$$
\mathbb{E}_t[\zeta_t(s,a)]
$$
$$
\leq 2\mathbb{V}^c(s)q^{\pi_t}(s,a)(1-\pi_t(a\mid s)) + 2H\sum_{s',a'} q^{\pi_t}(s',a')\mathbb{E}_t\big[(\ell_t(s',a') - \ell_t'(s',a') + \mu(s',a') - m_t(s',a'))^2\big]
$$

*for all state-action pairs $(s,a)$.*

*Proof.* This corollary can be viewed as a simple variant of Lemma E.7. Let $\kappa_t(s,a) := \ell_t'(s,a) - \mu(s,a)$. Since the uncorrupted losses are generated independently and are uncorrelated across layers, it holds that for any $(s_1,a_1) \neq (s_2,a_2)$,

$$
\mathbb{E}_t[\mathbb{I}_t(s_1,a_1)\mathbb{I}_t(s_2,a_2)\kappa_t(s_1,a_1)\kappa_t(s_2,a_2)] = 0.
$$

Then, for any function $\alpha : \mathcal{S}\times\mathcal{A} \to \mathbb{R}$, we have

$$
\left( \sum_{s,a} \alpha(s,a)\mathbb{I}_t(s,a)\kappa_t(s,a) \right)^2 = \sum_{s_1,a_1}\sum_{s_2,a_2} \alpha(s_1,a_1)\alpha(s_2,a_2)\mathbb{I}_t(s_1,a_1)\mathbb{I}_t(s_2,a_2)\kappa_t(s_1,a_1)\kappa_t(s_2,a_2)
$$
$$
= \sum_{s,a} \alpha(s,a)^2\mathbb{I}_t(s,a)\kappa_t(s,a)^2.
$$

Thus, for the first term in (71) is evaluated as

$$
2\mathbb{E}_t\left[ \left( \sum_{h'=h(s)}^{H-1} \kappa_t(s_{t,h'},a_{t,h'}) \right)^2 \Bigg| \mathbb{I}_t(s,b) = 1 \right] \leq 2\mathbb{E}_t\left[ \sum_{h'=h(s)}^{H-1} \kappa_t(s_{t,h'},a_{t,h'})^2 \Bigg| \mathbb{I}_t(s,b) = 1 \right]
$$
$$
= 2\sum_{s',a'} q^{\pi_t}(s',a'\mid s,b)\sigma^2(s',a')
$$
$$
\leq 2\mathbb{V}^c(s). \tag{74}
$$

Therefore, compared with Lemma E.7, we obtain an $H$-times sharper bound in (74) than (72). As a consequence, the corresponding $\mathbb{V}^c$ term is also improved by a factor of $H$. $\qquad\square$

**Lemma E.9.** *For each state-action pair $(s,a)$, it holds that*

$$
\mathbb{E}\left[ \sum_{t=1}^{T}\sum_{s,a} \mathbb{I}_t(s,a)(L_{t,h(s)} - M_{t,h(s)})^2 \right] \leq H^2\mathbb{E}\left[ \sum_{t=1}^{T}\sum_{s,a} \mathbb{I}_t(s,a)(\ell_t(s,a) - m_t(s,a))^2 \right].
$$

*Proof.* By the definitions of $L_{t,h(s)}$ and $M_{t,h(s)}$, we have

$$L_{t,h(s)} - M_{t,h(s)} = \sum_{h'=h(s)}^{H-1} (\ell_t(s_{t,h'}, a_{t,h'}) - m_t(s_{t,h'}, a_{t,h'})).$$

Hence, we have

$$\mathbb{E}\left[\sum_{t=1}^{T} \sum_{s,a} \mathbb{I}_t(s,a)\big(L_{t,h(s)} - M_{t,h(s)}\big)^2\right]$$

$$= \mathbb{E}\left[\sum_{t=1}^{T} \sum_{s,a} \mathbb{I}_t(s,a)\left(\sum_{h'=h(s)}^{H-1} (\ell_t(s_{t,h'}, a_{t,h'}) - m_t(s_{t,h'}, a_{t,h'}))\right)^2\right]$$

$$\leq H\mathbb{E}\left[\sum_{t=1}^{T} \sum_{s,a} \mathbb{I}_t(s,a) \sum_{h'=h(s)}^{H-1} (\ell_t(s_{t,h'}, a_{t,h'}) - m_t(s_{t,h'}, a_{t,h'}))^2\right] \qquad \text{(by the Cauchy–Schwarz inequality)}$$

$$\leq H\mathbb{E}\left[\sum_{t=1}^{T} \sum_{h=0}^{H-1} \sum_{(s,a)\in\mathcal{S}_h\times\mathcal{A}} \mathbb{I}_t(s,a) \sum_{h'=0}^{H-1} (\ell_t(s_{t,h'}, a_{t,h'}) - m_t(s_{t,h'}, a_{t,h'}))^2\right]$$

$$= H\mathbb{E}\left[\sum_{t=1}^{T} \sum_{h=0}^{H-1} \sum_{h'=0}^{H-1} (\ell_t(s_{t,h'}, a_{t,h'}) - m_t(s_{t,h'}, a_{t,h'}))^2 \sum_{(s,a)\in\mathcal{S}_h\times\mathcal{A}} \mathbb{I}_t(s,a)\right],$$

where the last equality rearranges the summations.

Since for each fixed $(t,h)$ exactly one state-action pair is visited, we have $\sum_{(s,a)\in\mathcal{S}_h\times\mathcal{A}} \mathbb{I}_t(s,a) = 1$, and then,

$$H\mathbb{E}\left[\sum_{t=1}^{T} \sum_{h=0}^{H-1} \sum_{h'=0}^{H-1} (\ell_t(s_{t,h'}, a_{t,h'}) - m_t(s_{t,h'}, a_{t,h'}))^2 \sum_{(s,a)\in\mathcal{S}_h\times\mathcal{A}} \mathbb{I}_t(s,a)\right]$$

$$= H\mathbb{E}\left[\sum_{t=1}^{T} \sum_{h=0}^{H-1} \sum_{h'=0}^{H-1} (\ell_t(s_{t,h'}, a_{t,h'}) - m_t(s_{t,h'}, a_{t,h'}))^2\right] \qquad (\sum_{(s,a)\in\mathcal{S}_h\times\mathcal{A}} \mathbb{I}_t(s,a) = 1)$$

$$= H^2\mathbb{E}\left[\sum_{t=1}^{T} \sum_{h'=0}^{H-1} (\ell_t(s_{t,h'}, a_{t,h'}) - m_t(s_{t,h'}, a_{t,h'}))^2\right]$$

$$= H^2\mathbb{E}\left[\sum_{t=1}^{T} \sum_{s,a} \mathbb{I}_t(s,a)(\ell_t(s,a) - m_t(s,a))^2\right],$$

which completes the proof. $\qquad\square$

### E.2. Common Regret Analysis

Now we are ready to upper bound the RHS of (63). We first consider the bias term, **bias-term**$(s)$.

**Lemma E.10.** *For each state $s \in \mathcal{S}$, it holds that*

$$\mathbb{E}[\textbf{bias-term}(s)] \leq 2\sum_{t=1}^{T} \frac{\gamma_t H}{q_t(s)} + H^2 SA \log^2(T).$$

*Proof.* From the definition of the $Q$-function estimator, we have

$$\mathbb{E}_t\Big[Q^{\pi_t}(s,a;\ell_t) - \widehat{Q}_t(s,a)\Big]$$

$$= \mathbb{E}_t\Big[Q^{\pi_t}(s,a;\ell_t) - \Big(Q^{\pi_t}(s,a;m_t) + \frac{q^{\pi_t}(s)}{q_t(s)}(Q^{\pi_t}(s,a;\ell_t) - Q^{\pi_t}(s,a;m_t))Y_t - \frac{\gamma_t}{q_t(s)}H\Big)\Big] \quad \text{(by Lemma E.2)}$$

$$= \mathbb{E}_t\Big[\frac{q_t(s) - q^{\pi_t}(s)Y_t}{q_t(s)}(Q^{\pi_t}(s,a;\ell_t) - Q^{\pi_t}(s,a;m_t)) + \frac{\gamma_t}{q_t(s)}H\Big]$$

$$= \begin{cases} -Q^{\pi_t}(s,a;m_t) + \dfrac{\gamma_t}{q_t(s)}H & \text{if } Y_t = 0 \\ \dfrac{\gamma_t}{q_t(s)}(Q^{\pi_t}(s,a;\ell_t) - Q^{\pi_t}(s,a;m_t) + H) & \text{if } Y_t = 1 \end{cases}.$$

When $Y_t = 1$, since $\ell_t, m_t \in [0,1]^{S \times A}$, we have

$$0 \le Q^{\pi_t}(s,a;\ell_t - m_t) + H \le 2H.$$

Therefore,

$$0 \le \mathbb{E}_t\Big[Q^{\pi_t}(s,a;\ell_t) - \widehat{Q}_t(s,a)\Big] \le \frac{2\gamma_t H}{q_t(s)}.$$

When $Y_t = 0$, we use $Q^{\pi_t}(s,a;m_t) \le H$ and obtain

$$-H \le \mathbb{E}_t\Big[Q^{\pi_t}(s,a;\ell_t) - \widehat{Q}_t(s,a)\Big] \le \frac{\gamma_t H}{q_t(s)}.$$

Using this bound, we obtain

$$\mathbb{E}[\textbf{bias-term}(s)] = \mathbb{E}\Bigg[\sum_{t=1}^T \sum_a (\pi_t(a \mid s) - \mathring{\pi}(a \mid s))\Big(Q^{\pi_t}(s,a;\ell_t) - \widehat{Q}_t(s,a)\Big)\Bigg]$$

$$\le \mathbb{E}\Bigg[2\sum_{t=1}^T \sum_a \pi_t(a \mid s)\frac{\gamma_t H}{q_t(s)}\Bigg] + \mathbb{E}\Bigg[\sum_{t \in \mathcal{T}_v} \sum_a \mathring{\pi}(a \mid s)H\Bigg]$$

$$= 2\sum_{t=1}^T \frac{\gamma_t H}{q_t(s)} + H|\mathcal{T}_v| \le 2\sum_{t=1}^T \frac{\gamma_t H}{q_t(s)} + H^2 SA \log^2(T),$$

where the last inequality follows from Lemma E.6, which guarantees that the number of virtual episodes satisfies $|\mathcal{T}_v| \le HSA\log^2(T)$. $\qquad\square$

We next consider **reg-term**$(s)$.

**Lemma E.11.** *For each state $s \in \mathcal{S}$, it holds that*

$$\mathbb{E}[\textbf{reg-term}(s)] \le O\big(H^3 A \log(T)\big) + 6\,\mathbb{E}\Bigg[\sum_{t=1}^T \sum_a \Big(\frac{1}{\eta_{t+1}(s,a)} - \frac{1}{\eta_t(s,a)}\Big)\log(T)\Bigg]$$

$$+ \mathbb{E}\Bigg[\frac{1}{H}\sum_{t=1}^T \sum_a \pi_t(a \mid s)B_t(s,a)\Bigg] + 3\sum_{t=1}^T \frac{\gamma_t H}{q_t(s)}.$$

*Proof.* We will apply Lemma C.4 with $p_t = \pi_t(\cdot \mid s)$ and $\ell_t = \widehat{Q}_t(s,a) - B_t(s,a)$ for each $s \in \mathcal{S}$. To do so, in what follows, we will check the conditions of Lemma C.4. Let

$$\widetilde{Q}(s,a) = Q^{\pi_t}(s,a;m_t) + \frac{\mathbb{I}_t(s,a)(L_{t,h(s)} - M_{t,h(s)})}{q_t(s)\pi_t(a \mid s)}Y_t.$$

Then, we have $\widehat{Q}_t(s,a) = \widetilde{Q}(s,a) - \frac{\gamma_t H}{q_t(s)}$. Define

$$x_t = \left\langle -\pi_t(\cdot \mid s), \widetilde{Q}_t(s,\cdot) - Q^{\pi_t}(s,\cdot;m_t) \right\rangle = -\frac{\mathbb{I}_t(s)(L_{t,h(s)} - M_{t,h(s)})}{q_t(s)}Y_t$$

and verify that for all $(s,a)$, $\eta_t(s,a)\pi_t(a \mid s)\left(\widehat{Q}_t(s,a) - B_t(s,a) - Q^{\pi_t}(s,a;m_t) + x_t\right) \geq -1/2$. Recall that in a virtual episode we have $Y_t = 0$ and $\ell_t(s,a) = 0$ for all state-action pairs $(s,a)$. Hence,

$$\eta_t(s,a)\pi_t(a \mid s)\left(\widehat{Q}_t(s,a) - B_t(s,a) - Q^{\pi_t}(s,a;m_t) + x_t\right)$$

$$= \eta_t(s,a)\pi_t(a \mid s)\left(\frac{\mathbb{I}_t(s,a)(L_{t,h(s)} - M_{t,h(s)})}{q_t(s)\pi_t(a \mid s)}Y_t - B_t(s,a) - \frac{\gamma_t H}{q_t(s)} - \frac{\mathbb{I}_t(s)(L_{t,h(s)} - M_{t,h(s)})}{q_t(s)}Y_t\right)$$

$$\geq -\frac{\eta_t(s,a)}{q_t(s)}M_{t,h(s)}Y_t - \eta_t(s,a)\pi_t(a \mid s)B_t(s,a) - \eta_t(s,a)H - \frac{\eta_t(s,a)}{q_t(s)}L_{t,h(s)}Y_t$$

$$\geq -\frac{2\eta_t(s,a)}{q_t(s)}HY_t - \eta_t(s,a)\pi_t(a \mid s)B_t(s,a) - \eta_1 H$$

$$\geq -\frac{1}{9\sqrt{HS}} - \frac{1}{5H} - \frac{1}{180H^2}$$

$$\geq -\frac{1}{2},$$

where the bounds in the third and fourth lines use $0 \leq L_{t,h(s)} \leq H$ and $0 \leq M_{t,h(s)} \leq H$, and the fifth line uses $\frac{\eta_t(s,a)}{q_t(s)} \leq \frac{1}{18\sqrt{H^3 S}}$ in real episodes together with $\eta_t(s,a)\pi_t(a \mid s)B_t(s,a) \leq \frac{1}{5H}$ from Lemma E.3.

Hence, by Lemma C.4, we obtain

$$\mathbb{E}[\textbf{reg-term}(s)]$$

$$\leq \frac{A\log(AT^2)}{\eta_1} + \mathbb{E}\left[\sum_{t=1}^{T}\sum_{a}\left(\frac{1}{\eta_{t+1}(s,a)} - \frac{1}{\eta_t(s,a)}\right)\log(AT^2)\right]$$

$$+ \mathbb{E}\left[\sum_{t=1}^{T}\sum_{a}\eta_t(s,a)\pi_t(a \mid s)^2\left(\left(\widetilde{Q}_t(s,a) - Q^{\pi_t}(s,a;m_t)\right) - B_t(s,a) - \frac{\gamma_t H}{q_t(s)} + x_t\right)^2\right]$$

$$+ \mathbb{E}\left[\frac{1}{T^2}\sum_{t=1}^{T}\left\langle -\mathring{\pi}(\cdot \mid s) + \frac{1}{A}\mathbf{1}, \widehat{Q}_t(s,\cdot) - B_t(s,\cdot)\right\rangle\right] + 2\mathbb{E}[\|Q^{\pi_t}(s,\cdot;m_{T+1})\|_\infty]$$

$$\leq \frac{3A\log(T)}{\eta_1} + 4 + \frac{30H^2}{T} + 2H + 3\mathbb{E}\left[\sum_{t=1}^{T}\sum_{a}\left(\frac{1}{\eta_{t+1}(s,a)} - \frac{1}{\eta_t(s,a)}\right)\log(T)\right]$$

$$\text{(by } T \geq A \text{ and } \|Q^{\pi_t}(s,\cdot;m_{T+1})\|_\infty \leq H)$$

$$+ 3\mathbb{E}\left[\sum_{t=1}^{T}\sum_{a}\eta_t(s,a)\pi_t(a \mid s)^2\left(\widetilde{Q}_t(s,a) - Q^{\pi_t}(s,a;m_t) + x_t\right)^2\right]$$

$$+ 3\mathbb{E}\left[\sum_{t=1}^{T}\sum_{a}\eta_t(s,a)\pi_t(a \mid s)^2\left(B_t(s,a)^2 + \frac{\gamma_t^2 H^2}{q_t(s)^2}\right)\right]$$

$$\leq O(H^3 A\log(T)) + 3\mathbb{E}\left[\sum_{t=1}^{T}\sum_{a}\left(\frac{1}{\eta_{t+1}(s,a)} - \frac{1}{\eta_t(s,a)}\right)\log(T)\right]$$

$$+ 3\underbrace{\mathbb{E}\left[\sum_{t=1}^{T}\sum_{a}\eta_t(s,a)\pi_t(a \mid s)^2\left(\widetilde{Q}_t(s,a) - Q^{\pi_t}(s,a;m_t) + x_t\right)^2\right]}_{\text{stability-term-2}}$$

$$+ \mathbb{E}\left[\frac{1}{H}\sum_{t=1}^{T}\sum_{a}\pi_t(a \mid s)B_t(s,a)\right] + 3\mathbb{E}\left[\sum_{t=1}^{T}\sum_{a}\eta_t(s,a)\pi_t(a \mid s)\frac{\gamma_t H^2}{q_t(s)}\right]. \tag{75}$$

Here, the second inequality follows from

$$\mathbb{E}\left[\frac{1}{T^2}\sum_{t=1}^{T}\left\langle -\mathring{\pi}(\cdot\mid s)+\frac{1}{A}\mathbf{1},\widehat{Q}_t(s,\cdot)-B_t(s,\cdot)\right\rangle\right]$$

$$=\frac{1}{T^2}\left\langle -\mathring{\pi}(\cdot\mid s)+\frac{1}{A}\mathbf{1},\mathbb{E}\left[\sum_{t=1}^{T}\mathbb{E}_t\left[\widehat{Q}_t(s,\cdot)-B_t(s,\cdot)\right]\right]\right\rangle$$

$$\leq\frac{1}{T^2}\left\|-\mathring{\pi}(\cdot\mid s)+\frac{1}{A}\mathbf{1}\right\|_1\left\|\mathbb{E}\left[\sum_{t=1}^{T}\mathbb{E}_t\left[\widehat{Q}_t(s,\cdot)-B_t(s,\cdot)\right]\right]\right\|_\infty$$

$$\leq\frac{2T(2T+15H^2)}{T^2}=4+\frac{30H^2}{T},$$

where we used $\left\|-\mathring{\pi}(\cdot\mid s)+\frac{1}{A}\mathbf{1}\right\|_1\leq 2$, $|\mathbb{E}_t[\widehat{Q}_t(s,a)]|\leq H$ and $B_t(s,a)\leq\frac{2\sqrt{HS}}{\gamma_t}+15H^2$ from Lemma E.3, which together imply $\left\|\mathbb{E}\left[\sum_{t=1}^{T}\mathbb{E}_t\left[\widehat{Q}_t(s,\cdot)-B_t(s,\cdot)\right]\right]\right\|_\infty\leq T(2T+15H^2)$, and the last inequality follows from $\gamma_t\leq q_t(s)$ and Lemma E.3. We further evaluate the stability-term-2 in the last inequality as

$$\eta_t(s,a)\pi_t(a\mid s)^2\left(\widetilde{Q}_t(s,a)-Q^{\pi_t}(s,a;m_t)+x_t\right)^2$$

$$=\eta_t(s,a)\pi_t(a\mid s)^2\left(\frac{\mathbb{I}_t(s,a)(L_{t,h(s)}-M_{t,h(s)})}{q_t(s)\pi_t(a\mid s)}-\frac{\mathbb{I}_t(s)(L_{t,h(s)}-M_{t,h(s)})}{q_t(s)}\right)^2 Y_t$$

$$=\eta_t(s,a)\left(\frac{\mathbb{I}_t(s,a)(L_{t,h(s)}-M_{t,h(s)})}{q_t(s)}-\frac{\pi_t(a\mid s)\mathbb{I}_t(s)(L_{t,h(s)}-M_{t,h(s)})}{q_t(s)}\right)^2 Y_t$$

$$=\frac{\eta_t(s,a)}{q_t(s)^2}(\mathbb{I}_t(s,a)-\pi_t(a\mid s)\mathbb{I}_t(s))^2(L_{t,h(s)}-M_{t,h(s)})^2 Y_t$$

$$=\frac{\eta_t(s,a)\zeta_t(s,a)}{q_t(s)^2}Y_t,$$

where the last equality follows from the definition of $\zeta_t(s,a)=(\mathbb{I}_t(s,a)-\pi_t(a\mid s)\mathbb{I}_t(s))^2(L_{t,h(s)}-M_{t,h(s)})^2$. Then, stability-term-2 is evaluated as

$$\text{stability-term-2}=\sum_{t=1}^{T}\sum_a\eta_t(s,a)\pi_t(a\mid s)^2\left(\widetilde{Q}_t(s,a)-Q^{\pi_t}(s,a;m_t)+x_t\right)^2$$

$$=\sum_{t=1}^{T}\sum_a\frac{\eta_t(s,a)\zeta_t(s,a)}{q_t(s)^2}Y_t$$

$$\leq\sum_{t=1}^{T}\sum_a\left(\frac{1}{\eta_{t+1}(s,a)}-\frac{1}{\eta_t(s,a)}\right)\log(T),$$

where the last inequality follows from (15). Then, together with (75) and $\eta_t(s,a)\leq\frac{1}{H}$, we obtain

$$\mathbb{E}[\textbf{reg-term}(s)]\leq O\left(H^3 A\log(T)\right)+6\mathbb{E}\left[\sum_{t=1}^{T}\sum_a\left(\frac{1}{\eta_{t+1}(s,a)}-\frac{1}{\eta_t(s,a)}\right)\log(T)\right]$$

$$+\mathbb{E}\left[\frac{1}{H}\sum_{t=1}^{T}\sum_a\pi_t(a\mid s)B_t(s,a)\right]+3\mathbb{E}\left[\sum_{t=1}^{T}\sum_a\pi_t(a\mid s)\frac{\gamma_t H}{q_t(s)}\right]$$

$$=O\left(H^3 A\log(T)\right)+6\mathbb{E}\left[\sum_{t=1}^{T}\sum_a\left(\frac{1}{\eta_{t+1}(s,a)}-\frac{1}{\eta_t(s,a)}\right)\log(T)\right]$$

$$+\mathbb{E}\left[\frac{1}{H}\sum_{t=1}^{T}\sum_a\pi_t(a\mid s)B_t(s,a)\right]+3\sum_{t=1}^{T}\frac{\gamma_t H}{q_t(s)}.$$

$\square$

**Lemma E.12.** *Algorithm 2 guarantees*

$$\mathrm{Reg}_T \le O\big(H^3 SA \log^2(T)\big) + 3\mathbb{E}\left[\sum_{t=1}^{T} V^{\pi_t}(s_0; b_t)\right],$$

*where $b_t$ is defined in (20).*

*Proof.* By the definition of the regret decomposition in (63),

$$\mathbb{E}\left[\sum_s q^{\mathring{\pi}}(s) \sum_{t,a} (\pi_t(a \mid s) - \mathring{\pi}(a \mid s))(Q^{\pi_t}(s,a; \ell_t) - B_t(s,a))\right]$$

$$= \mathbb{E}\left[\sum_s q^{\mathring{\pi}}(s) \cdot \mathbf{reg\text{-}term}(s)\right] + \mathbb{E}\left[\sum_s q^{\mathring{\pi}}(s) \cdot \mathbf{bias\text{-}term}(s)\right]$$

$$\le O\big(H^4 A \log(T)\big) + 6\mathbb{E}\left[\sum_{t=1}^{T} \sum_s q^{\mathring{\pi}}(s) \sum_a \left(\frac{1}{\eta_{t+1}(s,a)} - \frac{1}{\eta_t(s,a)}\right) \log(T)\right]$$

$$+ \mathbb{E}\left[\frac{1}{H} \sum_{t=1}^{T} \sum_s q^{\mathring{\pi}}(s) \sum_a \pi_t(a \mid s) B_t(s,a)\right] + 5\sum_{t=1}^{T} \sum_s q^{\mathring{\pi}}(s) \frac{\gamma_t H}{q_t(s)} + H^3 SA \log^2(T)$$

$$= O\big(H^3 SA \log^2(T)\big) + \mathbb{E}\left[\sum_{t=1}^{T} \sum_s q^{\mathring{\pi}}(s) b_t(s)\right] + \mathbb{E}\left[\frac{1}{H} \sum_{t=1}^{T} \sum_s q^{\mathring{\pi}}(s) \sum_a \pi_t(a \mid s) B_t(s,a)\right],$$

where we used Lemmas E.10 and E.11 and the definition of $b_t$. Combining the last inequality with Lemma E.1 completes the proof. $\square$

**Lemma E.13.** *It holds that*

$$\mathbb{E}\left[\sum_{t=1}^{T} V^{\pi_t}(s_0; b_t)\right] \lesssim \log(T) \sum_{s,a} \sqrt{\mathbb{E}\left[\sum_{t \in \mathcal{T}_r} \zeta_t(s,a)\right]} + H^{\frac{3}{2}} S^{\frac{3}{2}} A \log^2(T).$$

*Proof of Lemma E.13.* We use $\mathcal{T}_r$ and $\mathcal{T}_v$ to denote the set of real and virtual episodes, respectively. Then we have

$$\sum_{t=1}^{T} V^{\pi_t}(s_0; b_t)$$

$$= \sum_{t=1}^{T} \sum_s q^{\pi_t}(s) b_t(s)$$

$$= 6 \sum_{t \in \mathcal{T}_r} \sum_{s,a} q^{\pi_t}(s) \left(\frac{1}{\eta_{t+1}(s,a)} - \frac{1}{\eta_t(s,a)}\right) \log(T) + 6 \sum_{t \in \mathcal{T}_v} \sum_{s,a} q^{\pi_t}(s) \left(\frac{1}{\eta_{t+1}(s,a)} - \frac{1}{\eta_t(s,a)}\right) \log(T)$$

$$+ 5 \sum_{t=1}^{T} \sum_s q^{\pi_t}(s) \frac{\gamma_t H}{q_t(s)}$$

$$\lesssim \sum_{t \in \mathcal{T}_r} \sum_{s,a} q^{\pi_t}(s) \frac{\eta_t(s,a) \zeta_t(s,a)}{q_t(s)^2} + \sum_{t \in \mathcal{T}_v} \sum_{s,a} q^{\pi_t}(s) \frac{\mathbb{I}\{(s_t^\dagger, a_t^\dagger) = (s,a)\}}{\eta_t(s,a) H} + \sum_{t=1}^{T} \sum_s \gamma_t H$$

$$\le \sum_{t \in \mathcal{T}_r} \sum_{s,a} \frac{\eta_t(s,a) \zeta_t(s,a)}{q_t(s)} + \sum_{t \in \mathcal{T}_v} \frac{q^{\pi_t}(s_t^\dagger)}{\eta_t(s_t^\dagger, a_t^\dagger) H} + HS \sum_{t=1}^{T} \gamma_t$$

$$\lesssim \sum_{t \in \mathcal{T}_r} \sum_{s,a} \frac{\eta_t(s,a) \zeta_t(s,a)}{q_t(s)} + \sum_{t \in \mathcal{T}_v} \sqrt{HS} + H^{\frac{3}{2}} S^{\frac{3}{2}} \log(T) \qquad \left(\text{by } \frac{\eta_t(s_t^\dagger, a_t^\dagger)}{q_t(s_t^\dagger)} > \frac{1}{18\sqrt{H^3 S}} \text{ in virtual episodes}\right)$$

$$\lesssim \sqrt{\log(T)} \sum_{t \in \mathcal{T}_r} \sum_{s,a} \frac{\frac{\sqrt{\zeta_t(s,a)}}{q_t(s)} \times \sqrt{\zeta_t(s,a)}}{\sqrt{\sum_{\tau \le t: \tau \in \mathcal{T}_r} \frac{\zeta_\tau(s,a)}{q_\tau(s)^2}}} + \sqrt{HS}|\mathcal{T}_v| + H^{\frac{3}{2}} S^{\frac{3}{2}} \log(T) \qquad \text{(by Lemma E.5)}$$

$$\le \sqrt{\log(T)} \sum_{s,a} \sqrt{\sum_{t \in \mathcal{T}_r} \frac{\frac{\zeta_t(s,a)}{q_t(s)^2}}{\sum_{\tau \le t: \tau \in \mathcal{T}_r} \frac{\zeta_\tau(s,a)}{q_\tau(s)^2}}} \sqrt{\sum_{t \in \mathcal{T}_r} \zeta_t(s,a)} + H^{\frac{3}{2}} S^{\frac{3}{2}} A \log^2(T) + H^{\frac{3}{2}} S^{\frac{3}{2}} \log(T)$$

$$\text{(by the Cauchy–Schwarz inequality and Lemma E.6)}$$

$$\lesssim \log(T) \sum_{s,a} \sqrt{\sum_{t \in \mathcal{T}_r} \zeta_t(s,a)} + H^{\frac{3}{2}} S^{\frac{3}{2}} A \log^2(T),$$

where the last inequality follows from

$$\sqrt{\sum_{t \in \mathcal{T}_r} \frac{\frac{\zeta_t(s,a)}{q_t(s)^2}}{\sum_{\tau \le t: \tau \in \mathcal{T}_r} \frac{\zeta_\tau(s,a)}{q_\tau(s)^2}}} \le \sqrt{1 + \log\left(\sum_{\tau \in \mathcal{T}_r} \frac{\zeta_\tau(s,a)}{q_\tau(s)^2}\right)} \le \sqrt{1 + \log\left(\sum_{\tau \in \mathcal{T}_r} \frac{H^2 T}{H}\right)} \lesssim \sqrt{\log(T)}.$$

Combining the above arguments with $H \le S$, Lemmas E.12 and E.13, we obtain

$$\text{Reg}_T \lesssim \sum_{s,a} \sqrt{\log^2(T) \mathbb{E}\left[\sum_{t=1}^T \zeta_t(s,a)\right]} + H^{\frac{5}{2}} S^{\frac{3}{2}} A \log^2(T). \tag{76}$$

$\square$

### E.3. Proof of Theorem 5.2

Here we provide the proof of Theorem 5.2.

**Theorem E.14** (Restatement of Theorem 5.2)**.** *Algorithm 2 with the loss prediction $m_t$ defined in (2) guarantees*

$$\text{Reg}_T \lesssim \sqrt{H^2 SA \log^2(T) \min\{L^\star, HT - L^\star, Q_\infty, V_1\}} + H^{\frac{5}{2}} S^{\frac{3}{2}} A \log^2(T).$$

*Under the stochastic regime with adversarial corruption, it simultaneously ensures*

$$\text{Reg}_T \lesssim \sqrt{H^2 SA \log^2(T)(\mathbb{V}T + \mathcal{C})} + H^{\frac{5}{2}} S^{\frac{3}{2}} A \log^2(T),$$

*and*

$$\text{Reg}_T \lesssim U + \sqrt{U\mathcal{C}} + H^{\frac{5}{2}} S^{\frac{3}{2}} A \log^2(T),$$

*where $U = \sum_s \sum_{a \ne \pi^\star(s)} \frac{H^2 \log^2(T)}{\Delta(s,a)}$.*

*Proof.* We start from (76),

$$\text{Reg}_T \lesssim \sum_{s,a} \sqrt{\log^2(T) \mathbb{E}\left[\sum_{t=1}^T \zeta_t(s,a)\right]} + H^{\frac{5}{2}} S^{\frac{3}{2}} A \log^2(T). \tag{77}$$

By the definition of $\zeta_t(s, a)$, we have

$$\sum_{s,a} \sqrt{\log^2(T)\mathbb{E}\left[\sum_{t=1}^{T} \zeta_t(s, a)\right]}$$

$$= \sum_{s,a} \sqrt{\log^2(T)\mathbb{E}\left[\sum_{t=1}^{T} (\mathbb{I}_t(s, a) - \pi_t(a \mid s)\mathbb{I}_t(s))^2 (L_{t,h(s)} - M_{t,h(s)})^2\right]}$$

$$\leq \sqrt{SA\log^2(T)\mathbb{E}\left[\sum_{t=1}^{T} \sum_{s,a} (\mathbb{I}_t(s, a) - \pi_t(a \mid s)\mathbb{I}_t(s))^2 (L_{t,h(s)} - M_{t,h(s)})^2\right]} \quad \text{(by the Cauchy–Schwarz inequality)}$$

$$\leq \sqrt{SA\log^2(T)\mathbb{E}\left[\sum_{t\in\mathcal{T}_r} \sum_{s,a} 2\mathbb{I}_t(s, a)(L_{t,h(s)} - M_{t,h(s)})^2\right]}$$

$$\lesssim \sqrt{H^2SA\log^2(T)\mathbb{E}\left[\sum_{t\in\mathcal{T}_r} \sum_{s,a} \mathbb{I}_t(s, a)(\ell_t(s, a) - m_t(s, a))^2\right]}, \tag{78}$$

where the third line uses $\sum_a (\mathbb{I}_t(s, a) - \pi_t(a \mid s)\mathbb{I}_t(s))^2 \leq 2\mathbb{I}_t(s)$ for each fixed state $s$, and the last inequality follows from Lemma E.9.

**1. Bounds for the adversarial regime.** Applying Lemma F.12 to (78) yields

$$\sqrt{H^2SA\log^2(T)\mathbb{E}\left[\sum_{t=1}^{T} \sum_{s,a} \mathbb{I}_t(s, a)(\ell_t(s, a) - m_t(s, a))^2\right]}$$

$$\lesssim \sqrt{H^2SA\log^2(T)(\min\{L^\star + \mathrm{Reg}_T, HT - L^\star - \mathrm{Reg}_T, Q_\infty, V_1\} + SA)}$$

$$\leq \sqrt{H^2SA\log^2(T)\min\{L^\star + \mathrm{Reg}_T, HT - L^\star - \mathrm{Reg}_T, Q_\infty, V_1\}} + HSA\log(T).$$

Then, we obtain

$$\mathrm{Reg}_T \lesssim \sqrt{H^2SA\log^2(T)(L^\star + \mathrm{Reg}_T)} + H^{\frac{5}{2}}S^{\frac{3}{2}}A\log^2(T), \tag{79}$$

$$\mathrm{Reg}_T \lesssim \sqrt{H^2SA\log^2(T)(HT - L^\star - \mathrm{Reg}_T)} + H^{\frac{5}{2}}S^{\frac{3}{2}}A\log^2(T), \tag{80}$$

$$\mathrm{Reg}_T \lesssim \sqrt{H^2SA\log^2(T)Q_\infty} + H^{\frac{5}{2}}S^{\frac{3}{2}}A\log^2(T), \tag{81}$$

$$\mathrm{Reg}_T \lesssim \sqrt{H^2SA\log^2(T)V_1} + H^{\frac{5}{2}}S^{\frac{3}{2}}A\log^2(T). \tag{82}$$

From (79),

$$\mathrm{Reg}_T \leq c\sqrt{H^2SA\log^2(T)L^\star} + c\sqrt{H^2SA\log^2(T)\mathrm{Reg}_T} + cH^{\frac{5}{2}}S^{\frac{3}{2}}A\log^2(T) \quad \text{(for some absolute constant } c\text{)}$$

$$\leq c\sqrt{H^2SA\log^2(T)L^\star} + \frac{c^2}{2}H^2SA\log^2(T) + \frac{1}{2}\mathrm{Reg}_T + cH^{\frac{5}{2}}S^{\frac{3}{2}}A\log^2(T)$$

$$\leq \frac{1}{2}\mathrm{Reg}_T + O\left(\sqrt{H^2SA\log^2(T)L^\star} + H^{\frac{5}{2}}S^{\frac{3}{2}}A\log^2(T)\right),$$

where the second line follows from the AM–GM inequality. Therefore,

$$\mathrm{Reg}_T \lesssim \sqrt{H^2SA\log^2(T)L^\star} + H^{\frac{5}{2}}S^{\frac{3}{2}}A\log^2(T). \tag{83}$$

From (80), we also have

$$\text{Reg}_T \lesssim \sqrt{H^2 SA \log^2(T)(HT - L^\star - \text{Reg}_T)} + H^{\frac{5}{2}} S^{\frac{3}{2}} A \log^2(T)$$

$$\leq \sqrt{H^2 SA \log^2(T)(HT - L^\star)} + H^{\frac{5}{2}} S^{\frac{3}{2}} A \log^2(T). \tag{84}$$

Combining (81)–(84), we obtain

$$\text{Reg}_T \lesssim \sqrt{H^2 SA \log^2(T) \min\{L^\star, HT - L^\star, Q_\infty, V_1\}} + H^{\frac{5}{2}} S^{\frac{3}{2}} A \log^2(T).$$

**2. Stochastic variance bound.**   Under the stochastic regime, Lemma F.12 further implies

$$\text{Reg}_T \lesssim \sqrt{H^2 SA \log^2(T)(\mathbb{V}T + \mathcal{C})} + H^{\frac{5}{2}} S^{\frac{3}{2}} A \log^2(T).$$

**3. Stochastic gap-dependent bound.**   Moreover, we have

$$\sum_{s,a} \sqrt{\log^2(T)\mathbb{E}\left[\sum_{t=1}^{T} \zeta_t(s,a)\right]}$$

$$= \sum_{s,a} \sqrt{\log^2(T)\mathbb{E}\left[\sum_{t=1}^{T} (\mathbb{I}_t(s,a) - \pi_t(a \mid s)\mathbb{I}_t(s))^2 (L_{t,h(s)} - M_{t,h(s)})^2\right]}$$

$$\leq H \sum_{s,a} \sqrt{\log^2(T)\mathbb{E}\left[\sum_{t=1}^{T} (\mathbb{I}_t(s,a) - \pi_t(a \mid s)\mathbb{I}_t(s))^2\right]}$$

$$\leq H \sum_{s,a} \sqrt{\log^2(T)\mathbb{E}\left[\sum_{t=1}^{T} q^{\pi_t}(s)\pi_t(a \mid s)(1 - \pi_t(a \mid s))\right]}$$

$$\leq 2H \sum_{s} \sum_{a \neq \pi^\star(s)} \sqrt{\log^2(T)\mathbb{E}\left[\sum_{t=1}^{T} q^{\pi_t}(s,a)\right]}$$

Therefore, together with (77), we obtain

$$\text{Reg}_T \lesssim H \sum_{s} \sum_{a \neq \pi^\star(s)} \sqrt{\log^2(T)\mathbb{E}\left[\sum_{t=1}^{T} q^{\pi_t}(s,a)\right]} + H^{\frac{5}{2}} S^{\frac{3}{2}} A \log^2(T).$$

Finally, applying Lemma F.15 yields

$$\text{Reg}_T \lesssim U + \sqrt{UC} + H^{\frac{5}{2}} S^{\frac{3}{2}} A \log^2(T),$$

where $U = \sum_s \sum_{a \neq \pi^\star(s)} \frac{H^2 \log^2(T)}{\Delta(s,a)}$. □

### E.4. Proof of Theorem 5.3

Here we provide the proof of Theorem 5.3.

**Theorem E.15** (Restatement of Theorem 5.3). *Algorithm 2 with the loss prediction $m_t$ defined in (3) guarantees*

$$\text{Reg}_T \lesssim \sqrt{H^2 SA \log^2(T) \min\{L^\star, HT - L^\star, Q_\infty\}} + H^{\frac{5}{2}} S^{\frac{3}{2}} A \log^2(T).$$

*Under the stochastic regime with adversarial corruption, it simultaneously ensures*

$$\text{Reg}_T \lesssim \sqrt{H^2 SA \log^2(T)(\mathbb{V}T + \mathcal{C})} + H^{\frac{5}{2}} S^{\frac{3}{2}} A \log^2(T),$$

*and*

$$\text{Reg}_T \lesssim U_{\text{Var}} + \sqrt{U_{\text{Var}}\mathcal{C}} + \sqrt{HS^2 A^2 \mathcal{C}} \log^{\frac{3}{2}}(T) + H^{\frac{1}{2}} S^{\frac{3}{2}} A(H^2 + A^{\frac{1}{2}}) \log^2(T),$$

*where* $U_{\text{Var}} = \sum_s \sum_{a \neq \pi^\star(s)} \frac{H\mathbb{V}^c(s) \log^2(T)}{\Delta(s,a)}$.

*Proof.* The proof follows the same template as Theorem 5.2. The main differences are that we do not derive a path-length bound, and the stochastic gap-dependent bound is variance-aware.

We start from (76), which gives

$$\text{Reg}_T \lesssim \sum_{s,a} \sqrt{\log^2(T)\mathbb{E}\left[\sum_{t=1}^T \zeta_t(s,a)\right]} + H^{\frac{5}{2}} S^{\frac{3}{2}} A \log^2(T).$$

By the definition of $\zeta_t(s,a)$, the same argument as in (78) yields

$$\sum_{s,a} \sqrt{\log^2(T)\mathbb{E}\left[\sum_{t=1}^T \zeta_t(s,a)\right]} \lesssim \sqrt{H^2 SA \log^2(T)\mathbb{E}\left[\sum_{t=1}^T \sum_{s,a} \mathbb{I}_t(s,a)(\ell_t(s,a) - m_t(s,a))^2\right]}. \tag{85}$$

**1. Bounds for the adversarial regime.** Applying Lemma F.13 to (85) gives

$$\sqrt{H^2 SA \log^2(T)\mathbb{E}\left[\sum_{t=1}^T \sum_{s,a} \mathbb{I}_t(s,a)(\ell_t(s,a) - m_t(s,a))^2\right]}$$

$$\lesssim \sqrt{H^2 SA \log^2(T)(\min\{L^\star + \text{Reg}_T, HT - L^\star - \text{Reg}_T, Q_\infty\} + SA \log(T) + SA)}$$

$$\lesssim \sqrt{H^2 SA \log^2(T) \min\{L^\star + \text{Reg}_T, HT - L^\star - \text{Reg}_T, Q_\infty\}} + HSA \log^{\frac{3}{2}}(T).$$

Then, we obtain

$$\text{Reg}_T \lesssim \sqrt{H^2 SA \log^2(T)(L^\star + \text{Reg}_T)} + H^{\frac{5}{2}} S^{\frac{3}{2}} A \log^2(T),$$

$$\text{Reg}_T \lesssim \sqrt{H^2 SA \log^2(T)(HT - L^\star - \text{Reg}_T)} + H^{\frac{5}{2}} S^{\frac{3}{2}} A \log^2(T),$$

$$\text{Reg}_T \lesssim \sqrt{H^2 SA \log^2(T)Q_\infty} + H^{\frac{5}{2}} S^{\frac{3}{2}} A \log^2(T).$$

Applying the same calculation as in (79)–(81) gives

$$\text{Reg}_T \lesssim \sqrt{H^2 SA \log^2(T) \min\{L^\star, HT - L^\star, Q_\infty\}} + H^{\frac{5}{2}} S^{\frac{3}{2}} A \log^2(T).$$

**2. Stochastic variance bound.** Under the stochastic regime, Lemma F.13 further implies

$$\text{Reg}_T \lesssim \sqrt{H^2 SA \log^2(T)(\mathbb{V}T + \mathcal{C})} + H^{\frac{5}{2}} S^{\frac{3}{2}} A \log^2(T).$$

**3. Stochastic gap-dependent bound.** Moreover, by using Lemma E.7 and Lemma F.8, we have

$$
\sum_{s,a} \sqrt{\log^2(T)\mathbb{E}\left[\sum_{t=1}^{T} \zeta_t(s,a)\right]} \tag{86}
$$

$$
\leq \sum_{s,a} \sqrt{\log^2(T)\mathbb{E}\left[\sum_{t=1}^{T} 2H\mathbb{V}^c(s)q^{\pi_t}(s,a)(1-\pi_t(a\mid s))\right]} \tag{87}
$$

$$
+ \sum_{s,a} \sqrt{\log^2(T)\mathbb{E}\left[\sum_{t=1}^{T} 2H\sum_{s',a'} q^{\pi_t}(s',a')\mathbb{E}_t[(\ell_t(s',a')-\ell'_t(s',a')+\mu(s',a')-m_t(s',a'))^2]\right]} \quad \text{(by Lemma E.7)}
$$

$$
\lesssim \sum_{s,a} \sqrt{\log^2(T)\mathbb{E}\left[\sum_{t=1}^{T} H\mathbb{V}^c(s)q^{\pi_t}(s,a)(1-\pi_t(a\mid s))\right]} + \sum_{s,a} \sqrt{H\log^2(T)\left(SA\log^2(T)+\mathcal{C}\log(T)\right)}
$$

$$
\text{(by Lemma F.8)}
$$

$$
\leq \sum_{s,a} \sqrt{\log^2(T)\mathbb{E}\left[\sum_{t=1}^{T} H\mathbb{V}^c(s)q^{\pi_t}(s,a)(1-\pi_t(a\mid s))\right]} + \sqrt{HS^2A^2\mathcal{C}}\log^{\frac{3}{2}}(T) + H^{\frac{1}{2}}S^{\frac{3}{2}}A^{\frac{3}{2}}\log^2(T).
$$

$$
\leq \sum_{s} 2\sqrt{H\mathbb{V}^c(s)} \sum_{a\neq\pi^\star(s)} \sqrt{\log^2(T)\mathbb{E}\left[\sum_{t=1}^{T} q^{\pi_t}(s,a)\right]} + \sqrt{HS^2A^2\mathcal{C}}\log^{\frac{3}{2}}(T) + H^{\frac{1}{2}}S^{\frac{3}{2}}A^{\frac{3}{2}}\log^2(T),
$$

where the last inequality follows by the same argument as in (61).

Therefore, together with (77), we obtain

$$
\mathrm{Reg}_T \lesssim \sum_{s} \sqrt{H\mathbb{V}^c(s)} \sum_{a\neq\pi^\star(s)} \sqrt{\log^2(T)\mathbb{E}\left[\sum_{t=1}^{T} q^{\pi_t}(s,a)\right]} + \sqrt{HS^2A^2\mathcal{C}}\log^{\frac{3}{2}}(T) + H^{\frac{1}{2}}S^{\frac{3}{2}}A(H^2+A^{\frac{1}{2}})\log^2(T).
$$

Finally, applying Lemma F.15 yields

$$
\mathrm{Reg}_T \lesssim U_{\mathsf{Var}} + \sqrt{U_{\mathsf{Var}}\mathcal{C}} + \sqrt{HS^2A^2\mathcal{C}}\log^{\frac{3}{2}}(T) + H^{\frac{1}{2}}S^{\frac{3}{2}}A(H^2+A^{\frac{1}{2}})\log^2(T),
$$

where $U_{\mathsf{Var}} = \sum_s \sum_{a\neq\pi^\star(s)} \frac{H\mathbb{V}^c(s)\log^2(T)}{\Delta(s,a)}$. $\qquad\square$

*Remark* E.16 (Restatement of Remark 5.4). In the stochastic regime with adversarial corruption, suppose that the uncorrupted losses are generated independently and are uncorrelated across layers, Theorem 5.3 improves by a factor of $H$ to $U_{\mathsf{Var}} = \sum_s \sum_{a\neq\pi^\star(s)} \frac{\mathbb{V}^c(s)\log^2(T)}{\Delta(s,a)}$.

*Proof.* In the proof of Theorem 5.3, applying Corollary E.8 to (86) yields the following inequality in place of (87):

$$
\sum_{s,a} \sqrt{\log^2(T)\mathbb{E}\left[\sum_{t=1}^{T} \zeta_t(s,a)\right]}
$$

$$
\leq \sum_{s,a} \sqrt{\log^2(T)\mathbb{E}\left[\sum_{t=1}^{T} 2\mathbb{V}^c(s)q^{\pi_t}(s,a)(1-\pi_t(a\mid s))\right]}
$$

$$
+ \sum_{s,a} \sqrt{\log^2(T)\mathbb{E}\left[\sum_{t=1}^{T} 2H\sum_{s',a'} q^{\pi_t}(s',a')\mathbb{E}_t[(\ell_t(s',a')-\ell'_t(s',a')+\mu(s',a')-m_t(s',a'))^2]\right]}
$$

Compared to (87), this bound is improved by a factor of $H$, and can be interpreted as replacing the $H\mathbb{V}^c(s)$ term by $\mathbb{V}^c(s)$. The remainder of the proof follows by the same steps as in Theorem 5.3. $\qquad\square$

# F. Auxiliary Lemmas

This section provides auxiliary lemmas used in Appendices D and E.

## F.1. Concentration Bounds in the Stochastic Regime

This section establishes the key properties of the loss prediction (3) under the stochastic regime with adversarial corruption. In particular, we choose $m_t$ to be the empirical mean of the observed losses as follows:

$$m_t(s,a) = \frac{\sum_{\tau=1}^{t-1} \mathbb{I}_\tau(s,a)\, \ell_\tau(s,a)}{\max\{1, N_{t-1}(s,a)\}}.$$

Here, $N_{t-1}(s,a) = \sum_{\tau=1}^{t-1} \mathbb{I}_\tau(s,a)$ denotes the number of visits to the state-action pair $(s,a)$ up to episode $t-1$. For the analysis, we also introduce the following corresponding empirical mean computed from the uncorrupted losses $\ell'$.

$$m'_t(s,a) = \frac{\sum_{\tau=1}^{t-1} \mathbb{I}_\tau(s,a)\, \ell'_\tau(s,a)}{\max\{1, N_{t-1}(s,a)\}}. \tag{88}$$

**Lemma F.1** (Bennett's inequality, Maurer & Pontil 2009, Theorem 3). *Let $X_1, X_2, \ldots, X_n$ be i.i.d. random variables with values in $[0,1]$. Then, with probability at least $1 - 2\delta$, it holds that*

$$\left| \mathbb{E}[X_1] - \frac{1}{n}\sum_{t=1}^n X_i \right| \leq \sqrt{\frac{2\mathsf{Var}(X_1)\log(1/\delta)}{n}} + \frac{\log(1/\delta)}{3n},$$

*where $\mathsf{Var}(X_1)$ is the variance of $X_1$.*

**Lemma F.2.** *We have with probability at least $1 - 2\delta$,*

$$|\mu(s,a) - m'_t(s,a)| \leq \sqrt{\frac{2\sigma^2(s,a)\log(SAT/\delta)}{\max\{1, N_{t-1}(s,a)\}}} + \frac{\log(SAT/\delta)}{3\max\{1, N_{t-1}(s,a)\}}$$

*for all state-action pairs $(s,a)$ and $t \leq T$.*

*Proof.* Apply Lemma F.1 with $\delta' = \delta/(SAT)$ and take a union bound over all state-action pairs $(s,a) \in \mathcal{S} \times \mathcal{A}$ and all $t \leq T$, which completes the proof. $\square$

**Definition F.3.** Define $\mathcal{E}$ to be the event that Lemma F.2 holds. In this case, $\Pr(\mathcal{E}) \geq 1 - 2\delta$.

**Lemma F.4.** *On the event $\mathcal{E}$ defined in Definition F.3, it holds that*

$$(\mu(s,a) - m'_t(s,a))^2 \leq \frac{2\log(SAT/\delta)}{\max\{1, N_{t-1}(s,a)\}}$$

*for all state-action pairs $(s,a)$ and $t \leq T$.*

*Proof.* Let $x = \sqrt{\frac{\log(SAT/\delta)}{\max\{1, N_{t-1}(s,a)\}}}$. Since $\sigma^2(s,a) \leq \frac{1}{4}$, Lemma F.2 implies

$$|\mu(s,a) - m'_t(s,a)| \leq x + \frac{x^2}{3}.$$

Hence,

$$(\mu(s,a) - m'_t(s,a))^2 \leq \left(x + \frac{x^2}{3}\right)^2 \leq x^2\left(\frac{1}{9}x^2 + \frac{2}{3}x + 1\right).$$

Moreover, by using $(\mu(s,a) - m_t'(s,a))^2 \leq 1$, we have

$$(\mu(s,a) - m_t'(s,a))^2 \leq \min\left\{x^2\left(\frac{1}{9}x^2 + \frac{2}{3}x + 1\right), 1\right\}$$

$$\leq \begin{cases} 1 & \text{if } x \geq 1 \\ 2x^2 & \text{if } x < 1 \end{cases}$$

$$\leq 2x^2,$$

which concludes the proof. $\qquad\square$

**Lemma F.5.** *It holds that*

$$\mathbb{E}\left[\sum_{t=1}^{T}\sum_{s,a}\frac{q^{\pi_t}(s,a)}{\max\{1, N_{t-1}(s,a)\}}\right] \leq SA\log(T) + 2SA.$$

*Proof.* Using $\mathbb{E}_t[\mathbb{I}_t(s,a)] = q^{\pi_t}(s,a)$, we have

$$\mathbb{E}\left[\sum_{t=1}^{T}\sum_{s,a}\frac{q^{\pi_t}(s,a)}{\max\{1, N_{t-1}(s,a)\}}\right] = \mathbb{E}\left[\sum_{t=1}^{T}\sum_{s,a}\frac{\mathbb{I}_t(s,a)}{\max\{1, N_{t-1}(s,a)\}}\right]$$

$$= \mathbb{E}\left[\sum_{s,a}\sum_{i=0}^{N_T(s,a)-1}\frac{1}{\max\{1, i\}}\right]$$

$$\leq \sum_{s,a}\left(2 + \int_{1}^{T}\frac{1}{x}\mathrm{d}x\right)$$

$$= \sum_{s,a}(2 + \log(T))$$

$$\leq SA\log(T) + 2SA.$$

$\qquad\square$

**Lemma F.6.** *It holds that*

$$\mathbb{E}\left[\sum_{t=1}^{T}\sum_{s,a}q^{\pi_t}(s,a)(\mu(s,a) - m_t'(s,a))^2\right] \lesssim SA\log^2(T).$$

*Proof.* Let $\bar{\mathcal{E}}$ denote the complement of the event $\mathcal{E}$ in Definition F.3. Then,

$$\mathbb{E}\left[\sum_{t=1}^{T}\sum_{s,a}q^{\pi_t}(s,a)(\mu(s,a) - m_t'(s,a))^2\right] = \Pr(\mathcal{E})\mathbb{E}\left[\sum_{t=1}^{T}\sum_{s,a}q^{\pi_t}(s,a)(\mu(s,a) - m_t'(s,a))^2 \,\middle|\, \mathcal{E}\right]$$

$$+ \Pr(\bar{\mathcal{E}})\mathbb{E}\left[\sum_{t=1}^{T}\sum_{s,a}q^{\pi_t}(s,a)(\mu(s,a) - m_t'(s,a))^2 \,\middle|\, \bar{\mathcal{E}}\right]$$

$$\leq \mathbb{E}\left[\sum_{t=1}^{T}\sum_{s,a}q^{\pi_t}(s,a)(\mu(s,a) - m_t'(s,a))^2 \,\middle|\, \mathcal{E}\right] + 2HT\delta. \qquad (89)$$

On the event $\mathcal{E}$, we have

$$\mathbb{E}\left[\sum_{t=1}^{T}q^{\pi_t}(s,a)(\mu(s,a) - m_t'(s,a))^2 \,\middle|\, \mathcal{E}\right] \leq \mathbb{E}\left[\sum_{t=1}^{T}\sum_{s,a}q^{\pi_t}(s,a)\frac{2\log(SAT/\delta)}{\max\{1, N_{t-1}(s,a)\}}\right] \qquad \text{(by Lemma F.4)}$$

$$\leq 2(SA\log(T) + 2SA)\log(SAT/\delta) \qquad \text{(by Lemma F.5)}$$

$$\lesssim SA\log(T)\log(SAT/\delta).$$

Choosing $\delta = 1/T$ and combining with (89) yields

$$\mathbb{E}\left[\sum_{t=1}^{T}\sum_{s,a} q^{\pi_t}(s,a)(\mu(s,a) - m_t'(s,a))^2\right] \lesssim SA \log^2(T).$$

$\square$

**Lemma F.7.** *Suppose that $m_t$ is defined as* (3). *Then, it holds that*

$$\mathbb{E}\left[\sum_{t=1}^{T}\sum_{s,a} q^{\pi_t}(s,a)(m_t'(s,a) - m_t(s,a))^2\right] \leq \mathcal{C}\log(T) + 2\mathcal{C}.$$

*Proof.* By the definitions of $m_t$ in (3) and $m_t'$ in (88), we have

$$m_t'(s,a) - m_t(s,a) = \frac{\sum_{\tau=1}^{t-1} \mathbb{I}_\tau(s,a)(\ell_\tau'(s,a) - \ell_\tau(s,a))}{\max\{1, N_{t-1}(s,a)\}}.$$

Thus,

$$\begin{aligned}
(m_t'(s,a) - m_t(s,a))^2 &= \frac{\left(\sum_{\tau=1}^{t-1} \mathbb{I}_\tau(s,a)(\ell_\tau'(s,a) - \ell_\tau(s,a))\right)^2}{(\max\{1, N_{t-1}(s,a)\})^2} \\
&\leq \frac{N_{t-1}(s,a) \sum_{\tau=1}^{t-1} \mathbb{I}_\tau(s,a)(\ell_\tau'(s,a) - \ell_\tau(s,a))^2}{(\max\{1, N_{t-1}(s,a)\})^2} \\
&\leq \frac{\sum_{\tau=1}^{t-1} \mathbb{I}_\tau(s,a)(\ell_\tau'(s,a) - \ell_\tau(s,a))^2}{\max\{1, N_{t-1}(s,a)\}},
\end{aligned} \qquad (90)$$

where the first inequality follows from the Cauchy–Schwarz inequality.

From (90), we obtain

$$\begin{aligned}
&\mathbb{E}\left[\sum_{t=1}^{T}\sum_{s,a} q^{\pi_t}(s,a)(m_t'(s,a) - m_t(s,a))^2\right] \\
&\leq \mathbb{E}\left[\sum_{t=1}^{T}\sum_{s,a} q^{\pi_t}(s,a)\frac{\sum_{\tau=1}^{t-1} \mathbb{I}_\tau(s,a)(\ell_\tau'(s,a) - \ell_\tau(s,a))^2}{\max\{1, N_{t-1}(s,a)\}}\right] \\
&= \mathbb{E}\left[\sum_{s,a}\sum_{\tau=1}^{T-1} \mathbb{I}_\tau(s,a)(\ell_\tau'(s,a) - \ell_\tau(s,a))^2 \sum_{t=\tau+1}^{T} \frac{\mathbb{I}_t(s,a)}{\max\{1, N_{t-1}(s,a)\}}\right] \\
&\leq (2 + \log T)\mathbb{E}\left[\sum_{\tau=1}^{T}\sum_{s,a} \mathbb{I}_\tau(s,a)|\ell_\tau'(s,a) - \ell_\tau(s,a)|\right] \\
&\leq (2 + \log T)\mathcal{C},
\end{aligned} \qquad (91)$$

where (91) uses $-1 \leq \ell_\tau'(s,a) - \ell_\tau(s,a) \leq 1$ and

$$\sum_{t=\tau+1}^{T} \frac{\mathbb{I}_t(s,a)}{\max\{1, N_{t-1}(s,a)\}} \leq \sum_{i=0}^{N_T(s,a)-1} \frac{1}{\max\{1, i\}} \leq 2 + \int_1^T \frac{1}{x}\mathrm{d}x = 2 + \log(T).$$

The last inequality follows from the definition of the corruption budget $\mathbb{E}\left[\sum_{\tau=1}^{T}\sum_{s,a} \mathbb{I}_\tau(s,a)|\ell_\tau'(s,a) - \ell_\tau(s,a)|\right] \leq \mathcal{C}$, which completes the proof. $\square$

**Lemma F.8.** *Suppose $m_t$ is defined in* (3). *It holds that*

$$\mathbb{E}\left[\sum_{t=1}^{T}\sum_{s,a} q^{\pi_t}(s,a)(\ell_t(s,a) - \ell'_t(s,a) + \mu(s,a) - m_t(s,a))^2\right] \lesssim SA\log^2(T) + \mathcal{C}\log(T).$$

*Proof.* It holds that

$$\mathbb{E}\left[\sum_{t=1}^{T}\sum_{s,a} q^{\pi_t}(s,a)(\ell_t(s,a) - \ell'_t(s,a) + \mu(s,a) - m_t(s,a))^2\right]$$

$$= \mathbb{E}\left[\sum_{t=1}^{T}\sum_{s,a} q^{\pi_t}(s,a)((\ell_t(s,a) - \ell'_t(s,a)) + (\mu(s,a) - m'_t(s,a)) + (m'_t(s,a) - m_t(s,a)))^2\right]$$

$$\leq 3\mathbb{E}\left[\sum_{t=1}^{T}\sum_{s,a} q^{\pi_t}(s,a)(\ell_t(s,a) - \ell'_t(s,a))^2\right] + 3\mathbb{E}\left[\sum_{t=1}^{T}\sum_{s,a} q^{\pi_t}(s,a)(\mu(s,a) - m'_t(s,a))^2\right]$$

$$+ 3\mathbb{E}\left[\sum_{t=1}^{T}\sum_{s,a} q^{\pi_t}(s,a)(m'_t(s,a) - m_t(s,a))^2\right] \qquad \text{(by } (x+y+z)^2 \leq 3(x^2+y^2+z^2) \text{ for } x,y,z \in \mathbb{R})$$

$$\leq 3\mathbb{E}\left[\sum_{t=1}^{T}\sum_{s,a} q^{\pi_t}(s,a)|\ell_t(s,a) - \ell'_t(s,a)|\right]$$

$$+ 3\mathbb{E}\left[\sum_{t=1}^{T}\sum_{s,a} q^{\pi_t}(s,a)(\mu(s,a) - m'_t(s,a))^2\right] + 3\mathbb{E}\left[\sum_{t=1}^{T}\sum_{s,a} q^{\pi_t}(s,a)(m'_t(s,a) - m_t(s,a))^2\right].$$

The first term is bounded by the corruption budget as

$$\mathbb{E}\left[\sum_{t=1}^{T}\sum_{s,a} q^{\pi_t}(s,a)|\ell_t(s,a) - \ell'_t(s,a)|\right] \leq \mathcal{C}.$$

The second term is bounded by Lemma F.6 and the third term is bounded by Lemma F.7. Combining these bounds, we have

$$\mathbb{E}\left[\sum_{t=1}^{T}\sum_{s,a} q^{\pi_t}(s,a)(\ell_t(s,a) - \ell'_t(s,a) + \mu(s,a) - m_t(s,a))^2\right] \lesssim \mathcal{C} + SA\log^2(T) + \mathcal{C}\log(T),$$

which completes the proof. □

### F.2. General Lemmas for Data-Dependent and Best-of-Both-Worlds Bounds

In this section, we present general tools for deriving data-dependent bounds and for establishing self-bounding inequalities, which together yield best-of-both-worlds guarantees.

The first lemma is a standard tool for deriving path-length bounds when $m_t$ is updated as in (2). It appears in Ito (2021, Proposition 13) and Tsuchiya et al. (2023, Lemma 7). Here, we extend it to the MDP setting.

**Lemma F.9.** *Suppose $m_t$ is defined in* (2). *Then, for any sequence $m_t^* \in [0,1]^{S \times A}$ and any $\xi \in \left(0, \frac{1}{2}\right)$, we have*

$$\sum_{t=1}^{T}\sum_{s,a} \mathbb{I}_t(s,a)(\ell_t(s,a) - m_t(s,a))^2$$

$$\leq \frac{1}{1-2\xi}\sum_{t=1}^{T}\sum_{s,a} \mathbb{I}_t(s,a)(\ell_t(s,a) - m_t^*(s,a))^2 + \frac{1}{\xi(1-2\xi)}\left(\frac{SA}{4} + 2\sum_{t=1}^{T-1}\left\|m_{t+1}^* - m_t^*\right\|_1\right).$$

*Proof.* Fix any $(s, a)$. For episodes $t$ with $\mathbb{I}_t(s, a) = 1$, the update rule of $m_t$ (2) implies that

$$(\ell_t(s, a) - m_t(s, a))^2 - (\ell_t(s, a) - m_t^*(s, a))^2$$

$$\leq 2(\ell_t(s, a) - m_t(s, a))(m_t^*(s, a) - m_t(s, a))$$

$$= 2(\ell_t(s, a) - m_t(s, a))(m_{t+1}(s, a) - m_t(s, a) + m_t^*(s, a) - m_{t+1}(s, a))$$

$$= 2\xi(\ell_t(s, a) - m_t(s, a))^2 + \frac{2}{\xi}(m_{t+1}(s, a) - m_t(s, a))(m_t^*(s, a) - m_{t+1}(s, a))$$

$$\leq 2\xi(\ell_t(s, a) - m_t(s, a))^2 + \frac{1}{\xi}((m_t^*(s, a) - m_t(s, a))^2 - (m_t^*(s, a) - m_{t+1}(s, a))^2),$$

where the inequalities follow from $x^2 - y^2 = 2x(x - y) - (x - y)^2 \leq 2x(x - y)$ for $x, y \in \mathbb{R}$. Hence, we have

$$(\ell_t(s, a) - m_t(s, a))^2 \leq \frac{1}{1 - 2\xi}(\ell_t(s, a) - m_t^*(s, a))^2$$

$$+ \frac{1}{\xi(1 - 2\xi)}((m_t^*(s, a) - m_t(s, a))^2 - (m_t^*(s, a) - m_{t+1}(s, a))^2).$$

Then, for any $s, a$ and for $m_t = m_{t+1}$ when $\mathbb{I}_t(s, a) = 0$, we obtain

$$\sum_{t=1}^{T} \mathbb{I}_t(s, a)(\ell_t(s, a) - m_t(s, a))^2$$

$$\leq \frac{1}{1 - 2\xi} \sum_{t=1}^{T} \mathbb{I}_t(s, a)(\ell_t(s, a) - m_t^*(s, a))^2$$

$$+ \frac{1}{\xi(1 - 2\xi)} \sum_{t=1}^{T} ((m_t^*(s, a) - m_t(s, a))^2 - (m_t^*(s, a) - m_{t+1}(s, a))^2)$$

$$= \frac{1}{1 - 2\xi} \sum_{t=1}^{T} \mathbb{I}_t(s, a)(\ell_t(s, a) - m_t^*(s, a))^2$$

$$+ \frac{1}{\xi(1 - 2\xi)} \left\{ \sum_{t=1}^{T-1} ((m_{t+1}^*(s, a) - m_{t+1}(s, a))^2 - (m_t^*(s, a) - m_{t+1}(s, a))^2) + (m_1^*(s, a) - m_1(s, a))^2 \right\}$$

$$\leq \frac{1}{1 - 2\xi} \sum_{t=1}^{T} \mathbb{I}_t(s, a)(\ell_t(s, a) - m_t^*(s, a))^2$$

$$+ \frac{1}{\xi(1 - 2\xi)} \left\{ \sum_{t=1}^{T-1} (m_{t+1}^*(s, a) + m_t^*(s, a) - 2m_{t+1}(s, a))(m_{t+1}^*(s, a) - m_t^*(s, a)) + \frac{1}{4} \right\}$$

$$\leq \frac{1}{1 - 2\xi} \sum_{t=1}^{T} \mathbb{I}_t(s, a)(\ell_t(s, a) - m_t^*(s, a))^2 + \frac{1}{\xi(1 - 2\xi)} \left\{ 2 \sum_{t=1}^{T-1} |m_{t+1}^*(s, a) - m_t^*(s, a)| + \frac{1}{4} \right\}.$$

Therefore,

$$\sum_{t=1}^{T} \sum_{s,a} \mathbb{I}_t(s, a)(\ell_t(s, a) - m_t(s, a))^2$$

$$\leq \frac{1}{1 - 2\xi} \sum_{t=1}^{T} \sum_{s,a} \mathbb{I}_t(s, a)(\ell_t(s, a) - m_t^*(s, a))^2 + \frac{1}{\xi(1 - 2\xi)} \left( \frac{SA}{4} + 2 \sum_{t=1}^{T-1} \|m_{t+1}^* - m_t^*\|_1 \right).$$

$\square$

The next two lemmas concern the loss prediction $m_t$, which is updated as in (3).

**Lemma F.10.** *Let $\ell_1, \ldots, \ell_T \in [0, 1]$ be any sequence and let $m^* \in [0, 1]$ be arbitrary. Define*

$$m_t = \frac{1}{\max\{1, t-1\}} \sum_{\tau=1}^{t-1} \ell_\tau$$

*Then, it holds that*

$$\sum_{t=1}^{T} (\ell_t - m_t)^2 \leq \sum_{t=1}^{T} (\ell_t - m^*)^2 + \log(T) + 1.$$

*Proof.* For $t \geq 2$, $m_t$ is expressed as

$$m_t \in \arg\min_{m \in \mathbb{R}} \left\{ \sum_{\tau=1}^{t-1} (m - \ell_\tau)^2 \right\}.$$

Then, since $\sum_{\tau=1}^{t-1} (m - \ell_\tau)^2$ is a quadratic function of $m$, for any $m \in \mathbb{R}$,

$$\sum_{\tau=1}^{t-1} (m - \ell_\tau)^2 = \sum_{\tau=1}^{t-1} (m_t - \ell_\tau)^2 + (t-1)(m - m_t)^2. \tag{92}$$

Thus, by applying (92), we obtain

$$\sum_{t=1}^{T} (\ell_t - m^*)^2 = \sum_{t=1}^{T} (m_{T+1} - \ell_t)^2 + T(m^* - m_{T+1})^2 \qquad \text{(by (92))}$$

$$\geq \sum_{t=1}^{T} (m_{T+1} - \ell_t)^2$$

$$= \sum_{t=1}^{T-1} (m_{T+1} - \ell_t)^2 + (m_{T+1} - \ell_T)^2$$

$$= \sum_{t=1}^{T-1} (m_T - \ell_t)^2 + (T-1)(m_{T+1} - m_T)^2 + (m_{T+1} - \ell_T)^2 \qquad \text{(by repeatedly using (92))}$$

$$= \sum_{t=1}^{T} (t-1)(m_{t+1} - m_t)^2 + \sum_{t=1}^{T} (m_{t+1} - \ell_t)^2. \tag{93}$$

Then, we have

$$\sum_{t=1}^{T} (\ell_t - m_t)^2 - \sum_{t=1}^{T} (\ell_t - m^*)^2 \leq \sum_{t=1}^{T} (\ell_t - m_t)^2 - \sum_{t=1}^{T} (t-1)(m_{t+1} - m_t)^2 - \sum_{t=1}^{T} (m_{t+1} - \ell_t)^2 \qquad \text{(by (93))}$$

$$= \sum_{t=1}^{T} \left( (2\ell_t - m_t - m_{t+1})(m_{t+1} - m_t) - (t-1)(m_{t+1} - m_t)^2 \right)$$

$$= \sum_{t=1}^{T} \left( \frac{2t-1}{t^2} (\ell_t - m_t)^2 - \frac{t-1}{t^2} (\ell_t - m_t)^2 \right) \qquad \text{(by } m_{t+1} - m_t = \frac{1}{t}(\ell_t - m_t))$$

$$\leq \sum_{t=1}^{T} \frac{1}{t} \qquad \text{(by } (\ell_t(s,a) - m_t(s,a))^2 \leq 1)$$

$$\leq 1 + \int_1^T \frac{1}{x} \mathrm{d}x \leq 1 + \log(T),$$

which completes the proof. $\square$

**Lemma F.11.** *Suppose $m_t$ is defined in* (3). *Then, for any $m^* \in [0, 1]^{S \times A}$, we have*

$$\sum_{t=1}^{T} \sum_{s,a} \mathbb{I}_t(s, a)(\ell_t(s, a) - m_t(s, a))^2 \leq \sum_{t=1}^{T} \sum_{s,a} \mathbb{I}_t(s, a)(\ell_t(s, a) - m^*(s, a))^2 + SA \log(T) + SA.$$

*Proof.* Fix any $(s, a)$. Since $m_t$ is defined as $m_t(s, a) = \frac{\sum_{\tau=1}^{t-1} \mathbb{I}_\tau(s, a)\, \ell_\tau(s, a)}{\max\{1, N_{t-1}(s, a)\}}$, we use Lemma F.10 and obtain

$$\sum_{t=1}^{T} \mathbb{I}_t(s, a)(\ell_t(s, a) - m_t(s, a))^2 \leq \sum_{t=1}^{T} \mathbb{I}_t(s, a)(\ell_t(s, a) - m^*)^2 + \log(N_T(s, a)) + 1$$

$$\leq \sum_{t=1}^{T} \mathbb{I}_t(s, a)(\ell_t(s, a) - m^*)^2 + \log(T) + 1.$$

Summing the above inequality over all $(s, a) \in \mathcal{S} \times \mathcal{A}$ completes the proof. $\qquad\square$

The following lemma serves a variety of data-dependent bounds, and (in the stochastic regime) variance-dependent bounds.

**Lemma F.12.** *Suppose $m_t$ is defined in* (2). *Then, it holds that*

$$\mathbb{E}\left[\sum_{t=1}^{T} \sum_{s,a} \mathbb{I}_t(s, a)(\ell_t(s, a) - m_t(s, a))^2\right] \lesssim \min\{L^* + \mathrm{Reg}_T, HT - L^* - \mathrm{Reg}_T, Q_\infty, V_1\} + SA.$$

*Simultaneously, under the stochastic regime with adversarial corruption (Section 2.1), it holds that*

$$\mathbb{E}\left[\sum_{t=1}^{T} \sum_{s,a} \mathbb{I}_t(s, a)(\ell_t(s, a) - m_t(s, a))^2\right] \lesssim \mathbb{V}T + \mathcal{C} + SA.$$

*Proof.* By using Lemma F.9, for any $m_t^* \in [0, 1]^{S \times A}$ and any $\xi \in (0, \frac{1}{2})$, we obtain

$$\mathbb{E}\left[\sum_{t=1}^{T} \sum_{s,a} \mathbb{I}_t(s, a)(\ell_t(s, a) - m_t(s, a))^2\right]$$

$$\leq \underbrace{\frac{1}{1 - 2\xi} \mathbb{E}\left[\sum_{t=1}^{T} \sum_{s,a} \mathbb{I}_t(s, a)(\ell_t(s, a) - m_t^*(s, a))^2\right]}_{\textbf{term}_1} + \underbrace{\frac{1}{\xi(1 - 2\xi)}\left(\frac{SA}{4} + 2\mathbb{E}\left[\sum_{t=1}^{T-1} \left\|m_{t+1}^* - m_t^*\right\|_1\right]\right)}_{\textbf{term}_2}$$

In particular, if $m_t^*$ is time-invariant, then $\sum_{t=1}^{T-1} \left\|m_{t+1}^* - m_t^*\right\|_1 = 0$ and $\textbf{term}_2 = \dfrac{SA}{4\xi(1 - 2\xi)}$.

**1. First-order bound.** Taking $m^*(s, a) \equiv 0$, we obtain

$$\textbf{term}_1 = \frac{1}{1 - 2\xi}\mathbb{E}\left[\sum_{t=1}^{T} \sum_{s,a} \mathbb{I}_t(s, a)\ell_t(s, a)^2\right]$$

$$\leq \frac{1}{1 - 2\xi}\mathbb{E}\left[\sum_{t=1}^{T} \sum_{s,a} \mathbb{I}_t(s, a)\ell_t(s, a)\right]$$

$$= \frac{1}{1 - 2\xi}\mathbb{E}\left[\sum_{t=1}^{T} V^{\pi_t}(s_0; \ell_t)\right]$$

$$= \frac{1}{1 - 2\xi}(L^* + \mathrm{Reg}_T). \tag{94}$$

Similarly, taking $m^*(s,a) \equiv 1$ yields

$$\begin{aligned}
\mathbf{term}_1 &= \frac{1}{1-2\xi}\mathbb{E}\left[\sum_{t=1}^T \sum_{s,a} \mathbb{I}_t(s,a)(\ell_t(s,a)-1)^2\right]\\
&\leq \frac{1}{1-2\xi}\mathbb{E}\left[\sum_{t=1}^T \sum_{s,a} \mathbb{I}_t(s,a)(1-\ell_t(s,a))\right]\\
&= \frac{1}{1-2\xi}\mathbb{E}\left[\sum_{t=1}^T (H - V^{\pi_t}(s_0;\ell_t))\right]\\
&= \frac{1}{1-2\xi}(HT - L^\star - \mathrm{Reg}_T).
\end{aligned}$$
(95)

**2. Second-order bound.** For any time-invariant $m^* \in [0,1]^{S\times A}$,

$$\begin{aligned}
\mathbf{term}_1 &= \frac{1}{1-2\xi}\mathbb{E}\left[\sum_{t=1}^T \sum_{h=0}^{H-1} \sum_{(s,a)\in\mathcal{S}_h\times\mathcal{A}} \mathbb{I}_t(s,a)(\ell_t(s,a)-m^*(s,a))^2\right]\\
&\leq \frac{1}{1-2\xi}\mathbb{E}\left[\sum_{t=1}^T \sum_{h=0}^{H-1} \|\ell_t(h)-m^*(h)\|_\infty^2\right]\\
&\leq \frac{1}{1-2\xi}Q_\infty.
\end{aligned}$$
(96)

**3. Path-length bound.** Taking $m_t^*(s,a) = \ell_t(s,a)$ yields $\mathbf{term}_1 = 0$ and

$$\mathbf{term}_2 = \frac{1}{\xi(1-2\xi)}\left(\frac{SA}{4} + 2\sum_{t=1}^{T-1}\|\ell_{t+1}-\ell_t\|_1\right) = \frac{1}{\xi(1-2\xi)}\left(\frac{SA}{4} + 2V_1\right).$$
(97)

Combining (94)–(97), we get

$$\begin{aligned}
&\mathbb{E}\left[\sum_{t=1}^T \sum_{s,a} \mathbb{I}_t(s,a)(\ell_t(s,a)-m_t(s,a))^2\right]\\
&\leq \frac{1}{1-2\xi}\min\left\{L^\star + \mathrm{Reg}_T, HT - L^\star - \mathrm{Reg}_T, Q_\infty, \frac{V_1}{\xi}\right\} + \frac{SA}{4\xi(1-2\xi)}\\
&\lesssim \min\{L^\star + \mathrm{Reg}_T, HT - L^\star - \mathrm{Reg}_T, Q_\infty, V_1\} + SA,
\end{aligned}$$

where we absorb the $\xi$-dependent constants into $\lesssim$.

**4. Stochastic variance bound.** Under the stochastic regime with adversarial corruption, recall that $\mu(s,a)$ and $\sigma^2(s,a)$ denote the mean and variance of the uncorrupted losses $\ell_t'$, respectively.

We set the predictor to the mean $m^\star \equiv \mu$, and we obtain

$$\begin{aligned}
\mathbf{term}_1 &= \frac{1}{1-2\xi}\mathbb{E}\left[\sum_{t=1}^T \sum_{s,a} \mathbb{I}_t(s,a)(\ell_t(s,a)-\mu(s,a))^2\right]\\
&= \frac{1}{1-2\xi}\mathbb{E}\left[\sum_{t=1}^T \sum_{s,a} \mathbb{I}_t(s,a)(\ell_t(s,a)-\ell_t'(s,a)+\ell_t'(s,a)-\mu(s,a))^2\right]\\
&\leq \frac{2}{1-2\xi}\mathbb{E}\left[\sum_{t=1}^T \sum_{s,a} \mathbb{I}_t(s,a)\left((\ell_t(s,a)-\ell_t'(s,a))^2 + (\ell_t'(s,a)-\mu(s,a))^2\right)\right]
\end{aligned}$$

$$\leq \frac{2}{1-2\xi}\mathbb{E}\left[\sum_{t=1}^{T}\sum_{h=0}^{H-1}\sum_{(s,a)\in\mathcal{S}_h\times\mathcal{A}}\mathbb{I}_t(s,a)|\ell_t(s,a)-\ell'_t(s,a)| + \sum_{t=1}^{T}\sum_{s,a}q^{\pi_t}(s,a)\sigma^2(s,a)\right]$$

$$= \frac{2}{1-2\xi}\mathbb{E}\left[\sum_{t=1}^{T}\sum_{h=0}^{H-1}\|\ell'_t(h)-\ell_t(h)\|_\infty + \sum_{t=1}^{T}\sum_{s,a}q^{\pi_t}(s,a)\sigma^2(s,a)\right]$$

$$= \frac{2}{1-2\xi}(\mathcal{C}+\mathbb{V}T), \tag{98}$$

Therefore, in the stochastic regime with adversarial corruption, we have

$$\mathbb{E}\left[\sum_{t=1}^{T}\sum_{s,a}\mathbb{I}_t(s,a)(\ell_t(s,a)-m_t(s,a))^2\right] \leq \frac{2}{1-2\xi}(\mathbb{V}T+\mathcal{C}) + \frac{SA}{4\xi(1-2\xi)}$$

$$\lesssim \mathbb{V}T+\mathcal{C}+SA.$$

$\square$

**Lemma F.13.** *Suppose $m_t$ is defined in* (3). *Then, it holds that*

$$\mathbb{E}\left[\sum_{t=1}^{T}\sum_{s,a}\mathbb{I}_t(s,a)(\ell_t(s,a)-m_t(s,a))^2\right] \leq \min\{L^\star+\mathrm{Reg}_T, HT-L^\star-\mathrm{Reg}_T, Q_\infty\} + SA\log(T)+SA.$$

*Simultaneously, under the stochastic regime with adversarial corruption, it holds that*

$$\mathbb{E}\left[\sum_{t=1}^{T}\sum_{s,a}\mathbb{I}_t(s,a)(\ell_t(s,a)-m_t(s,a))^2\right] \leq \mathbb{V}T+\mathcal{C}+SA\log(T)+SA.$$

*Proof.* The argument follows the same lines as Lemma F.12.

By using Lemma F.11, for any $m_t^* \in [0,1]^{S\times A}$, we obtain

$$\mathbb{E}\left[\sum_{t=1}^{T}\sum_{s,a}\mathbb{I}_t(s,a)(\ell_t(s,a)-m_t(s,a))^2\right] \leq \mathbb{E}\left[\sum_{t=1}^{T}\sum_{s,a}\mathbb{I}_t(s,a)(\ell_t(s,a)-m^*(s,a))^2\right] + SA\log(T)+SA. \tag{99}$$

**1. First-order bound.** Taking $m^*(s,a)\equiv 0$ in (99) and proceeding as in (94) yields

$$\mathbb{E}\left[\sum_{t=1}^{T}\sum_{s,a}\mathbb{I}_t(s,a)(\ell_t(s,a)-m_t(s,a))^2\right] \leq (L^\star+\mathrm{Reg}_T) + SA\log(T)+SA. \tag{100}$$

Similarly, taking $m^*(s,a)\equiv 1$ and proceeding as in (95) yields

$$\mathbb{E}\left[\sum_{t=1}^{T}\sum_{s,a}\mathbb{I}_t(s,a)(\ell_t(s,a)-m_t(s,a))^2\right] \leq (HT-L^\star-\mathrm{Reg}_T) + SA\log(T)+SA. \tag{101}$$

**2. Second-order bound.** Since (99) holds for any $m^\star$, the same argument as in (96) gives

$$\mathbb{E}\left[\sum_{t=1}^{T}\sum_{s,a}\mathbb{I}_t(s,a)(\ell_t(s,a)-m_t(s,a))^2\right] \leq Q_\infty + SA\log(T)+SA. \tag{102}$$

Combining (100)–(102), we get

$$\mathbb{E}\left[\sum_{t=1}^{T}\sum_{s,a}\mathbb{I}_t(s,a)(\ell_t(s,a)-m_t(s,a))^2\right] \leq \min\{L^\star+\mathrm{Reg}_T, HT-L^\star-\mathrm{Reg}_T, Q_\infty\} + SA\log(T)+SA.$$

**3. Stochastic variance bound.** In the stochastic regime with adversarial corruption, we take $m^\star \equiv \mu$ and proceed as in (98) to obtain

$$\mathbb{E}\left[\sum_{t=1}^{T}\sum_{s,a}\mathbb{I}_t(s,a)(\ell_t(s,a) - m_t(s,a))^2\right] \le (\mathcal{C} + \mathbb{V}T) + SA\log(T) + SA,$$

which completes the proof. $\qquad\square$

Finally, we generalize a self-bounding argument that appears in Dann et al. (2023a, Appendix H) and Jin & Luo (2020, Appendix A.1), which is useful in deriving gap-dependent bounds in the stochastic regime with adversarial corruption. We first note that in the stochastic regime with adversarial corruption, the regret is lower bounded as follows:

**Lemma F.14** (Jin et al. 2020, Section 2.1). *Under the stochastic regime with adversarial corruption, for any sequence of policies $\{\pi_t\}_{t=1}^{T}$, the regret satisfies the following $(\Delta, 2\mathcal{C}, T)$ self-bounding constraint:*

$$\mathrm{Reg}_T \ge \mathbb{E}\left[\sum_{t=1}^{T}\sum_{s}\sum_{a\neq\pi^\star(s)} q^{\pi_t}(s,a)\,\Delta(s,a)\right] - 2\mathcal{C}.$$

We now use Lemma F.14 to prove the following lemma based on the self-bounding argument.

**Lemma F.15.** *Let $G(s,a)$ be any nonnegative function and $J > 0$. Suppose that*

$$\mathrm{Reg}_T \lesssim \sum_{s}\sum_{a\neq\pi^\star(s)} G(s,a)\sqrt{\mathbb{E}\left[\sum_{t=1}^{T} q^{\pi_t}(s,a)\right]} + J.$$

*Then, under the stochastic regime with adversarial corruption, it holds that*

$$\mathrm{Reg}_T \lesssim U + \sqrt{U\mathcal{C}} + J \quad for \quad U = \sum_{s}\sum_{a\neq\pi^\star(s)} \frac{G(s,a)^2}{\Delta(s,a)}.$$

*Proof.* By Lemma F.14, in the stochastic regime with adversarial corruption it holds that $\mathrm{Reg}_T \ge \mathbb{E}\left[\sum_{t=1}^{T}\sum_{s}\sum_{a\neq\pi^\star(s)} q^{\pi_t}(s,a)\Delta(s,a)\right] - 2\mathcal{C}$. Then, for any $\alpha \in (0, 1/2]$, we obtain

$$\mathrm{Reg}_T \le c\sum_{s}\sum_{a\neq\pi^\star(s)} G(s,a)\sqrt{\mathbb{E}\left[\sum_{t=1}^{T} q^{\pi_t}(s,a)\right]} + cJ \qquad \text{(for some absolute constant } c\text{)}$$

$$\le \sum_{s}\sum_{a\neq\pi^\star(s)} G(s,a)\left(\frac{\alpha}{G(s,a)}\mathbb{E}\left[\sum_{t=1}^{T} q^{\pi_t}(s,a)\Delta(s,a)\right] + \frac{c^2 G(s,a)}{\alpha\Delta(s,a)}\right) + cJ \qquad \text{(by the AM–GM inequality)}$$

$$\le \alpha\mathbb{E}\left[\sum_{t=1}^{T}\sum_{s}\sum_{a\neq\pi^\star(s)} q^{\pi_t}(s,a)\Delta(s,a)\right] + \sum_{s}\sum_{a\neq\pi^\star(s)} \frac{c^2 G(s,a)^2}{\alpha\Delta(s,a)} + cJ$$

$$\le \alpha(\mathrm{Reg}_T + 2\mathcal{C}) + \sum_{s}\sum_{a\neq\pi^\star(s)} \frac{c^2 G(s,a)^2}{\alpha\Delta(s,a)} + cJ.$$

Choosing $\alpha = \min\left\{\frac{1}{2}, \sqrt{\frac{U}{\mathcal{C}}}\right\}$ with $U = \sum_{s}\sum_{a\neq\pi^\star(s)} \frac{G(s,a)^2}{\Delta(s,a)}$ and absorbing the $\alpha\mathrm{Reg}_T$ term yields

$$\mathrm{Reg}_T \lesssim U + \sqrt{U\mathcal{C}} + J.$$

$\qquad\square$

# G. Proofs of Regret Lower Bounds (deferred from Section 6)

In this section, we provide complete proofs of the lower bounds stated in Section 6. We first establish an information-theoretic lower bound under a convenient stochastic loss model on a layered MDP with uniform transitions (Lemma G.2). We then prove Theorems G.3–G.5 and Theorem G.6, which are detailed versions of Theorem 6.1 and Theorem 6.2, respectively.

Here, we write $\mathsf{Ber}(p)$ for the Bernoulli distribution with mean $p$ and $\mathsf{Unif}(\mathcal{A})$ for the uniform distribution over $\mathcal{A}$. We also use $\mathsf{KL}(\mathbb{P}, \mathbb{P}')$ to denote the Kullback–Leibler (KL) divergence between distributions $\mathbb{P}$ and $\mathbb{P}'$, and use $\mathsf{kl}(p, q)$ to denote the KL divergence between Bernoulli distributions with means $p$ and $q$. We also define the regret without expectation given by

$$\mathsf{R}_T(\pi) := \sum_{t=1}^{T} V^{\pi_t}(s_0; \ell_t) - \sum_{t=1}^{T} V^{\pi}(s_0; \ell_t).$$

Note that it holds that $\mathsf{Reg}_T = \max_{\pi \in \Pi} \mathbb{E}[\mathsf{R}_T(\pi)]$.

## G.1. General Regret Lower Bound for Tabular MDPs

We first state a general regret lower bound for tabular MDPs due to Zimin & Neu (2013); Tsuchiya et al. (2025b). We include the proof to make the constants and the dependence on $H$, $S$, $A$, and $T$ explicit. The hard instance constructed for this lower bound will also be used in the proof of Theorem 6.2.

**Lemma G.1** (Tsybakov 2009). *Let $p, q \in [0, 1]$. Then the KL divergence between Bernoulli distributions with parameters $p, q$ satisfies*

$$\mathsf{kl}(p, q) \le \chi^2(p, q) = \frac{(p - q)^2}{q(1 - q)}.$$

We now consider the following instance of online episodic tabular MDPs to prove a lower bound. Let $\widetilde{\mathcal{S}} = \mathcal{S} \setminus \{s_0\}$ (note that, for simplicity, we define the state space $\mathcal{S}$ to exclude the terminal state $s_H$). We assume that $(S - 1)/(H - 1)$ is an integer and that each non-initial layer has the same number of states, i.e., $S' := |\mathcal{S}_h| = (S - 1)/(H - 1)$ for all $h \ne 0, H$. The non-integral case can be treated by a standard floor/ceiling adjustment, which changes the bounds only by constant factors. We use

$$N_T(s, a) = \sum_{t=1}^{T} \frac{1}{S'} \pi_t(a \mid s)$$

to denote the expected number of times the state-action pair $(s, a) \in \widetilde{\mathcal{S}} \times \mathcal{A}$ is visited.

We then define the following episodic MDP with stochastic loss models. The models are specified as follows:

- Transitions occur uniformly at random to states in the next layer. Specifically, for any $(s, a) \in \mathcal{S}_h \times \mathcal{A}$, it holds that $P(s' \mid s, a) = 1/|\mathcal{S}_{h+1}|$ for all $s' \in \mathcal{S}_{h+1}$.

- All random losses $\ell_t(s, a)$ are assumed to be independent. For policy $\widetilde{\pi} \in \Pi_{\mathsf{det}}$ and $\varepsilon \in (0, 1/2]$, we consider the following two stochastic loss models:

$$\ell_t^{(\widetilde{\pi})}(s, a) \sim \begin{cases} \mathsf{Ber}(1/2) & \text{if } a = \widetilde{\pi}(s), \\ \mathsf{Ber}(1/2 + \varepsilon) & \text{otherwise}, \end{cases}$$

$$\ell_t^{(\widetilde{\pi}, \widetilde{s})}(s, a) \sim \begin{cases} \mathsf{Ber}(1/2) & \text{if } a = \widetilde{\pi}(s) \text{ and } s \ne \widetilde{s}, \\ \mathsf{Ber}(1/2 + \varepsilon) & \text{otherwise}. \end{cases}$$

These specifications define two episodic MDP instances, denoted by $\mathcal{M}(\widetilde{\pi})$ and $\mathcal{M}(\widetilde{\pi}, \widetilde{s})$, respectively.

Let $\mathbb{P}_{\widetilde{\pi}}$ and $\mathbb{P}_{\widetilde{\pi}, \widetilde{s}}$ be the probability distribution induced by $\mathcal{M}(\widetilde{\pi})$ and $\mathcal{M}(\widetilde{\pi}, \widetilde{s})$, respectively. We also denote by $\mathbb{E}_{(\mathbb{P}_{\widetilde{\pi}})}[\cdot]$ and $\mathbb{E}_{(\mathbb{P}_{\widetilde{\pi}, \widetilde{s}})}[\cdot]$ the expectations under the MDPs induced by $\mathcal{M}(\widetilde{\pi})$ and $\mathcal{M}(\widetilde{\pi}, \widetilde{s})$, respectively.

**Lemma G.2.** *Suppose that $H \geq 3$, $A \geq 3$ and $T \geq \frac{SA}{8H}$. Then, for any policy $\{\pi_t\}_{t=1}^{T}$, there exists $\widetilde{\pi} \in \Pi_{\mathrm{det}}$ such that*

$$\max_{\pi \in \Pi} \mathbb{E}_{(\mathbb{P}_{\widetilde{\pi}})}[\mathsf{R}_T(\pi)] \geq c\sqrt{HSAT}.$$

*Here the expectation is with respect to $\mathbb{P}_{\widetilde{\pi}}$ and $c = \frac{\sqrt{2}}{16}\left(\frac{1}{2} - \frac{1}{A}\right)^2$.*

*Proof.* We can lower bound the regret under $\mathcal{M}(\widetilde{\pi})$ as

$$\max_{\pi \in \Pi} \mathbb{E}_{(\mathbb{P}_{\widetilde{\pi}})}[\mathsf{R}_T(\pi)] \geq \varepsilon(H-1)T - \varepsilon \sum_{s \in \widetilde{\mathcal{S}}} \mathbb{E}_{(\mathbb{P}_{\widetilde{\pi}})}[N_T(s, \widetilde{\pi}(s))]$$

$$= \frac{\varepsilon T}{S'}\left(S'(H-1) - \frac{S'}{T}\sum_{s \in \widetilde{\mathcal{S}}} \mathbb{E}_{(\mathbb{P}_{\widetilde{\pi}})}[N_T(s, \widetilde{\pi}(s))]\right)$$

$$= \frac{\varepsilon T}{S'}\left(S - 1 - \frac{S'}{T}\sum_{s \in \widetilde{\mathcal{S}}} \mathbb{E}_{(\mathbb{P}_{\widetilde{\pi}})}[N_T(s, \widetilde{\pi}(s))]\right). \tag{103}$$

In what follows, we will upper bound $\mathbb{E}_{\mathbb{P}_{\widetilde{\pi}}}[N_T(s, \widetilde{\pi}(s))]$. Note that the only difference between $\mathcal{M}(\widetilde{\pi})$ and $\mathcal{M}(\widetilde{\pi}, \widetilde{s})$ lies in the expected value of the loss at the state-action pair $(\widetilde{s}, \widetilde{\pi}(\widetilde{s}))$.

Then, using the fact that $\frac{S'}{T}N_T(s, \widetilde{\pi}(s)) \in [0, 1]$ for $s \neq s_0$ and Pinsker's inequality, for any $\widetilde{s} \in \mathcal{S} \setminus \{s_0\}$ we have

$$\frac{S'}{T}\mathbb{E}_{(\mathbb{P}_{\widetilde{\pi}})}[N_T(\widetilde{s}, \widetilde{\pi}(\widetilde{s}))] \leq \frac{S'}{T}\mathbb{E}_{(\mathbb{P}_{\widetilde{\pi}, \widetilde{s}})}[N_T(\widetilde{s}, \widetilde{\pi}(\widetilde{s}))] + \|\mathbb{P}_{\widetilde{\pi}, \widetilde{s}} - \mathbb{P}_{\widetilde{\pi}}\|_{\mathrm{TV}}$$

$$\leq \frac{S'}{T}\mathbb{E}_{(\mathbb{P}_{\widetilde{\pi}, \widetilde{s}})}[N_T(\widetilde{s}, \widetilde{\pi}(\widetilde{s}))] + \sqrt{\frac{1}{2}\mathsf{KL}(\mathbb{P}_{\widetilde{\pi}, \widetilde{s}}, \mathbb{P}_{\widetilde{\pi}})}. \tag{104}$$

Then, from the chain rule of the KL divergence, we can evaluate the KL divergence in the last inequality as

$$\mathsf{KL}(\mathbb{P}_{\widetilde{\pi}, \widetilde{s}}, \mathbb{P}_{\widetilde{\pi}}) = \mathbb{E}_{(\mathbb{P}_{\widetilde{\pi}, \widetilde{s}})}[N_T(\widetilde{s}, \widetilde{\pi}(\widetilde{s}))]\, \mathrm{kl}(1/2 + \varepsilon, 1/2)$$

$$\leq 4\varepsilon^2 \mathbb{E}_{(\mathbb{P}_{\widetilde{\pi}, \widetilde{s}})}[N_T(\widetilde{s}, \widetilde{\pi}(\widetilde{s}))]. \tag{105}$$

where we used Lemma G.1. Taking the uniform average over $\Pi_{\mathrm{det}}$ for the RHS of (105), for any $\widetilde{s} \in \widetilde{\mathcal{S}}$ we have

$$\mathbb{E}_{\widetilde{\pi} \sim \mathsf{Unif}(\Pi_{\mathrm{det}})}\left[\mathbb{E}_{(\mathbb{P}_{\widetilde{\pi}, \widetilde{s}})}[N_T(\widetilde{s}, \widetilde{\pi}(\widetilde{s}))]\right]$$

$$= \sum_{a \in \mathcal{A}} \Pr[\widetilde{\pi}(\widetilde{s}) = a]\mathbb{E}_{\widetilde{\pi} \sim \mathsf{Unif}(\Pi_{\mathrm{det}})}\left[\mathbb{E}_{(\mathbb{P}_{\widetilde{\pi}, \widetilde{s}})}[N_T(\widetilde{s}, \widetilde{\pi}(\widetilde{s}))]\Big|\widetilde{\pi}(\widetilde{s}) = a\right]$$

$$= \frac{1}{A}\sum_{a \in \mathcal{A}} \mathbb{E}_{\widetilde{\pi} \sim \mathsf{Unif}(\Pi_{\mathrm{det}})}\left[\mathbb{E}_{(\mathbb{P}_{\widetilde{\pi}, \widetilde{s}})}[N_T(\widetilde{s}, a)]\right]$$

$$= \frac{T}{S'A}, \tag{106}$$

where the last equality follows from the definition of $N_T$. By summing over $\widetilde{s} \in \widetilde{\mathcal{S}}$ in (106),

$$\sum_{\widetilde{s} \in \widetilde{\mathcal{S}}} \mathbb{E}_{\widetilde{\pi} \sim \mathsf{Unif}(\Pi_{\mathrm{det}})}\left[\mathbb{E}^{(\widetilde{\pi}, \widetilde{s})}[N_T(\widetilde{s}, \widetilde{\pi}(\widetilde{s}))]\right] = \frac{(S-1)T}{S'A} \leq \frac{ST}{S'A}. \tag{107}$$

Using the last inequality, we also have

$$\sum_{\widetilde{s} \in \widetilde{\mathcal{S}}} \mathbb{E}_{\widetilde{\pi} \sim \mathsf{Unif}(\Pi_{\mathsf{det}})} \left[ \sqrt{\frac{1}{2} \mathsf{KL}(\mathbb{P}_{\widetilde{\pi}, \widetilde{s}}, \mathbb{P}_{\widetilde{\pi}})} \right]$$

$$\leq \varepsilon \sum_{\widetilde{s} \in \widetilde{\mathcal{S}}} \mathbb{E}_{\widetilde{\pi} \sim \mathsf{Unif}(\Pi_{\mathsf{det}})} \left[ \sqrt{2 \mathbb{E}_{(\mathbb{P}_{\widetilde{\pi}, \widetilde{s}})} [N_T(\widetilde{s}, \widetilde{\pi}(\widetilde{s}))]} \right]$$

$$\leq \varepsilon \sqrt{2(S-1) \sum_{\widetilde{s} \in \widetilde{\mathcal{S}}} \mathbb{E}_{\widetilde{\pi} \sim \mathsf{Unif}(\Pi_{\mathsf{det}})} \left[ \mathbb{E}_{(\mathbb{P}_{\widetilde{\pi}, \widetilde{s}})} [N_T(\widetilde{s}, \widetilde{\pi}(\widetilde{s}))] \right]}$$

$$\leq \varepsilon \sqrt{\frac{2HST}{A}}, \tag{108}$$

where the first inequality follows from (105), the second inequality follows from the Cauchy–Schwarz inequality and Jensen's inequality, and the last inequality follows from (107).

Therefore, by (104), (107) and (108),

$$\sum_{\widetilde{s} \in \widetilde{\mathcal{S}}} \frac{S'}{T} \mathbb{E}_{\widetilde{\pi} \sim \mathsf{Unif}(\Pi_{\mathsf{det}})} [\mathbb{E}_{\mathbb{P}_{\widetilde{\pi}}} [N_T(\widetilde{s}, \widetilde{\pi}(\widetilde{s}))]] \leq \frac{S}{A} + \varepsilon \sqrt{\frac{2HST}{A}}. \tag{109}$$

Finally, combining everything together, we have

$$\max_{\widetilde{\pi} \in \Pi_{\mathsf{det}}} \left\{ \max_{\pi \in \Pi} \mathbb{E}_{(\mathbb{P}_{\widetilde{\pi}})} [\mathsf{R}_T(\pi)] \right\} \geq \mathbb{E}_{\widetilde{\pi} \sim \mathsf{Unif}(\Pi_{\mathsf{det}})} \left[ \left\{ \max_{\pi \in \Pi} \mathbb{E}_{(\mathbb{P}_{\widetilde{\pi}})} [\mathsf{R}_T(\pi)] \right\} \right]$$

$$\geq \frac{\varepsilon T}{S'} \left( \frac{S}{2} - \frac{S'}{T} \sum_{s \in \widetilde{\mathcal{S}}} \mathbb{E}_{\widetilde{\pi} \sim \mathsf{Unif}(\Pi_{\mathsf{det}})} \left[ \mathbb{E}_{(\mathbb{P}_{\widetilde{\pi}})} [N_T(s, \widetilde{\pi}(s))] \right] \right)$$

$$\text{(by (103) and } S - 1 \geq \frac{S}{2} \text{ when } H \geq 3\text{)}$$

$$\geq \frac{\varepsilon T}{S'} \left( \frac{S}{2} - \frac{S}{A} - \varepsilon \sqrt{\frac{2HST}{A}} \right) \qquad \text{(by (109))}$$

$$\geq \frac{\varepsilon HT}{2} \left( \frac{1}{2} - \frac{1}{A} - \varepsilon \sqrt{\frac{2HT}{SA}} \right). \qquad (\frac{S}{S'} \geq \frac{H}{2} \text{ when } H \geq 3)$$

Choosing the optimal $\varepsilon = (\frac{1}{4} - \frac{1}{2A}) \sqrt{\frac{SA}{2HT}}$, which lies in $(0, 1/2)$ whenever $A \geq 3$ and $T \geq \frac{SA}{8H}$, we obtain

$$\max_{\widetilde{\pi} \in \Pi_{\mathsf{det}}} \left\{ \max_{\pi \in \Pi} \mathbb{E}_{(\mathbb{P}_{\widetilde{\pi}})} [\mathsf{R}_T(\pi)] \right\} \geq c\sqrt{HSAT},$$

where $c = \frac{\sqrt{2}}{16} \left( \frac{1}{2} - \frac{1}{A} \right)^2$. □

This lower bound is useful for deriving the regret lower bound $\max_{\pi \in \Pi} \mathbb{E}[\mathsf{R}_T(\pi)] \geq \Omega(\sqrt{HSAT})$ for online episodic tabular MDPs with adversarial losses.

### G.2. Proof of Theorem 6.1

Here we provide the proof of Theorem 6.1.

**Theorem G.3** (First-order lower bound). *Suppose that $H \geq 3$, $A \geq 3$, $T \geq \frac{SA}{8H}$ and $\alpha \in \left[ \frac{\lceil SA/8H \rceil}{T}, 1 \right]$. Then, for any policy $\{\pi_t\}_{t=1}^T$, there exists an episodic MDP with adversarial losses satisfying*

$$L^\star = \min_{\pi \in \Pi} \sum_{t=1}^T V^\pi(s_0; \ell_t) \leq \alpha HT$$

*such that*

$$\max_{\pi \in \Pi} \mathbb{E}[\mathsf{R}_T(\pi)] \geq \Omega(\sqrt{\alpha HSAT}) = \Omega(\sqrt{SAL^\star}).$$

*Proof.* Fix any $\alpha \in \left[\frac{\lceil SA/8H \rceil}{T}, 1\right]$ and split the horizon into an active phase $t = 1, \ldots, \lfloor \alpha T \rfloor$ and an inactive phase $t = \lfloor \alpha T \rfloor + 1, \ldots, T$. In the inactive phase, we set all losses to zero. As a result, it holds that

$$L^\star \leq \alpha HT.$$

On the other hand, by applying Lemma G.2 to the active phase, we obtain

$$\max_{\pi \in \Pi} \mathbb{E}[\mathsf{R}_T(\pi)] \geq \Omega(\sqrt{\alpha HSAT}) + 0 = \Omega(\sqrt{\alpha HSAT}).$$

$\square$

**Theorem G.4** (Second-order lower bound). *Suppose that $H \geq 3$, $A \geq 3$, $T \geq \frac{SA}{8H}$ and $\alpha \in \left[\frac{\lceil SA/8H \rceil}{T}, 1\right]$. Then, for any policy $\{\pi_t\}_{t=1}^T$, there exists an episodic MDP with adversarial losses satisfying*

$$Q_\infty = \min_{\ell^\star \in [0,1]^{S \times A}} \sum_{t=1}^T \sum_{h=0}^{H-1} \|\ell_t(h) - \ell^\star(h)\|_\infty^2 \leq \alpha HT$$

*such that*

$$\max_{\pi \in \Pi} \mathbb{E}[\mathsf{R}_T(\pi)] \geq \Omega(\sqrt{\alpha HSAT}) = \Omega(\sqrt{SAQ_\infty}).$$

*Proof.* Fix any $\alpha \in \left[\frac{\lceil SA/8H \rceil}{T}, 1\right]$ and split the horizon into an active phase $t = 1, \ldots, \lfloor \alpha T \rfloor$ and an inactive phase $t = \lfloor \alpha T \rfloor + 1, \ldots, T$. In the inactive phase, we set all losses to zero. As a result, it holds that

$$Q_\infty = \min_{\ell^\star \in [0,1]^{S \times A}} \sum_{t=1}^T \sum_{h=0}^{H-1} \|\ell_t(h) - \ell^\star(h)\|_\infty^2 \leq \sum_{t=1}^T \sum_{h=0}^{H-1} \|\ell_t(h)\|_\infty^2 \leq \alpha HT.$$

On the other hand, by applying Lemma G.2 to the active phase, we obtain

$$\max_{\pi \in \Pi} \mathbb{E}[\mathsf{R}_T(\pi)] \geq \Omega(\sqrt{\alpha HSAT}) + 0 = \Omega(\sqrt{\alpha HSAT}).$$

$\square$

**Theorem G.5** (Path-length lower bound). *Suppose that $H \geq 3$, $A \geq 3$, $T \geq \frac{SA}{8H}$ and $\alpha \in \left[\frac{\lceil SA/8H \rceil}{T}, 1\right]$. Then, for any policy $\{\pi_t\}_{t=1}^T$, there exists an episodic MDP with adversarial losses satisfying*

$$V_1 = \mathbb{E}\left[\sum_{t=1}^{T-1} \|\ell_{t+1} - \ell_t\|_1\right] \leq \alpha SAT$$

*such that*

$$\max_{\pi \in \Pi} \mathbb{E}[\mathsf{R}_T(\pi)] \geq \Omega(\sqrt{\alpha HSAT}) = \Omega(\sqrt{HV_1}).$$

*Proof.* Fix any $\alpha \in \left[\frac{\lceil SA/8H \rceil}{T}, 1\right]$ and split the horizon into an active phase $t = 1, \ldots, \lfloor \alpha T \rfloor$ and an inactive phase $t = \lfloor \alpha T \rfloor + 1, \ldots, T$. In the inactive phase, we set all losses to zero. As a result, it holds that

$$V_1 = \mathbb{E}\left[\sum_{t=1}^{T-1} \|\ell_{t+1} - \ell_t\|_1\right] \leq \alpha SAT.$$

On the other hand, by applying Lemma G.2 to the active phase, we obtain

$$\max_{\pi \in \Pi} \mathbb{E}[\mathsf{R}_T(\pi)] \geq \Omega(\sqrt{\alpha HSAT}) + 0 = \Omega(\sqrt{\alpha HSAT}).$$

$\square$

## G.3. Proof of Theorem 6.2

Here we provide the proof of Theorem 6.2.

**Theorem G.6.** *Suppose that $H \geq 3$, $A \geq 3$, $T \geq \frac{SA}{8H}$, and $\alpha \in (0, \frac{1}{4}]$. Then, for any policy $\{\pi_t\}_{t=1}^T$, there exists an episodic MDP with stochastic losses satisfying*

$$\mathbb{V} := \max_{\pi} \mathbb{E}\left[\sum_{s,a} q^\pi(s,a)\sigma^2(s,a)\right] \leq \alpha H$$

*such that*

$$\max_{\pi \in \Pi} \mathbb{E}[\mathsf{R}_T(\pi)] \geq \Omega(\sqrt{\alpha H S A T}) = \Omega(\sqrt{S A \mathbb{V} T}).$$

*Proof.* Let $\beta = 2\sqrt{\alpha} \in (0,1]$. We take the hard instance $\mathcal{M}(\widetilde{\pi})$ in Lemma G.2 and scale all losses by $\beta$. We define

$$\Lambda_t^{(\widetilde{\pi})}(s,a) = \beta \ell_t^{(\widetilde{\pi})}(s,a).$$

Let $\mathcal{M}^{(\beta)}(\widetilde{\pi})$ be the MDP obtained from $\mathcal{M}(\widetilde{\pi})$ by replacing each loss $\ell_t^{(\widetilde{\pi})}(s,a)$ with $\Lambda_t^{(\widetilde{\pi})}(s,a)$ while keeping the transition kernel unchanged. Let $\mathbb{P}_{\widetilde{\pi}}^{(\beta)}$ be the probability distribution induced by $\mathcal{M}^{(\beta)}(\widetilde{\pi})$. We also denote by $\mathbb{E}_{\mathbb{P}_{\widetilde{\pi}}^{(\beta)}}[\cdot]$ the expectation under the MDP induced by $\mathcal{M}^{(\beta)}(\widetilde{\pi})$.

Then, for any state-action pair $(s,a)$, we have

$$\mathsf{Var}\left(\Lambda_t^{(\widetilde{\pi})}(s,a)\right) = \beta^2 \mathsf{Var}\left(\ell_t^{(\widetilde{\pi})}(s,a)\right) \leq \frac{\beta^2}{4} = \alpha.$$

Therefore, the occupancy-weighted variance satisfies

$$\mathbb{V} = \max_{\pi} \mathbb{E}_{\mathbb{P}_{\widetilde{\pi}}^{(\beta)}}\left[\sum_{s,a} q^\pi(s,a)\sigma^2(s,a)\right] \leq \alpha \max_{\pi} \mathbb{E}_{\mathbb{P}_{\widetilde{\pi}}^{(\beta)}}\left[\sum_{s,a} q^\pi(s,a)\right] = \alpha H. \tag{110}$$

Thus, the constructed instance satisfies the variance condition $\mathbb{V} \leq \alpha H$.

Finally, since scaling all losses by $\beta$ scales the regret by the same factor, Lemma G.2 implies that there exists $\widetilde{\pi} \in \Pi_{\mathsf{det}}$ such that

$$\max_{\pi \in \Pi} \mathbb{E}_{\mathbb{P}_{\widetilde{\pi}}^{(\beta)}}[\mathsf{R}_T(\pi)] \geq 2c\sqrt{\alpha H S A T},$$

where $c = \frac{\sqrt{2}}{16}\left(\frac{1}{2} - \frac{1}{A}\right)^2$.

Finally, using $\alpha H \geq \mathbb{V}$ from (110), we have

$$\max_{\pi \in \Pi} \mathbb{E}_{\mathbb{P}_{\widetilde{\pi}}^{(\beta)}}[\mathsf{R}_T(\pi)] \geq 2c\sqrt{\alpha H S A T} \geq 2c\sqrt{S A \mathbb{V} T}.$$

This completes the proof. $\qquad\square$

