# OpenReview forum: "Data- and Variance-dependent Regret Bounds for Online Tabular MDPs"
_ICML.cc/2026/Conference — ICML 2026 regular_

### Official Review · Reviewer_iKWy · 2026-03-10

**Soundness:** 3
**Presentation:** 2
**Significance:** 3
**Originality:** 3
**Overall Recommendation:** 5
**Confidence:** 3

**Summary:**

This paper studies online episodic tabular MDPs with known transitions under bandit feedback. It proposes  a unified framework based on optimistic follow-the-regularized-leader with log-barrier regularization and adaptive learning rates. Two algorithmic approaches are developed: global optimization and policy optimization. In adversarial settings, the algorithms achieve first-order, second-order, and path-length regret bounds. In the stochastic setting, the algorithms further achieve variance-aware gap-independent and variance-aware gap-dependent bounds. The paper also provides matching lower bounds.

**Compliance With Llm Reviewing Policy:**

Affirmed.

**Final Justification:**

My concerns are addressed and I remain my original score.

**Key Questions For Authors:**

Please see the weaknesses above. Additionally, could the authors discuss how the results might extend to the setting with unknown transitions?

**Limitations:**

yes

**Strengths And Weaknesses:**

### Strengths:
1. This paper quantifies MDP complexity using first-order, second-order, path-length quantities. It proposes algorithms based on both global optimization and policy optimization and proves their upper bounds and lower bounds. The discussion is relatively comprehensive in the setting of online episodic MDPs with known transitions. This paper bridges the gap in online MDP literature by extending adaptivity via several data-dependent quantities.
2. The authors design a single best-of-the-worlds framework that works for both adversarial setting and stochastic setting.
3. For policy optimization, the authors utilize a margin-subtraction mechanism in the optimistic $Q$-function estimator to enforce a controlled bias. This refines the bound.
### Weakness:
1. The statement of Theorem 5.3 is inconsistent with its restatement in Appendix E.4, where the lower-order additive terms differ. It is unclear which version is actually supported by the analysis.
2. The lower bound theorems in Section 6 suppress the parameter conditions in Appendix. It weakens the precision of the theorem statements.
3. The policy-optimization results incur an extra factor in $H$ which is a known suboptimality gap. It would be helpful if the authors could discuss whether this extra $H$ dependence is inherent to the policy-optimization framework or whether it can potentially be improved.
4. In Section 3, the authors compare $\mathbb{V}$ with $Var_{\max}$, but the discussion of variance-aware RL under known transitions is somewhat limited. Could the authors provide further analysis or discussion about it?

---

> ### Author Rebuttal · Authors · 2026-03-30
>
> Thank you for your valuable time and constructive feedback. Below are our responses to the review.
>
> > The statement of Theorem 5.3 is inconsistent with its restatement in Appendix E.4, where the lower-order additive terms differ. It is unclear which version is actually supported by the analysis.
>
> Thank you for pointing this out. In the revised version, we will fix the issues in Appendix E.4 (lines 3031, 3108, and 3112). We will also ensure that the statement of Appendix E.4 is consistent with Theorem 5.3.
>
> > The lower bound theorems in Section 6 suppress the parameter conditions in the Appendix. It weakens the precision of the theorem statements.
>
> Due to space limitations, the statements in Section 6 are not fully specified, and the full parameter conditions are deferred to the appendix. We will revise the paper to include clearer and more precise theorem statements in Section 6.
>
> > The policy-optimization results incur an extra factor in $H$ which is a known suboptimality gap. It would be helpful if the authors could discuss whether this extra $H$ dependence is inherent to the policy-optimization framework or whether it can potentially be improved.
>
> We believe that the additional $H$-dependence is intrinsic to the policy optimization approach. Unlike global optimization over occupancy measures, policy optimization operates locally over policies and does not achieve global optimality over occupancy measures. As a result, the performance difference lemma reduces the problem to optimizing losses defined by Q-functions. Due to the Q-function estimation, the variance accumulates across the horizon, leading to an additional $H$ factor compared to global optimization. This type of $H$-dependence also appears in recent works such as Luo et al. (“Policy Optimization in Adversarial MDPs: Improved Exploration via Dilated Bonuses”, 2021) and Dann et al. (“Best of Both Worlds Policy Optimization”, 2023). Whether this dependence can be avoided remains an interesting future work and an open question in the theory of online MDPs.
>
> > In Section 3, the authors compare $\mathbb{V}$ with $\mathrm{Var}_\max$, but the discussion of variance-aware RL under known transitions is somewhat limited. Could the authors provide further analysis or discussion about it?
>
> Prior approaches incorporate both loss variance and transition-induced variance into the value function. In contrast, our variance measures avoid the additional variance introduced by transition randomness. Moreover, as further discussed in Appendix B.2, a key distinction is that our measures capture localized, state-dependent variance rather than aggregated trajectory-level variance. Consequently, compared to prior variance measures designed for unknown-transition settings, this leads to an $H^2$ improvement in the known-transition setting, and provides a more refined and suitable characterization of variance for the known-transition setting.
>
> > Additionally, could the authors discuss how the results might extend to the setting with unknown transitions?
>
> Extending our results to the unknown-transition setting requires controlling an additional transition-estimation error term in a data-dependent manner, which remains challenging. Following prior analyses (e.g., Jin et al., “The best of both worlds: stochastic and adversarial episodic MDPs with unknown transition”, 2021; Dann et al., “Best of Both Worlds Policy Optimization”, 2023), the regret can typically be decomposed into an FTRL-regret term and an additional error term arising from transition estimation. Our contribution focuses on the former by making an FTRL-regret term data-dependent, as in the known-transition setting. To extend our approach, it would be necessary to introduce a suitable data-dependent bonus to control the transition-induced error term. Overall, our analysis isolates the key difficulty and provides a foundation for future work on achieving refined guarantees under unknown transitions.

---

> > ### Author Rebuttal · Reviewer_iKWy · 2026-04-04
> >
> > My comments are resolved. I keep my positive score.

---

### Official Review · Reviewer_VDag · 2026-03-12

**Soundness:** 3
**Presentation:** 3
**Significance:** 3
**Originality:** 3
**Overall Recommendation:** 5
**Confidence:** 3

**Summary:**

This paper studies episodic tabular MDPs with bandit feedback under the assumption of known transition dynamics, aiming to obtain best-of-both-worlds regret guarantees across multiple regimes. The paper analyzes the adversarial regime, the stochastic regime, and the stochastic regime with adversarial corruption, using different optimistic prediction vectors and corresponding bounding terms to derive refined data-dependent guarantees. Two approaches are presented: (i) a global optimization method based on OFTRL over occupancy measures with log-barrier regularization and loss shifting, and (ii) a policy optimization method that combines dilated bonuses, virtual episodes, and a novel optimistic Q-function estimator.

**Compliance With Llm Reviewing Policy:**

Affirmed.

**Final Justification:**

The rebuttal addresses my concerns and I maintain my initial rating.

**Key Questions For Authors:**

1. Although some attribution is provided in the introduction and Section 5.1, it remains somewhat unclear which components of the methods are genuinely new and which are adaptations of existing tools, particularly in the global optimization approach. Could the authors briefly clarify which technical components constitute the main new contributions and which are primarily inherited from prior work?
2. Can the algorithm select the predictor $m_t$ automatically? In the paper, different regimes appear to require different predictors (e.g., a gradient-descent-style predictor versus an empirical-mean predictor). In practice, however, one typically does not know in advance whether the environment is closer to stochastic or adversarial (or how much corruption is present). Is there a possible way to make the choice of predictors in data-driven ways so that the algorithm can adaptively choose the predictors while still retaining the desired best-of-both-worlds guarantees?
3. What happens if the global optimization problem is solved only approximately? In practice, solving the occupancy-measure optimization exactly in every episode may be computationally costly. If only an approximate solution is computed, how would the regret guarantees change? In particular, do the refined bounds degrade gracefully, for example by introducing an additional approximation-error term?

**Limitations:**

yes.

**Strengths And Weaknesses:**

*Strengths:
1. It is appealing to obtain best-of-both-worlds guarantees, where a single algorithm performs well not only in the fully adversarial setting but also in the stochastic setting and the stochastic setting with adversarial corruption. The regret bounds adapt accordingly across these regimes.
2. The regret bounds have refined forms that depend on several data-dependent quantities, including a second-order term, a path-length measure, and a variance-based measure. These quantities provide a more detailed characterization of the problem complexity.
3. While obtaining such refined data-dependent bounds is relatively well understood in the full-information setting, achieving similar guarantees in the bandit setting is more challenging and less well developed. The paper succeeds in obtaining such results for bandit MDPs, which represents a meaningful advancement for this problem.
4. From a theoretical standpoint, the results appear technically strong, with rigorous and reasonably clear proofs. They contribute to the theoretical understanding of the fundamental problem of online reinforcement learning.

*Weaknesses:
1. The results rely on the assumption that the transition dynamics are known. This is a fairly strong assumption and may limit the broader applicability of the results. It would be helpful to discuss whether standard techniques for estimating transition dynamics could be incorporated, and how doing so might affect the regret guarantees.
2. The global optimization approach may be difficult to scale in practice, since optimizing over occupancy measures can be computationally expensive. It would be useful to discuss practical methods for solving this optimization problem and how approximate optimization might affect the guarantees.
3. The policy optimization approach is computationally more efficient, but it incurs additional horizon-dependent factors in the regret bounds compared to the global optimization approach. This raises the question of whether these extra factors are inherent or whether improved algorithms could avoid them.

---

> ### Author Rebuttal · Authors · 2026-03-30
>
> Thank you for your valuable time and thoughtful feedback. Below are our responses to the review.
>
> > The results rely on the assumption that the transition dynamics are known. …
>
> We agree that extending our results to the unknown-transition setting is an important and challenging direction. As discussed in the introduction, unknown transitions introduce additional estimation errors and exploration requirements, which significantly complicate obtaining data-dependent guarantees. At the same time, following prior analyses (e.g., Jin et al., “The best of both worlds: stochastic and adversarial episodic MDPs with unknown transition”, 2021; Dann et al., “Best of Both Worlds Policy Optimization”, 2023), the regret can typically be decomposed into an FTRL-regret term and an additional error term arising from transition estimation. Our contribution focuses on the former by making the FTRL regret term data-dependent, as in the known-transition setting. Therefore, in scenarios where the transition-induced error is small (e.g., when the transition model has favorable structure), our results already yield refined guarantees. More importantly, our analysis separates the problem into an FTRL-regret term and a transition-estimation error term, where we resolve the former while the latter remains a key challenge, thereby providing a useful foundation for future work.
>
> > Although some attribution is provided in the introduction and Section 5.1, it remains somewhat unclear which components of the methods are genuinely new and which are adaptations of existing tools, particularly in the global optimization approach. Could the authors briefly clarify which technical components constitute the main new contributions and which are primarily inherited from prior work?
>
> Thanks for bringing up this important point.
>
> For global optimization, the overall framework and analysis largely follow prior work (Jin et al., “The best of both worlds: stochastic and adversarial episodic MDPs with unknown transition”, 2021). Our main technical contribution lies in adapting the analysis to obtain refined data-dependent guarantees.  In particular, we use a log-barrier regularizer and extend the shifting function to fit the optimistic FTRL framework.
>
> For policy optimization, we discuss the main technical challenge and its resolution in the paragraph “New Q-function estimator” (p.7, line 370), but we will revise the paper so that this point is stated more explicitly. The key issue is that a naive extension of optimistic FTRL to MDPs leads to Q-function estimators with potentially negative bias, which can be amplified by the optimal policy and makes it difficult to obtain regret guarantees. Moreover, as in prior work (Dann et al., “Best of Both Worlds Policy Optimization”, 2023), virtual episodes are needed to stabilize the learning rate, but incorporating optimistic predictions breaks unbiasedness.
>
> To address these challenges, we introduce an optimistic estimator with an additional correction term in the Q-function (defined in Eq. (10)). We keep the estimator unbiased in real episodes, while allowing a small controlled bias in virtual episodes since their number is only logarithmic. This design balances optimism and unbiasedness in the MDP setting, which is essential for our analysis.
>
> > Can the algorithm select the predictor $m_t$ automatically? …
>
> We believe that automatically selecting the predictor $m_t$ is non-trivial. In our setting, different predictors are designed to achieve different types of guarantees (path-length-dependent (adversarial) vs. variance-gap–dependent (stochastic)), and it is generally unclear how to select among them without prior knowledge of the environment. Still, for either choice of predictor $m_t$, our algorithm guarantees strong performance across regimes, achieving logarithmic and variance-dependent gap-independent bounds in the stochastic setting and first- and second-order bounds in the adversarial setting, as shown in Tables 1 and 2. We consider this an interesting direction for future work.
>
> > What happens if the global optimization problem is solved only approximately? …
>
> When the global optimization problem is solved approximately, an additional term corresponding to the optimization error would appear in the regret bound. By ensuring that this error is sufficiently small (e.g., at a polylogarithmic scale in T), the refined data-dependent guarantees can still be preserved. We also note that the occupancy-measure set $\Omega (P)$ is a polytope with $O(SA)$ constraints and can be solved efficiently using convex optimization techniques (Zimin and Neu, “Online Learning in Episodic Markovian Decision Processes by Relative Entropy Policy Search”, 2013). Moreover, prior studies on occupancy-measure optimization often employ similar or the same regularizers (e.g., Jin et al., “No-Regret Online Reinforcement Learning with Adversarial Losses and Transitions”, 2023).

---

> > ### Author Rebuttal · Reviewer_VDag · 2026-04-02
> >
> > Thanks for the detailed response. I have no further questions and will retain my original score.

---

### Official Review · Reviewer_9JFD · 2026-03-12

**Soundness:** 3
**Presentation:** 2
**Significance:** 3
**Originality:** 2
**Overall Recommendation:** 4
**Confidence:** 2

**Summary:**

This paper studies online finite-horizon tabular MDPs with known transitions and investigates whether a single algorithm can simultaneously achieve refined data-dependent regret bounds in adversarial environments and variance-dependent guarantees in stochastic environments. To this end, the authors introduce a family of complexity measures: first-order, second-order, and path-length quantities for the adversarial regime, and occupancy-weighted variance measures for the stochastic regime, and develop two OFTRL-based algorithms with log-barrier regularization. The first operates via global optimization over occupancy measures; the second is a policy-optimization method that introduces a new optimistic Q-function estimator. The paper establishes first-order, second-order, and path-length regret bounds in the adversarial setting, and variance-aware gap-independent as well as polylogarithmic gap-dependent bounds under adversarial corruption. Lower bounds are also provided for the global-optimization route, establishing near-optimality with respect to several of the proposed complexity measures.

**Compliance With Llm Reviewing Policy:**

Affirmed.

**Final Justification:**

My concerns are mostly resolved in the rebuttal. I keep my positive score unchanged.

**Key Questions For Authors:**

See weaknesses.

**Limitations:**

Yes

**Strengths And Weaknesses:**

This is a strong and technically theory paper. Its main strength is breadth: it does not only target the usual best-of-both-worlds adversarial/stochastic dichotomy, but also tries to capture richer problem-dependent structure such as second-order and path-length quantities on the adversarial side and variance-aware, gap-dependent behavior on the stochastic side. I also like that the paper develops both a global-optimization route and a policy-optimization route, which makes the contribution broader than a single algorithmic template. The lower-bound section further strengthens the significance of the work, especially for the global-optimization results, since it helps justify that the upper bounds are close to the right scale.

The main weaknesses lie in the presentation and the readability of the proofs. The paper is dense and highly notational, and much of the technical burden is deferred to the appendix. As a result, it is difficult to distinguish the core ideas of the proof from technical bookkeeping. In addition, the authors do not clearly discuss the main technical challenges or explain how these challenges are resolved, which makes it hard to assess the true technical novelty of the work. Overall, the paper feels more like a combination of existing ideas, rather than a genuinely new conceptual contribution.

On soundness, I did not find an obvious fatal flaw, but I do think the appendix should be cleaned up before publication. In particular, in the proof of Theorem 5.3, the final displayed bound after invoking Lemma F.15 appears inconsistent with the theorem statement: the line shown in the appendix seems to repeat a corruption-related remainder while omitting the  $\sqrt{U_{var} C}$ term that the lemma would naturally suggest.

A second limitation is scope. The results are proved for the known-transition setting, and the introduction itself acknowledges that extending comparable refined guarantees to unknown transitions is substantially harder and remains open in several directions. This does not reduce the technical value of the current paper, but it does narrow the practical and conceptual reach of the results compared with what one might hope for in general online RL.

---

> ### Author Rebuttal · Authors · 2026-03-30
>
> We are grateful for taking your valuable time to review our paper and provide helpful comments. Below are our responses to the review.
>
> > The main weaknesses lie in the presentation and the readability of the proofs. The paper is dense and highly notational, and much of the technical burden is deferred to the appendix. As a result, it is difficult to distinguish the core ideas of the proof from technical bookkeeping.
>
> Thank you for the comment. We agree that this is a valid concern. At the same time, the settings of online MDPs are inherently complex, and our paper addresses both global optimization and policy optimization, which leads to dense notation and many technical details being deferred to the appendix. In this line of work, the notation tends to become heavy even in prior studies because of the complexity of the problem setting. That said, we believe that the core proof ideas are already presented in the main text, and we will revise the paper to make these ideas more clearly distinguishable from the technical details.
>
> > In addition, the authors do not clearly discuss the main technical challenges or explain how these challenges are resolved, which makes it hard to assess the true technical novelty of the work. Overall, the paper feels more like a combination of existing ideas, rather than a genuinely new conceptual contribution.
>
> Thanks for this important comment. We discuss the main technical challenge and its resolution in the paragraph “New Q-function estimator” (p.7, line 370), but we will revise the paper so that this point is stated more explicitly. The key issue is that a naive extension of optimistic FTRL to MDPs leads to Q-function estimators with potentially negative bias, which can be amplified by the optimal policy and makes it difficult to obtain regret guarantees. Moreover, as in prior work (Dann et al., “Best of Both Worlds Policy Optimization”, 2023), virtual episodes are needed to stabilize the learning rate, but incorporating optimistic predictions breaks unbiasedness.
>
> To address these challenges, we introduce an optimistic estimator with an additional correction term in the Q-function (defined in Eq. (10)). We keep the estimator unbiased in real episodes, while allowing a small controlled bias in virtual episodes since their number is only logarithmic. This design balances optimism and unbiasedness in the MDP setting, which is essential for our analysis. We will revise the paper to more clearly highlight these challenges and the corresponding technical contributions in the main text.
>
> > In particular, in the proof of Theorem 5.3, the final displayed bound after invoking Lemma F.15 appears inconsistent with the theorem statement: …
>
> Thank you for pointing this out. In the revised version, we will fix those typos in the appendix (lines 3031, 3108, and 3112). We will also fix the statement of Theorem E.15, so that it is consistent with Theorem 5.3.
>
> > A second limitation is scope. …
>
> We agree that extending our results to the unknown-transition setting is an important and challenging direction. As discussed in the introduction, unknown transitions introduce additional estimation errors and exploration requirements, which significantly complicate obtaining data-dependent guarantees. At the same time, following prior analyses (e.g., Jin et al., “The best of both worlds: stochastic and adversarial episodic MDPs with unknown transition”, 2021; Dann et al., “Best of Both Worlds Policy Optimization”, 2023), the regret can typically be decomposed into an FTRL-regret term and an additional error term arising from transition estimation. Our contribution focuses on the former by making the FTRL regret term data-dependent, as in the known-transition setting. Therefore, in scenarios where the transition-induced error is small (e.g., when the transition model has favorable structure), our results already yield refined guarantees. More importantly, our analysis separates the problem into an FTRL-regret term and a transition-estimation error term, where we resolve the former while the latter remains a key challenge, thereby providing a useful foundation for future work.

---

> > ### Author Rebuttal · Reviewer_9JFD · 2026-04-04
> >
> > Thanks for the rebuttal. My concerns are mostly resolved. The clarification on the technical challenge and the new optimistic Q-function estimator is helpful, and the appendix issue also seems fixable. I still think the presentation could be improved in the revision, especially to make the main proof ideas easier to follow, but overall I keep my positive score unchanged.

---

### Official Review · Reviewer_dyUb · 2026-03-13

**Soundness:** 3
**Presentation:** 3
**Significance:** 3
**Originality:** 3
**Overall Recommendation:** 4
**Confidence:** 1

**Summary:**

This paper studies finite-horizon episodic Markov decision processes (MDPs) with known transitions, and they consider the adversarial reward case and the stochastic reward case together. For the adversarial setting, the paper provides refined regret upper and lower bounds based on a first-order quantity, that is, the cumulative loss of the best fixed policy in hindsight ($L^*$), a second-order quantity ($Q_\infty$), and a path-length measure ($V_1$). For the stochastic reward regime, the paper considers variance-based complexity measures including the occupancy-weighted variance ($\mathbb{V}$) and the conditional occupancy-weighted variance ($\mathbb{V}^c$), as well as variance-dependent complexity measures including the maximum (unconditional) total variance and the maximum conditional total variance. Using these data- and variance-dependent measures, the paper shows that a single best-of-both-worlds algorithm achieves first-order, second-order, and path-length bounds in the adversarial regime and achieves variance-dependent bounds that are gap-independent or gap-dependent in the stochastic regime.

**Compliance With Llm Reviewing Policy:**

Affirmed.

**Final Justification:**

After reading other reviewers' comments and the rebuttals, I maintain my original score.

**Key Questions For Authors:**

- What are the challenges that make it hard to incorporate unknown transitions?
- Under unknown transitions, is it still possible to design an optimism-based algorithm that guarantees regret upper bounds?

**Limitations:**

yes

**Strengths And Weaknesses:**

Strength
- The paper is theoretically solid. It improves upon the existing works, by providing refined regret upper and lower bounds. Moreover, what is perhaps more interesting is that a single algorithm can achieve an improved regret upper bound as the theorem statements describe.
- The policy optimization-based algorithm has novel and sophisticated components. Although some algorithmic components are motivated from previous work, it seems nontrivial to design the current format of the algorithm that leads to the desired guarantees.
- On top of the algorithm design and associated regret upper bounds, the paper provides improved regret lower bounds as well,

Weakness
- Just to nudge some weakness points, as the authors mention in the paper, the paper assumes that the transition model is known to the decision maker. It would be great if the authors state and explain how the results of this paper can be extended to the unknown transition setting. Even for policy optimization, the authors may suggest a way to obtain data-dependent bounds.

---

> ### Author Rebuttal · Authors · 2026-03-30
>
> Thank you for your valuable time and for providing insightful feedback. Below are our responses to the review.
>
> > ...  It would be great if the authors state and explain how the results of this paper can be extended to the unknown transition setting ...
>
> > What are the challenges that make it hard to incorporate unknown transitions?
>
> The main challenge is that, under unknown transitions, one must control not only the FTRL-regret term but also an additional transition-estimation error term in a data-dependent manner. Within the standard regret analysis framework used in prior work, this requires introducing a transition-dependent bonus term and designing an appropriate optimistic estimator to control the error induced by transition estimation. In particular, when using Q-function estimators as in prior work, the error term depends on the confidence radii of the transition estimates, which is difficult to control in a data-dependent manner. Moreover, constructing such bonus terms in a way that is both data-dependent and computable is non-trivial. As a result, unlike in the known-transition setting, one must simultaneously control both the FTRL-regret term and the transition-estimation error term, which significantly complicates the analysis.
>
> > Under unknown transitions, is it still possible to design an optimism-based algorithm that guarantees regret upper bounds?
>
> We expect the answer to be yes, but it remains an open problem to achieve refined data-dependent guarantees in the unknown-transition setting. To our knowledge, there is no existing work that establishes first-order regret bounds for policy optimization under unknown transitions. In fact, even one of the strongest prior works (Dann et al., “Best of Both Worlds Policy Optimization”, 2023) explicitly identifies achieving such guarantees for policy optimization as an open problem. The main difficulty lies in controlling the additional transition-estimation error term in a data-dependent manner, as discussed above. While incorporating optimism may offer a promising direction toward achieving such refined guarantees, we currently do not know how to realize this. We therefore view this as an important direction for future work.

---

> > ### Author Rebuttal · Reviewer_dyUb · 2026-04-03
> >
> > Thank you for providing and outlining an idea in regards to extending the work to the unknown transition setting. I will keep my positive score.

---

### Decision · Program_Chairs · 2026-04-30

**Decision:**

Accept (regular)

**Comment:**

This paper is a strong theoretical work that obtains improved data dependent bounds in tabular MDPs.

A major limiting factor is that they require knowledge of the transition kernel, which turns this into an optimization problem over occupancy measures with trajectory feedback.

Since data-dependent bounds in MDPs is a hard and largely unexplored problem, I think it is fine to start with this simplistic setup. All reviewers agree that this is a clear accept, but probably not a "srong accept".